# A Review on Friction Stir Welding/Processing: Numerical Modeling

**DOI:** 10.3390/ma16175890

**Published:** 2023-08-28

**Authors:** Mostafa Akbari, Parviz Asadi, Tomasz Sadowski

**Affiliations:** 1Department of Mechanical Engineering, Technical and Vocational University (TVU), Tehran 14357-61137, Iran; mo-akbari@tvu.ac.ir; 2Department of Mechanical Engineering, Faculty of Engineering, Imam Khomeini International University, Qazvin 34148-96818, Iran; 3Department of Solid Mechanics, The Lublin University of Technology, Nadbystrzycka 40 Str., 20-216 Lublin, Poland

**Keywords:** friction stir welding, numerical models, strain and temperature distributions, material flow, force and torque

## Abstract

Friction stir welding (FSW) is a manufacturing process that many industries have adopted to join metals in a solid state, resulting in unique properties. However, studying aspects like temperature distribution, stress distribution, and material flow experimentally is challenging due to severe plastic deformation in the weld zone. Therefore, numerical methods are utilized to investigate these parameters and gain a better understanding of the FSW process. Numerical models are employed to simulate material flow, temperature distribution, and stress state during welding. This allows for the identification of potential defect-prone zones. This paper presents a comprehensive review of research activities and advancements in numerical analysis techniques specifically designed for friction stir welding, with a focus on their applicability to component manufacturing. The paper begins by examining various types of numerical methods and modeling techniques used in FSW analysis, including finite element analysis, computational fluid dynamics, and other simulation approaches. The advantages and limitations of each method are discussed, providing insights into their suitability for FSW simulations. Furthermore, the paper delves into the crucial variables that play a significant role in the numerical modeling of the FSW process.

## 1. Introduction

Friction stir welding (FSW) as a solid-state welding technique is attracting more attention in industrial applications and research [1]. These demands among engineers, manufacturers, and the market increase the desire for supply by the researchers and analysts in the diverse applications of FSW. However, experimental research imposition a very high rate of time and cost. In this situation, the simulation and modeling methods will lead us to a better, in-depth, low-cost, and fast cognition of the process.

Simulating the FSW process is a challenging task due to various factors such as intricate physical couplings between heat transfer and mechanics, significant strain rates and deformations in the stir zone (SZ) surrounding the pin, and the need to track the material flow. Despite its appeal, FSW simulation remains a complex problem that requires careful consideration of numerous interrelated factors [2,3]. Numerical simulation of the FSW process enables optimum selection of various process parameters such as rotational and traverse speeds [4,5,6], tool penetration depth and tilt angle [7,8,9,10,11], and tool design parameters such as tool shape and dimensions, shoulder and pin geometries [12,13,14,15,16,17,18,19,20,21], etc. The process of FSW simulating is a complex undertaking, as it requires accounting for the interplay between various thermal and mechanical factors. Nevertheless, advanced modeling methods have been developed that could effectively elucidate and forecast essential aspects of FSW process physics. These modeling techniques encompass a vast range of complexity, from elementary conduction heat transfer models to more intricate models that incorporate material flow, as well as fully coupled models that model heat transfer and viscoplastic flow to predict the distributions of temperature, strain, stress, and residual stress along with the microstructure and texture distributions [22,23,24,25].

FSW represents an advanced iteration of the conventional friction welding process and was initially conceived by The Welding Institute (TWI) in 1991 [26,27]. Since then, it has emerged as a leading metal joining technique and is widely regarded as the most crucial development in this field over the past two decades. Originally, FSW garnered significant attention as a solid-state joining process for aluminum alloys; however, its applications have now extended to more complex metallic materials, as well as plastics [28]. 

This process is carried out with the aid of a particular tool that comprises a shoulder and a profiled pin. This tool is gradually inserted into the joint between two substrates that are firmly clamped and supported by a backing plate. To provide a visual representation of FSW, Figure 1 presents a schematic diagram that showcases the cylindrical tool pin profile and a custom fixture designed to hold the plates securely. During the welding process, the tool’s shoulder makes strong contact with the upper surface of the workpiece while an axial force is applied. The heat required to soften the material is produced through plastic deformation and friction [29]. During the process, frictional heat is generated between the workpieces being joined and the welding tool (consisting of a shoulder and a pin). It is worth noting that the tool’s shoulder generates a greater amount of heat than the pin surface. Additionally, the rotation of the tool pin causes deformation or stirring within the materials, which generates additional heat [30,31]. When the material surrounding the pin becomes plasticized, the tool translation is applied, which leads to severe plastic deformation. This generates a robust flow of material that is rounded from the tool’s leading edge to its trailing edge, where it is forged into the joint by the tool shoulder assisted by the tilt angle. Ultimately, this results in forming a solid-state bond between two plates being joined [32]. Due to the combination of the tool’s traverse and rotational movements, the velocities of two symmetric points on the retreating and advancing sides during FSW are not identical. This asymmetry in motion results in a corresponding asymmetry in material flow and heat transfer between two sides of the weld [33,34]. It is worth noting that during the process, the directions of the tool traverse and rotational speeds are in alignment on the advancing side (AS), while they are opposite on the retreating side (RS), as shown in Figure 1b.

The developed friction stir processing (FSP) on the basis of FSW principles by Mishra et al. [28] is employed as a process for modifying metal microstructures. The basis and parameters of these two processes are similar, and there are slight differences between them. Also, there is no difference between the two methods in terms of microstructural changes and material properties.

FSP is a technique in which a rotating pin penetrates the workpiece and makes local microstructural modifications that can enhance the material properties. It should be noted that FSP is not intended for joining two parts; instead, its main objective is to modify the structure of the material. Through FSP, it is possible to improve the strength, modify the microstructure, and create a uniform grain size distribution. Additionally, sediment distribution can be altered, and surface composites can be produced as a result of this process [35]. FSP has been used for various applications, including producing surface composites, homogenizing parts produced by powder metallurgy, modifying microstructures in metal-based composites, and improving cast alloys’ properties [36].

Friction stir processing has so far been used with and without the use of additive particles. In the case of no additive particles, the process is precisely the same as the FSW, except that the join of the two pieces is not complete [37]. The rotating tool enters the material and modifies the microstructure of the material by applying high heat and strain but using additive particles to produce composites; this process is a bit different from FSW.

With the development of computers and FEM software, the development of numerical models to simulate the welding process has increased. Since the friction stir welding process is challenged by including severe plastic deformations, it is sometimes impossible to study the parameters such as material flow or temperature and residual stress distribution experimentally. Therefore, it is essential to develop numerical models to study these parameters. Based on the importance of FEM models in the FSW/FSP technique, this review article aims to cover the following outlets:Examining different numerical models for process simulation.Temperature, stress, and strain distributions during the process.Modeling the material flow in different types of FSW.Modeling the microstructural evolutions during the process.

## 2. Process Modeling Techniques

Considerable numerical research has been published on FSW recently, with significant attention given to residual stress, heat transfer, and material flow [38]. Two primary thermomechanical modeling techniques are used for numerical simulation: 1.Computational fluid dynamics (CFD) models [14,39,40,41,42,43,44,45,46,47,48];2.Computational solid mechanics (CSM) models. Within CSM, two principal methods have been employed: the arbitrary Lagrangian–Eulerian (ALE) [49,50,51,52,53,54,55,56,57,58,59,60,61,62] method, the coupled Eulerian–Lagrangian (CEL) approach [63,64,65,66,67,68,69,70,71,72,73,74,75,76,77,78,79,80,81,82,83,84], and the smoothed particle hydrodynamics (SPH) method.


CFD models are based on the assumption that the input heat is due to viscous loss. Equivalent density is usually derived from experimental results such as strain, stress, or temperature distribution. These techniques are deemed appropriate for metal flow solutions that require simultaneous evaluation. However, it is worth noting that CFD methods are often underutilized in other scenarios.

In addition to fluid models, some models based on solid mechanics have been developed to understand the mechanism of material flow in the FSW process. The ALE method prevents excessive distortion of the meshes in modeling the material flow around the pin. This method creates more accurate models to study the material flow than fluid models.

The CEL approach incorporates the best aspects of both the Lagrangian and Eulerian approaches [80]. Its key strength is the ability to control significant distortion issues that can occur during finite element simulations, like those in the FSW process. Mesh distortion in FSW simulation can be avoided via Eulerian analysis, where the material flows through the network of mesh while the nodes are fixed.

### 2.1. Computational Fluid Dynamics Models (CFD)

In the context of a Eulerian approach, the positions of nodes are assumed to be fixed in space while the material is observed to flow through non-deforming elements. Not all Eulerian elements will always be filled with material; several may be partially or entirely vacant (Figure 2). Therefore, the Eulerian material boundary, which generally does not match an element boundary, must be calculated for each time increment. To provide the material space to maneuver and flex, the Eulerian mesh is often a straightforward rectangular grid of components that are built to extend far beyond the boundaries of Eulerian material. Eulerian material is lost from the simulation if it goes beyond the Eulerian mesh. The Eulerian formulation is appropriate for resolving the fluid dynamics issue since it permits material to pass across the mesh. The accurate interface description offered by the Lagrangian formulation is lost by using a Eulerian reference frame, but it does prevent the problem of mesh tangling in the target.

In general, this approach provides a natural perspective on control volume problems in fluid mechanics. This arrangement is appropriate for steady-state analyses of forming processes in the field of solid mechanics, where the material flow is considerable, but the deformation on the free borders in the boundary shape is minimal. However, because the network does not change, applying this approach to represent the deformation of the free borders that arise throughout the deformation is challenging. Due to the erroneous formulation of free boundaries, the Eulerian approach is generally inappropriate for evaluating regions with shifting borders.

In CFD simulations aimed at investigating the thermal–mechanical conditions during FSW, the general governing equations for material flow and heat transfer for incompressible fluids are usually utilized. This approach assumes a rigid body and excludes the welding tool from the fluid computational domain. The workpiece is taken as a single-phase [85] or multiphase [86] viscous fluid in the literature. In this approach, the mass flow is regarded as an incompressible, non-Newtonian, viscoplastic material flow.

The continuity equation and momentum conservation equation governing the flow of incompressible, single-phase fluid are as follows [44]:(1)$$\frac{\partial \rho }{\partial t}+\nabla .(\rho \vec{v})=0$$
(2)$$\frac{\partial \rho \vec{v}}{\partial t}+\nabla .(\rho \vec{v}\vec{v})=-p+\nabla .(\mu (\nabla \vec{v}+\nabla {\vec{v}}^{T})$$
where ρ is the density, *μ* is the viscosity, $\overset{\to }{v}$ is the velocity vector, *p* is the pressure, and *t* is the flow time. The energy conservation equation was given by:(3)$$\frac{\partial \rho H}{\partial t}+\nabla .(\rho \vec{v}H)=\nabla .(k\nabla T)+S_{V}$$
where *H* is the enthalpy, *k* is the thermal conductivity, *T* is the temperature and *S_V_* is the spatial source term regarding the volumetric heat generation due to plastic deformation, which was described below. The enthalpy *H* was given by:(4)$$H=\int_{T_{ref}}^{T}C_{P}dT$$
where *C_P_* was the heat capacity, *T_ref_* was the reference temperature, and *T* was the temperature.

Colegrove et al. [39] utilized the CFD code FLUENT to model the two-dimensional material flow during FSW. Their newly created “slip” model relied on local shear stresses to control the interface conditions. Subrata Pal et al. [40] employed the polycrystalline cubic boron nitride pin to study the heat transfer and material flow during the FSW of SS304 using the three-dimensional CFD code FLUENT. Their simulation was accurate in predicting the FSW material flow characteristics. Tiwari et al. [87] investigated a modified material transfer and heat transfer model for FSW of DH36 steel, utilizing the Eulerian framework in a steady state. They modeled the material viscosity as a non-Newtonian viscoplastic fluid that varied according to flow stress and temperature. The researchers showed that the highest temperature occurred at the point where the sheet and shoulder met and that maximum temperatures varied along the direction of thickness. Using the CFD package FLUENT, Hasan et al. [41] created a couple-thermo flow model for friction stir welding to simulate the flash generation phenomena that take place during the FSW process. Single-phase flow models and multiple-phase flow models were contrasted. When a two-phase flow model was used, a considerable decrease in the pressure values was seen.

Savaş et al. [42] employed the Comsol Multiphysics CFD code to predict the three-dimensional metal flow during FSW of AA 2024-T3 aluminum alloy. Meanwhile, Colegrove et al. [43] utilized a CFD code to examine the material flow around a complex FSW tool and assessed the impact of the rotation speed and tool rake angle. The researchers found that their model overestimated the weld temperature due to an excessive amount of heat production.

Chen et al. [44] conducted a three-dimensional numerical simulation using CFD to investigate the plastic deformation and heat transfer behavior during the FSW of AA2024. They utilized both boundary shear stress (BSS) and boundary velocity (BV) models to evaluate their ability to predict temperature and material deformation in FSW. While there was a notable discrepancy between the predictions made by the experimental data and the BV models, the predicted geometry of the deformation zone by the BSS model was found to be congruent with the experimental findings.

Chen et al. [52] created a CFD-based thermomechanical coupled model to explore the heat production and the heat flux spatial distribution during the FSW of aluminum alloy. Mohan et al. [46] utilized CFD to investigate the temperature distribution, heat generation, and material flow within the stir zone during FSW at extremely high tool rotational speeds. The researchers assumed that the tool and workpiece were in a partial sliding–sticking contact condition, and they incorporated boundary conditions to account for the partial melting possibility at high rotational speeds. They found that plastic heat creation at high rotational speeds had a more significant impact on heat generation than frictional heat generation, and they observed that partial melting did not occur. Yang et al. [88] integrated the anisotropy of mechanical characteristics into a CFD model of the FSW process for the AA6061 alloy. The researchers evaluated the anisotropy distribution and the degree of drop in flow stress/yield strength because of the mechanical anisotropy based on the computed results. Pankaj et al. [89] utilized multiphase CFD simulation to simulate the dissimilar FSW of 6061-T6 aluminum alloy and DH36 shipbuilding steel. The researchers investigated how the rotational speed impacted the temperature and flow of plasticized material through the tool–material interaction. Yang et al. [90] simulated Al/Mg dissimilar FSW using the CFD technique. By treating the material mixing in the SZ as a functionally graded material (FGM) for the computation of associated thermophysical parameters, they were able to obtain a more accurate prediction of the production of Al/Mg FSW joints with greater precision.

Al/Cu dissimilar FSW was studied using FLUENT software by Kadian and Biswas [91], who found that the weld had an uneven velocity distribution. To model the Al/steel dissimilar FSW, Liu et al. [86] used the volume of fluid (VOF) technique in FLUENT. The AA6061 and TRIP steels’ different thermophysical properties led to the discovery of an asymmetric temperature field during the steel/Al dissimilar FSW. Using a CFD technique, Bokov et al. [92] investigated how tool shape affected the heating process in a steel/Al dissimilar FSW. They found that the material flow velocity on the aluminum side was higher than on the steel side. Using a CFD and volume of fluid technique, Zhang et al. [93] developed a multiphase model for studying the heat and mass transfer behavior in dissimilar FSW of AA2024-T4 aluminum alloy and TC4 titanium alloy.

Jiang et al. [94] created a model that utilizes the VOF approach to analyze the mixing of material and material flow in dissimilar FSW of Mg/Al alloys. The model incorporates mass, energy conservation, and momentum equations to evaluate material flow while considering quasi-steady state heat transfer conditions. To achieve this, the researchers assumed that materials behave as a non-Newtonian incompressible viscous fluid. Yang et al. [95] developed a CFD model while conducting ultrasonic vibration-enhanced FSW of 6061-T6 Al alloy. The model presumed that the material acted like a non-Newtonian and incompressible viscoplastic fluid. By comparing their estimated thermomechanically affected zone (TMAZ) boundary and thermal cycles with experimental measurements, the researchers were able to obtain fairly consistent results. The material flow during the FSW of the 5083 aluminum alloy and stainless steel was examined by Sadeghian et al. [96] using CFD-based simulations. They considered the heat generation in their numerical model based on the FSW tool shape, rotational speed, and yield shear stress of the parent material.

Carlone et al. [97] used the commercial ANSYS CFX package to solve a CFD-based model of the FSW and validate it against thermographic observations of the process. To study the impact of tool tilt angle on the material flow, temperature distribution, and heat generation in FSW, Zhai et al. [98] undertook both experimental work and numerical modeling based on CFD. The study found good agreement between the projected temperature field and TMAZ from the numerical simulation and experimental observation.

Su et al. [99] introduced an innovative asymmetrical boundary condition for the FSW process of AA2024-T4 alloy at the tool–workpiece contact interface. To investigate the thermal and plastic material flow characteristics, they utilized a three-dimensional computational fluid dynamics model and compared the behavior under both symmetrical and asymmetrical boundary conditions. Their findings revealed that the asymmetrical boundary condition outperformed the symmetrical counterpart in accurately predicting vertical plastic material flow, temperature fluctuations, and tunnel formation during FSW.

He et al. [100] conducted an analysis of the effects of threads and flats in Al-Cu dissimilar FSW by examining three distinct features of pin models. Utilizing CFD, they discovered that different pin profiles have unique strengthening impacts on plastic flow. The presence of threads was found to effectively agitate insufficient flow at the bottom of the pin, thereby facilitating longitudinal material movement. On the other hand, the inclusion of flats was observed to enhance upward material migration through screw guiding, playing a crucial role in augmenting horizontal plastic mixing. Ultimately, the combined influence of “thread + flat” exhibited a periodic strengthening effect on the mixing flow of the materials, further enhancing the overall welding process.

### 2.2. Numerical Models Based on Solid Mechanics (CMS)

The finite element method (FEM), in recent decades, has been accepted as one of the most effective simulation tools in designing and analyzing many phenomena as a standard tool in the engineering community. Some of the challenges that arise among finite element analysis in the field of solid mechanics problems are: choosing a comprehensive appropriate configuration to analyze the problem, providing a suitable method to prevent network distortion due to large deformations, contact modeling methods at contact surfaces, and expression of proper contact behavior with appropriate friction models.

One of the fundamental principles in the numerical simulation of processes that involve significant deformations in nonlinear solid mechanics is utilizing a suitable kinematic depiction for the continuously deforming environments. The Lagrangian and Eulerian views are the two primary classical perspectives regarding the kinematic description of a continuous environment.

When adopting a Lagrangian perspective, the configuration network remains attached to the material and monitors its deformation. This approach is widely used in solid mechanics and proves particularly effective when dealing with the unconstrained flow at free boundaries, interference between various materials, and materials for which the deformation history is critical. The use of this type of networking has significant drawbacks, including network and element distortions. The Lagrangian method makes it simple to monitor surfaces and apply boundary conditions since it allows the mesh and the material to move simultaneously. The material border always coincides with an element boundary since Lagrangian elements are constantly filled with a single material. The Lagrangian formulation functioning is depicted in Figure 3.

Gould and Feng [101] proposed a simple Lagrangian model to forecast the heat transfer and temperature profile inside the FSW process. Some models [25,52,55,56,58,102,103] that accurately anticipated the results of thermal analysis and stress used the Lagrangian approach with severe re-meshing. In the study by Buffa et al. [58], a coupled thermomechanical FEM in commercial DEFORM-3D software was used, along with a Lagrangian formulation and implicit approach. The welding seam was modeled in the model using continuum elements. Additionally, the impact of the welding parameters on strain rate, welding forces, temperature history, and material flow was investigated using a rigid viscoplastic material description. Finally, favorable agreements regarding welding forces and temperatures were reached. The temperature distribution across the welding center line was found to be roughly symmetrical and affected mainly by rotational speed. However, it was shown that the material flow in the stir zone was an asymmetrical phenomenon, primarily controlled by welding traverse and rotational speeds.

The Lagrangian technique was employed by Dong et al. [104] and Chao et al. [105] to compare the experimental and numerical achievements of the heat energy produced during FSW. In the model by Chao et al. [106], the tool was under a steady state condition while the plates were under a transient situation. An experimental temperature recording was performed to verify the findings. Utilizing the WELDSIM code, the workpiece was subjected to simulation. However, only half of the workpiece was modeled since a symmetric condition was presumed between the AS and the RS. The results revealed that nearly 95% of the heat produced during welding is directed toward the sheets, with the remaining 5% allocated to the welding tool. To address the issue of over-prediction in Lagrangian finite element models, some designs incorporate experimental power data to serve as an input parameter for the model.

Chen and Kovacevic [107] developed coupled thermomechanical modeling of FSW using the Lagrangian and re-meshing techniques. The model simulated the tool shoulder as the heat source. Additionally, the effect of the moving heat source on the temperature of the material was studied. The model has limitations in terms of stresses, strain rate, and forces because the influence of the pin was believed to be minimal. Using a thermomechanically coupled Lagrangian model, Akbari et al. [108] investigated the contribution of each part of the pin and shoulder to the formation of strain and heat. The findings demonstrate that frictional heat generation, as opposed to plastic deformation, is the primary process for raising workpiece temperature. Additionally, the tool shoulder produces roughly 90% of the heat; the tool pin cannot produce heat or strain in the samples on its own. A thermomechanical FSP simulation was created by Asadi et al. [109] utilizing the DEFORM 3D program based on Lagrangian implicit. They assumed that the tool is rigid, the material is characterized as a rigid viscoplastic material, and the friction is constant. They used the Arrhenius equation to correlate the flow stress with temperature and strain rate at high temperatures. For the simulation, they created a non-uniform mesh with automatic re-meshing. To achieve accuracy in the contact between the material and tool, fine elements with a mean size of 0.85 mm were located around the tool pin and under the shoulder (Figure 4).

Given the justifications, it may be inferred that neither the Lagrangian nor the Eulerian viewpoints are sufficient to model significantly deformed processes like friction stir welding. Both strategies are complementary, and the strength of each is the weakness of the other. Therefore, new configurations are more suited to replicating the FSW process because they can incorporate the advantages of Lagrangian and Eulerian networking.

The arbitrary Lagrangian–Eulerian (ALE) along with the coupled Eulerian–Lagrangian (CEL) are two numerical techniques used in computational mechanics to solve problems involving large deformations of materials or fluids. These techniques combine the benefits of both Eulerian and Lagrangian approaches to offer a more comprehensive solution. In the ALE approach, the network components in the space are neither fixed nor attached to the object or material being studied. Instead, they have the necessary motion to model the system’s dynamic behavior accurately. This can be particularly useful when simulating fluid–structure interactions or other problems where the mesh deformation needs to be tracked as part of the solution. By contrast, the CEL approach combines elements of both Lagrangian and Eulerian approaches by using different reference frames for different regions of the problem domain. This allows for greater flexibility in modeling complex systems, such as fluid flow around a moving object.

The ALE approach combines the two networking approaches discussed above with the relative mobility of the network components and the material. As a result, the material configuration and each network component can move independently. This element of the ALE approach is an effective tool for simulating processes in which the material experiences large-scale local deformation and large-scale unrestricted flow across free boundaries. This allows ALE to be turned into Lagrangian on free boundaries while remaining Eulerian in regions with significant deformation.

The mesh is regularly smoothed in intervals to minimize element distortion and preserve favorable aspect ratios of elements while also retaining the same mesh topology as a fundamental aspect of the adaptive meshing capability (Figure 5). The number of components and nodes, as well as their connectivity, remain constant. Both Lagrangian and Eulerian (transient and steady-state) issues can be analyzed using it.

In a Lagrangian description, the material domain utilizes material coordinates *X*, whereas the Eulerian formulation is characterized by the spatial domain with fixed spatial coordinates *x*. In the ALE description, a referential domain is employed, which consists of grid or mesh coordinates *χ*. It is crucial to establish biunivocal transformations between these domains. Spatial coordinates can be expressed as a function of the material coordinates *X*, and of time *t*(*x*(*X*,*t*)) or the mesh coordinates *χ* and of time *t*(*x*(*χ*,*t*)).

By taking the derivative of mesh coordinates and material with respect to time (*t*), while keeping the mesh coordinate or the material coordinate (*χ*) fixed, we can determine the mesh velocity and material velocity [110].
(5)$$v=\partial x(X,t)\partial t_{x},\overset{⌢}{v}=\partial x(\chi ,t)\partial t_{\chi }$$

Convective velocity refers to the discrepancy between material velocity and mesh velocity.
(6)$$c=v-\overset{⌢}{v}$$

In order to guarantee that the nodes located on the edge of the mesh remain on the solid boundary, the convective velocity’s normal projection must satisfy the following equation.
(7)$$c.n^{b}=0$$

Let (·) denote the scalar product of vectors, and *n^b^* is the normal to the boundary. The conventional convective derivative represents the material time derivative for any vector function *f_i_*.
(8)$$\partial f_{i}\partial t_{X}=\partial f_{i}\partial t_{x}+\partial x_{j}\partial t_{X}\partial f_{i}\partial x_{j}=\partial f_{i}\partial t_{x}+{\overset{⌢}{v}}_{j}\partial f_{i}\partial x_{j}$$

In the same manner, we are able to establish a definition for a reference time derivative as:(9)$$\partial f_{i}\partial t_{\chi }=\partial f_{i}\partial t_{x}+\partial x_{j}\partial t_{\chi }\partial f_{i}\partial x_{j}=\partial f_{i}\partial t_{x}+{\overset{⌢}{v}}_{j}\partial f_{i}\partial x_{j}$$

By combining the material derivative results
(10)$$\partial f_{i}\partial t_{X}=\partial f_{i}\partial t_{\chi }+c_{j}\partial f_{i}\partial x_{j}$$

The final term in the equation accounts for the convective influence caused by the relative movement between the substance and the mesh.

This relationship can be divided into two specific scenarios:
1.The Lagrangian description (*v_j_* = ${\overset{⌢}{v}}_{j}$) states that the material and mesh are interconnected.
(11)$$\partial f_{i}\partial t_{X}=\partial f_{i}\partial t_{\chi }$$2.Eulerian description (${\overset{⌢}{v}}_{j}$ = 0): the mesh is fixed.
(12)$$\partial f_{i}\partial t_{X}=\partial f_{i}\partial t_{\chi }+v_{j}\partial f_{i}\partial x_{j}$$


Utilizing the ALE method, a thermomechanical FEM model was fully coupled to simulate FSW. The model incorporated inelastic heat generation to enhance the accuracy of the numerical simulations [106]. Numerical simulations were conducted to simulate the temperature fields during FSW at 600 and 800 rpm rotational speeds. The results indicated that the numerical model was able to replicate the temperature distribution during the FSW process accurately. Xu and Deng [111] modeled the plastic strain and material flow using ABAQUS. The ALE-based model was used to develop the FE modeling of the process using an adaptive meshing technique, temperature-dependent material properties, and massive elastoplastic deformations. The temperature was defined as an input parameter to the model because it was not assumed to be a fully coupled thermomechanical model. It should be noted that the temperature specified was determined via an experiment.

Additionally, the velocity gradient at the pin side area was recorded. It was found that the temperature and velocity are higher on the AS rather than on the RS. Finally, the distributions of the equivalent plastic strain and the distribution of the microstructural zones were compared between the numerical findings and the experimental observations, and a satisfactory correlation was attained. 

Schmidt and Hattel [112] utilized a comprehensive 3D thermomechanical finite element model, which was fully coupled to capture the intricate interplay between heat transfer, material deformation, and mechanical loads during FSW. They leveraged ABAQUS/Explicit to implement the ALE method and Johnson–Cook material model, which enabled them to accurately predict the workpiece plastic deformation and failure behavior during welding. To simplify the connection between the rigid tool and workpiece, Schmidt and Hattel cleverly utilized the inherent flexibility of the FSW machine as a cylindrical volume model with inlet and outlet boundary conditions. This allowed for easier modeling of the machine rigidity and stiffness, which would have been more difficult to simulate accurately otherwise.

Mandal et al. [113] focused their modeling efforts on addressing the difficulties associated with temperatures and the high strain rates involved in FSW, which present significant numerical challenges when dealing with nonlinear materials. To address these issues, the researchers opted to use the Johnson–Cook elastoplastic constitutive law within the FE code ABAQUS to model the entire FSW. One of the main challenges in simulating is managing excessive element distortion, which can cause the simulation to terminate prematurely. Mandal et al. used an ALE approach with extensive re-meshing to mitigate this issue and ensure accurate results. By doing so, they were able to produce a sophisticated model that effectively captures the complex thermomechanical behavior of the material during FSW. 

By using the ALE technique, Assidi et al. [114] created an FE model for the FSW as it was employed by FORGE 3 FE software. The interaction between the material and the tool and the flash creation that takes place during welding were the main subjects of the investigation. This constitutive model employed Norton and Coulomb’s laws for friction contact modeling and the Hansel–Spitter material model. The investigation found that the distribution of welding forces and temperatures and actual data exhibited good agreement. The work gave researchers new insight into the FSW process regarding the incidence of surface flash and the circumstances at the tool–workpiece contact.

The CEL method is another finite element networking method utilized in procedures with significant plastic deformation like FSW. This technique avoids significant network distortion by modeling the sample using Eulerian relations. Lagrangian relations are also employed in the tool modeling. In this method, the welding process causes significant deformations in the Eulerian network, representing the material, without any issues with network distortion. Lagrangian can interact with Eulerian (Figure 6).

In this method, the nodes are moved, and the mapping process is performed after Lagrangian calculations. This method is a kind of simulation of Eulerian finite elements with a Lagrangian stage. The most important feature of the CEL approach is in the definition of free surfaces where the boundaries of matter do not match the boundaries of components. In the CEL approach, the components can be empty, to an occupied size, or filled with material (Figure 7) [63]. In this case, the volume of material within the components is measured by volume fraction, which is obtained from the ratio of the volume of material within the component to the total volume of that component. A part that is filled with material will have a volume fraction of one, and an empty part will have a volume fraction of zero. The component can be occupied by more than one type of material simultaneously.

In the CEL, the material domain was represented using the Eulerian formulation, which ensures the conservation of mass, energy, and momentum. Equation (13) delineates the mass conservation equation, addressing both the mass outflow rate and the rate of mass alterations within the control volume:(13)$$\frac{\partial \rho }{\partial t}+\nabla .(\rho v)=0$$
where *ρ* is material density; *v* is material velocity. The equation for momentum conservation (Equation (14)) states that the change in momentum of the system is equal to the sum of the spatial time derivative of the Cauchy stress tensor and the gravitational force acting on the system.
(14)$$\rho (\frac{\partial v}{\partial t}+\nabla .(v⊗v))=\nabla .\sigma +\rho g$$
where *σ* is the Cauchy stress tensor; *g* is the gravitational constant. The energy equation takes into account the work rate applied to each element, the heat transfer through conduction into the element, and the generation of heat within the element.
(15)$$\rho C_{p}(\frac{\partial T}{\partial t}+v.\nabla T)=\nabla .(K\nabla T)+\nabla .(\sigma .v)+\dot{Q}$$

*K* is the material thermal conductivity, *C_p_* is the specific heat, *T* is the temperature in the Kelvin scale, and $\dot{Q}$ represents the volumetric heat generation rate. Equations (13)–(15) discussed the Eulerian governing equations. Equation (16) presents these equations in a general conservation form.
(16)$$\frac{\partial \phi }{\partial t}+\nabla .\Phi =S$$
where Φ is the flux function, and *S* is the source term. The operator splitting algorithm breaks down Equation (16) into two parts: the Lagrangian step, which includes the source term, and the Eulerian step, which includes the convective term. This breakdown can be seen in Equations (17) and (18). To solve these equations, the CEL method is used in a sequential manner.
(17)$$\frac{\partial \varphi }{\partial t}=S$$
(18)$$\frac{\partial \varphi }{\partial t}+\nabla .\Phi =0$$

Figure 7 illustrates the schematic representation of the split operator for each step of the CEL method.

Many researchers have considered the coupled Eulerian–Lagrange method to model the friction stir welding process [65,115]. In this method, the material is modeled using Eulerian relations, and the welding tool is modeled using Lagrangian relations. The Eulerian region is formed in two parts, filled with material and the empty region, to carefully examine the flow of material and the formation of defects. The presence of an empty area is necessary to check for defects in the sample. 

The CEL approach has recently attracted the attention of researchers for FSW modeling. By tracking the material flow into and out of the computational/void domain using a 3D thermomechanical-based CEL approach, Bhattacharjee et al. [64] projected surface, sub-surface, and volumetric defects. Iordache et al. [69] suggested a method based on the CEL approach for identifying the best parameters for a butt-welded joint of copper plates. They drastically cut the simulation duration by employing the mass scaling technique. 

To explore the FSW of 1Cr11Ni2W2MoV steel, researchers Ragab et al. [82] developed a 3D thermomechanical finite element model utilizing the CEL approach. The team incorporated a velocity boundary condition, depicted in Figure 8, to control the material flow within the Eulerian domain.

The FSW of aluminum and brass was studied by Akbari et al. [63] using the CEL model. Their investigation appropriately recognized both the intermetallic composition of aluminum and brass and the influencing factors. To forecast potential circumstances that could result in defect creation, such as excess flash formation and tunnel cavities during the FSW, Ajri and Shin [116] built models using the CEL approach.

Akbari et al. [26] simulated the FSW of AA5083 to AA7075 aluminum alloys using a CEL approach. When the stir zone of the joint was compared to what the modeling had indicated, it was found that the CEL approach had accurately predicted how the material would mix in the SZ. This study applies the Lagrangian formulation to the tool, which is modeled as a rigid body, and the Eulerian domain to the material (AA5083 and AA7075). The Eulerian component was split into two areas containing alloys AA5083 and AA7075 (Figure 9). Then, two halves’ different materials were indicated by the VOF feature.

Chen et al. [117] utilized the CEL approach for simulation to investigate material flow at the interface between dissimilar steel and aluminum alloys during friction stir spot welding. They analyzed the distribution of material at different plunging depths and observed how the hook was formed during the welding process. Chauhan et al. [118] employed the CEL approach to predict the occurrence of faults in FSW. Through their model, they could accurately forecast the axial force, spindle torque, and tunnel defect that may arise during the process.

Al-Badour et al. [119] predicted material flow and volumetric flaws like the tunnel during the friction stir welding of aluminum using the CEL approach. Safari et al. [120] created the FEM model of the dissimilar FSW employing the CEL approach. Their findings show that a grooved pin design enhances material velocity around the FSW tool. Using a CEL approach, Ghiasvand et al. [77] studied the impact of FSW tool positioning parameters on the peak temperature in the dissimilar FSW of the alloys AA7075-T6 and AA6061-T6. Due to increased contact surfaces and friction between the sample and tool, the peak process temperature rose dramatically when the plunge depth was increased.

Grujicic et al. [53] explored dissimilar filler metal friction stir welding using a CEL approach to understand inter-material mixing and weld-flaw generation. Using a coupled Eulerian–Lagrangian methodology, Das et al. [70] looked into the material mixing, flaws in FSW of different materials, and the influence of progressive tool wear. With a maximum deviation of 1.2 mm, the created model accurately predicted surface tunnels, excessive flash formations, exit holes, and subsurface tunnel flaws. 

Lu et al. [121] conducted a study on the temperature distribution of 2219 aluminum alloy using a CEL approach. The experimental findings revealed that the model’s maximum relative error in predicting the peak temperature in the core area was 4.4%, with an average relative error of 2.46%. By considering the reasonable welding temperature range of 2219 aluminum alloy (438.4–519.2 °C) as a constraint and minimizing the temperature difference across the thickness in the core area as the optimization objective, the following optimized tool geometric parameters were obtained: shoulder diameter of 36 mm, pin cone angle of 6°, shoulder concavity of 2.5°, and thread angle of 11°.

When dealing with large deformations in material simulations, mesh distortion can often be a problem. An alternative method to solve this issue is to use meshless or mesh-free methods. These approaches rely on creating interpolation functions at arbitrary discrete points within the domain rather than relying on elements and a structured mesh. One such technique is the smoothed particle hydrodynamics (SPH) method, which has proven to be an effective simulation tool in continuous media mechanics for modeling the FSW process.

The SPH model is Lagrangian in nature, allowing each material node to be tracked over time. This makes it an attractive option for scientists interested in tracking materials during large deformation and strains. In contrast to conventional grid-based approaches, Lagrangian particle methods such as smoothed particle hydrodynamics can replicate interface dynamics, material strain, significant material deformations, and temperature history without the need for intricate tracking mechanisms.

The SPH approach is a mesh-free numerical technique used to investigate force, temperature, strain, and stress distribution. Instead of using fixed grids, SPH spreads numerical nodes over the problem environment, converting the continuous model into a discretized one. Each node is then moved and accelerated based on effective stress or applied hydrostatic pressure, and the kernel function is used to determine the effect of each node on its neighboring nodes.

Compared to the finite element method, SPH has several advantages, such as the absence of a grid and continuum discretization, making it less prone to mesh distortion issues when simulating large deformations. The particles are used as the basis for an interpolator scheme based on the kernel function, which depends on the smoothing length. In SPH, problem variables such as density, velocity, deformation, and stresses are computed using the kernel interpolation function based on the weighted average value of numerical nodes over their neighbors.

Figure 10 visually demonstrates the similarities between FE and SPH approximations and displays a cluster of components in both of them. Interpolation functions are superimposed on the central element of the FE network of mesh, while the central particle’s kernel is depicted as encompassing its neighboring particles in the SPH network of mesh.

Various types of interpolation kernel smoothing functions are selected according to the physical characteristics of the problem being modeled, particularly for large deformation problems. For 2D and 3D models, the effective radius of the central element is typically around 27 and 56 elements, respectively, when there is a finite number of neighbors. Additionally, a scaling factor (*λ*) can be applied to increase or decrease the number of neighboring elements within the effective radius, which directly impacts the model’s accuracy and the computation time required for calculations.

Figure 11 illustrates the weighting function of *W*, which takes on a bell-shaped form with a kernel length denoted by *λh*. The value of *λ* is determined based on the type of weighting kernel function that is assumed for summing over all particles situated at a distance of *λh* from *r*.

By examining Figure 11, it is clear that the spatial integration process occurs directly at the particle level. Hence, by taking into account the weighted sum of the *i*th node over its neighboring nodes, denoted by the *j*th node, within its practical domain, the value of the field variable for the *i*th node can be determined. The approximate calculation for the discretized field variable, represented as *f*, at the given spatial location of the *i*th material node can be expressed as follows [122]:(19)$$r=|r_{i}^{\alpha }-r_{j}^{\alpha }|$$

The scalar gradient and vector divergence can be expressed as:(20)$$\nabla f_{i}=\sum_{}^{}\frac{M_{j}}{\rho_{j}}f_{j}.\nabla_{i}W(r_{i}-r_{j},h)$$
and
(21)$$\nabla f_{{}_{i}}^{a}=\sum_{j=q}^{N_{i}}\frac{M_{j}}{\rho_{j}}f_{{}_{j}}^{a}.\nabla_{i}W(r_{i}-r_{j},h)$$
*M_j_* and *ρ_j_* denote the element’s mass and density, respectively, while *W(r, h)* depicts an interpolation Kernel function with the given properties. The equations shown above represent the sum of all particles denoted as *j*, applied to the specific particle of interest *i*. It is important to note that the chosen value for *h* (smooth length) and the defined number of *j* particles (resolution) directly impact the characteristics and behavior of the *i* particle. Therefore, the model’s efficiency is enhanced when the local particle density determines the value of *h*. Additionally, in the SPH model, the mass of particle *i* remains constant and is proportional to the product of *ρ_i_* and *h_i_* cubed. Consequently, the value of *h_i_* can be determined through calculation:(22)$$\frac{\partial h_{i}}{\partial t}=-(\frac{h_{i}}{3\rho_{i}})\frac{\partial \rho_{i}}{\partial t}$$

In order to address the SPH model, the material momentum, mass conservation, and energy conservation equations can be reformulated using a system of ordinary differential equations [123]:(23)$$\frac{\partial \rho_{i}}{\partial t}=\rho_{i}\sum_{j=1}^{N_{i}}\frac{M_{j}}{\rho_{j}}V_{ij}.\nabla_{i}W(r_{i}-r_{j},h)$$
(24)$$\frac{\partial V_{i}}{\partial t}=-\sum_{j=1}^{N_{i}}M(\frac{p_{i}+p_{j}}{\rho_{i}\rho_{j}}+\prod_{ij}).\nabla_{i}W(r_{i}-r_{j},h)+F_{i}$$
(25)$$\frac{\partial T_{i}}{\partial t}=-\frac{1}{\rho_{i}C_{p,i}}\left[\sum_{j=1}^{N_{i}}\frac{M_{j}}{\rho_{i}}\frac{4K_{i}K_{j}}{K_{i}+K_{j}}\frac{\left(T_{i}-T_{j}\right)}{{\left|r_{ij}\right|}^{2}}r_{ij}\nabla_{i}W(r_{i}-r_{j},h)-\frac{1}{2}\sum_{j=1}^{N_{i}}M_{j}\prod_{ij}V_{ij}\nabla_{i}W(r_{i}-r_{j},h)-RT_{i}\delta_{i}\right]$$

The symbols $\Pi_{ij}$, *V_ij_*, *F_i_*, *p_i_*, *R*, and *K* represent the viscous term, the velocity of particle *i* at the boundary, force, fluid pressure at position *r_i_*, heat transfer coefficient, and heat conductivity, respectively. When referring to particles located at the boundary, *δ* is assigned a value of one, whereas, for particles outside the boundary, *δ* is set to zero.

As such, researchers have recently turned to the SPH method as a means of modeling the welding process. With its ability to accurately capture the effects of large deformations and temperature changes on material behavior, it has shown great promise in advancing our understanding of FSW and allied processes. Pan et al. [124] proposed an SPH model to investigate the FSW of AZ31 magnesium alloy. Their model provided insights into temperature distribution and history, grain size, texture evolution, and microhardness during the process. Bagheri et al. [125] created a similar model to examine how vibration affects heat generation, temperature history, and mechanical properties during FSW with varying traverse speeds. After conducting the analysis, it was observed that the application of vibration during FSW positively impacted both the material flow and weld area size. Additionally, this technique proved to be effective in minimizing or eliminating any joining defects, ultimately improving the welding quality. In addition, they employed an SPH model for numerical analysis of FSW both in underwater and air conditions [126]. Various mesh density and friction coefficient values were utilized to attain the best possible outcome. The findings indicated that during UWFSW, the maximum temperature was lower in comparison to CFSW at all welding velocities. This can be attributed to the superior convective heat transfer coefficient and specific heat involved.

Shishova et al. [127] introduced a bonding mechanism simulation to accurately represent FSW. This extension of the SPH algorithm tracks the formation of the continuum during the welding process, considering the process parameters. The study found that this extension was valuable in comprehending the physics of FSW, including the transition from contact to continuum, along with other complex phenomena. Hannachi et al. [128] conducted a comparison between the ALE and CEL methods in the numerical simulation of the three-dimensional FSSW process of an AA6082-T6 aluminum alloy. Their findings revealed that the CEL modeling technique was not only more accurate, but also simpler to use and more computationally efficient in comparison to the ALE modeling technique. Hannachi and colleagues [129] examined and compared various friction models for their applicability in modeling the FSSW process of AA6082-T6 aluminum utilizing a CEL formulation. The models they investigated included the conventional Coulomb model, the modified Coulomb model, and the temperature-dependent friction coefficient-based model. After thorough analysis, they determined that the friction model based on the temperature-dependent friction coefficient was the most appropriate for accurately predicting the actual temperature changes during the alloy welding. This particular model takes into account the softening of the material at elevated temperatures, making it the most compatible with real-time temperature evolution during the welding process.

Different commercial codes that are available for modeling FSW utilize various techniques, as previously discussed. Researchers have employed different software packages, such as DEFORM, ABAQUS, ANSYS-Fluent, Forge, etc., to simulate FSW. Table 1 briefly compares different methods of friction stir welding modeling.

## 3. Validation of Numerical Model Using Experimental Date

Validation of a numerical model using experimental data is essential in engineering and scientific fields. It involves comparing the numerical predictions from the model with real-world experimental results to ensure that the model accurately represents the physical behavior of the system being studied. So far, researchers have used different methods to validate their proposed FSW model.

For example, the authors in a previous work [25] compared the temperature history from experimental and simulation data to determine the performance of finite element models. Their results showed that simulation models have an extraordinary ability to predict. As shown in Figure 12, there is a very good agreement between the experimental data and the simulation. These results show that the use of simulation models to estimate the temperature in the welding area is a suitable method and does not have the problems of experimental methods.

Ansari and colleagues [122] conducted a study where they utilized smoothed particle hydrodynamics to create a three-dimensional numerical model for simulating the plunging stage of FSW. To determine the accuracy of their model, they compared the predicted welding force with experimental data and found a satisfactory level of validation.

Talebizadehsardari et al. [130] employed the CFD technique to simulate the UFSW process of Al-Mg aluminum alloys. They aimed to gain a deeper comprehension of the material flow characteristics and the formation of defects in this welding process. Three K-type thermocouples were inserted at various locations to measure the temperature throughout the process. In this research, they validated the numerical results by comparing the temperature recorded by thermocouples and the values obtained by the numerical method (Figure 13).

Trimble et al. [59] conducted an experimental investigation on the tool forces by utilizing a rotating component dynamometer and developing an FE model. They employed DEFORM 3D finite element package to simulate the FSW process on aluminum 2024-T3 plates. To capture the tool forces, a rotating component dynamometer was integrated into the welding setup. This dynamometer enabled the measurement of torque and forces along the X, Y, and Z axes. Piezo-electric crystals were employed to measure the forces, with their signals continuously sampled at a rate of 7.8 kHz per channel. These signals were then processed through the Kistler 9124B multi-channel signal conditioner.

The model accuracy was verified by comparing the forces predicted with the forces obtained through experiments (Figure 14). The researchers concluded that the fluctuation in the predicted forces was primarily caused by the intermittent contact between the tool and the workpiece nodes, as well as the significant amount of remeshing required to maintain the integrity of the mesh. While the predicted forces did not exactly match the experimental results, they exhibited similar behavior. Both the predicted and experimental forces reached peak values during the plunge stage, and all forces approached steady-state values during the translational stage. However, it was noted that the model slightly overestimated the forces, with the torque, vertical, and welding forces being 13.6%, 14.1%, and 18.7% higher than the experimental results, respectively.

Tutunchilar et al. [131], to find the most accurate approximation for the friction factor, conducted an FE model analysis to calculate the axial forces for three different friction factors: 0.4, 0.5, and 0.6. After comparing the simulated axial forces with the experimental measurements, they determined that a friction factor of 0.5 provided the closest match (Figure 15).

Different approaches were utilized to validate material flow obtained from the numerical model. During FSP, Akbari et al. [25] utilized a thermomechanically coupled 3D FE model to numerically simulate the material flow caused by the threaded and circular tool pin profiles. To accurately represent the process, they incorporated particles into a groove with dimensions measuring 3.5 mm in depth and 1.4 mm in width. They validated their material flow results by comparing the SZ shape obtained from the simulation with the SZ shape obtained from the experiment (Figure 16). At the beginning of the simulation, the points can be found along the center line. The figure clearly shows that the simulated SZ shape closely matches the experimental results, indicating a strong agreement between the two.

In order to validate microstructural predictions of FSW using numerical methods, grain size predictions are typically compared to experimental microstructural results. For example, the cellular automaton (CA) method coupled with the Laasraoui Jonas (LJ) model was used to determine the microstructural evolution of AA1100 during FSW [132]. Figure 17 displays the simulated microstructure of the base material of AA1100. In this figure, different colors represent different grains, and grain boundaries are indicated by black lines. It is evident from the figure that the simulated grain structure closely matches the experimental microstructure results, indicating that this model can be effectively utilized for predicting grain size and microstructure in friction stir welds.

## 4. Temperature Distribution

In the FSW process, the temperature in the stir zone increases due to the severe deformation around the tool as well as the frictional heat generation. The temperature distribution in and around this zone directly affects the microstructural properties such as grain size, grain boundaries, formation and dissolution of sediments, precipitates, intermetallics, size, etc. In addition, the temperature distribution determines the residual stress in the weld metal and around it. Therefore, the temperature distribution will have a significant effect on the mechanical properties of the weld. So, it is necessary to study how the temperature is distributed during welding to understand the mechanical and microstructural properties of the joints.

Determining the temperature distribution in the SZ is very challenging due to the severe deformation, so the maximum temperature in this area is often guessed by the structure of the microstructures formed after the welding process or by installing thermocouples around the weld zone on the heat-affected zone (HAZ) or base metal. 

So far, three methods have been used to study the temperature distribution in the stir zone:The first is the experimental temperature measurement using instruments such as thermocouples in the welding area.The second approach is estimating the temperature in the welding area according to the microstructures formed after welding.The third method is to use models or process simulations to calculate the temperature. The use of this method is less challenging than the first and second approaches, and a large part of the research has used this method. The following is a review of each of these methods.

One of the most suitable temperature cycle analysis methods is mathematical equations and finite element simulation. In this method, the temperature can be accurately calculated at any point of the workpiece without problems such as thermocouple placement or calibration. In general, two approaches have been used to calculate the temperature in this area. The first approach is the analytical calculation of temperature using mathematical equations. The second approach simulates the temperature in different areas using the finite element solution.

Heat is created in FSW due to plastic deformation surrounding the rotating tool and friction between the material and the tool. The product of tool speed and frictional force is frictional heat generation. Shear stress and the material velocity adhered to the FSW tool combine to generate heat as a result of plastic deformation around the rotating tool. The frictional heat generation on an element *dA* is given as [133]:(26)$$dQ_{f}=(1-\delta )\omega r\mu pdA$$
where *δ* is the contact state variable, *ω* is the tool angular rotation speed, *r* is the position along tool radius, *µ* is friction coefficient, and *p* is contact pressure. Furthermore, the computation of thermal energy resulting from the plastic deformation can be determined as follows:(27)$$dQ_{p}=\delta wr\tau_{y}dA$$

Therefore, total heat generation can be defined by:(28)$$dQ=dQ_{f}+dQ_{P}=\omega rdA[(\mu p-\delta \mu p)+\delta \tau_{y}]$$

And as a result [133]:(29)$$\tau_{contact}=[(\mu p-\delta \mu p)+\delta \tau_{y}]$$
where *τ_contact_* is contact shear stress. The heat production occurring at the contact interfaces between the tool and workpiece can be broken down into three distinct components, namely: (1) heat generated beneath the tool shoulder (*Q*_1_), (2) heat generated on the pin side of the tool (*Q*_2_), and (3) heat generated at the tip of the tool pin (*Q*_3_).
(30)$$Q_{1}=\frac{2}{3}\pi \tau_{contact}\omega (R_{Shoulder}^{3}-R_{Pin}^{3})$$
(31)$$Q_{2}=2\pi \tau_{contact}\omega R_{probe}^{2}H_{probe}$$
(32)$$Q_{3}=\frac{2}{3}\pi \tau_{contact}\omega R_{Pin}^{3}$$
where *R_shoulder_* is the tool shoulder radius, *R_Pin_* is the tool pin radius, and *H_pin_* is the tool probe height. As a result, the total heat generation would be:(33)$$\begin{array}{l} Q_{total}=Q_{1}+Q_{2}+Q_{3}\\ =\frac{2}{3}\pi \tau_{contact}\omega (R_{Shoulder}^{3})+2\pi \tau_{contact}\omega R_{probe}^{2}H_{probe} \end{array}$$

As can be seen from this calculation, increasing shoulder diameter increases heat generation significantly. Similarly, increasing the diameter of the pin will increase the amount of heat generated. However, because pin diameter has a smaller impact than shoulder diameter, heat generation increases with shoulder diameter growth would be more dominant than pin diameter growth.

As mentioned, measuring the temperature in the stir zone is complicated due to the severe plastic change in the material, and an attempt is made to use simulation to predict the temperature. Simulation models have shown an extraordinary ability to predict temperature. This method estimates the temperature distribution in the weld zone using a finite element solution.

Shojaeefard et al. [102] simulated FSW using the FEM to study the temperature history and distribution in the welding area in 5083 aluminum. The results of this research showed that the temperature distribution around the welding center is almost symmetric. This symmetry can be explained by considering the rotational and traverse speeds (Figure 18). With the rotational speed being significantly greater than the traverse speed, it plays an important role in determining the amount of heat generated, while the traverse speed has minimal effect on the temperature distribution, resulting in a nearly symmetrical temperature profile. For instance, if the rotational speed is approximately 1400 rpm and the pin diameter is 3 mm, then the linear speed of the pin owing to the rotational speed would be around 13,194 mm/min, which is much higher than the tool traverse speed that usually ranges from 50 to 120 mm/min.

According to Chao et al. [105], the majority of heat produced by friction—approximately 95%—is transferred to the workpiece, while only a small amount of 5% is transferred to the tool. Additionally, roughly 80% of the plastic work rate is dissipated as heat. Larsen et al. [134] utilized inverse modeling techniques to determine the heat transfer coefficient between the backing plate and material, aiming to improve temperature calculations in friction stir welding by reducing the discrepancy between experimental and 3D FE model temperatures. The study revealed that the heat transfer coefficient was not uniform, with its highest value located beneath the welding tool. To address this issue, the researchers employed an optimization based on the gradient method and a non-uniform parameterization of the coefficient of heat transfer in their method of inverse modeling. This innovative technique allowed them to obtain a more precise determination of the heat transfer coefficient, making it the first instance of this methodology being employed in FSW.

Prasanna et al. [135] developed a finite element (FE) simulation model with enhanced predictive capabilities for temperature history in stainless steel. To validate the model, they tested it against existing experimental data on 304 L stainless steel and found that the maximum temperature obtained was 1057 °C, significantly lower than the steel’s melting point of 1450 °C. A paper by Xu et al. presented the findings of a numerical and experimental observation of heat dissipation and generation during FSW. The study aimed to improve the selection of process parameters and the design of welding tools for manufacturing combat vehicle structures [136].

A 3D FE-coupled thermal stress model of the friction stir spot weld (FSSW) has been established using Abaqus/Explicit code [137]. For elastic–plastic work deformations, a rate-dependent Johnson–Cook material model was utilized. The model’s energy dissipation and temperature distribution have been analyzed in detail. 

Utilizing the STAR-CCM+ software, a commercial finite volume method (FVM) code, thermomechanical simulation was conducted for sheets of AA5083-H18 in the FSW process [138]. The simulation involved calculating strain rate and temperature distributions under steady-state conditions. Afterward, the simulated temperatures, including peak values and profiles, were compared to the experimental findings. Notably, accurate simulation outcomes were achieved by incorporating an appropriate thermal boundary condition for the backing plate (anvil). Using the thermal elastic–plastic FE model, a simulation was performed on the transient temperature distribution in 2024-T4 Al FSW joints [139]. It was observed that the temperature value decreases gradually towards the weld periphery in a radiating pattern around the pin, and the peak temperature in the weld can be reached up to around 400 °C.

Rajesh et al. [140] studied the FSW of A16061 plates and its associated temperature distribution, utilizing an asymmetrical analytical model that considers the heat input resulting from the combined rotational and traverse motion of the tool shoulder and pin [141]. To evaluate the heat distribution at the FSW joint, a 3D FE heat transfer analysis program was employed, resulting in a plotted representation of the heat distribution in the A1 6061 plate. This study found that the heat distribution from the FE analysis agreed with the values obtained through experimental measurements. This work showcases the potential of analytical and computational tools for understanding the fundamental aspects of FSW and optimizing the welding process for enhanced performance. The welding parameters for FSW were comprehensively examined, including the pin angular velocity, pin and shoulder radius, cone angle, and screw thread angle [142]. The results demonstrated that increasing the radius of the pin and angular velocity while staying within the FSW limits led to a higher maximum temperature. Conversely, an increase in the cone angle and screw thread angle resulted in a decrease in the maximum temperature. These findings highlight the critical role of these parameters in determining the temperature distribution during FSW and can provide valuable insights for optimizing the process to enhance its efficiency and quality.

As the largest source of frictional heat generation, the tool shoulder size plays the most critical role in the heat generated. The authors used numerical methods to examine the amount of heat generated by various tools (Figure 19). As it turns out, with increasing shoulder diameter, the temperature rises dramatically.

It was also shown that the HAZ area, the main area for fragment failure, broadens with a larger shoulder diameter. The HAZ zone is selected according to the definition of the zone where the temperature is above 400 degrees (Figure 20). As previously stated, a greater maximum temperature in the FSW joint promotes solid diffusion, plasticized mixing, and improved weld quality regarding material flow. On the other hand, because the HAZ is one of the most competent zones for fracture, it causes the weld quality to deteriorate. As a result, the best rotational speed is needed to maintain a high peak temperature in the weld zone while minimizing HAZ expansion.

The tool pin diameter is another critical parameter in determining the temperature of the workpiece. As shown in Figure 20, as the diameter of the tool pin increases, the temperature of the stir zone increases. A rise in temperature occurs due to an increase in pin diameter for two reasons. On the one hand, due to the increase in the contact surface, more frictional heat is produced, and on the other hand, due to more plastic deformation, the heat resulting from the material deformation increases. As shown in this figure, this increase in temperature is minimal compared to the effect of the shoulder diameter, which indicates that the tool shoulder plays a major role in determining temperature rather than the pin.

The tool pin shape is another variable that affects the temperature of the stir zone. Different pins can affect the temperature of the workpiece due to different strain rates as well as different contact levels with the material. Akbari et al. [35] studied the effect of FSP tools with different pin shapes, consisting of hexagonal, triflate, triangular, square, and cylindrical, on temperature using the CEL approach (Figure 21). 

After analyzing the data, the researchers determined that there were minor temperature variations among the samples. This was attributed to the tool shoulder being the primary heat source during the process. Since all samples utilized tools with the same shoulder size, the temperature difference between them was negligible. However, they observed that the sample created using a cylindrical tool had elevated temperatures in the region affected by the pin due to the larger surface area of the pin. Conversely, the sample made with a triangular-shaped tool experienced lower temperatures in this particular area due to its smaller pin surface area (Figure 22).

Akbari et al. [108] investigated the role of frictional heat and plastic deformation in heat production. Upon examining two models, one for calculating heat resulting from plastic deformation without accounting for friction and the other for determining the amount of frictional heat without factoring in the conversion of plastic deformation to heat, it was observed that the foremost reason for the rising temperature of the material is the heat generated by friction between the material and the tool (as depicted in Figure 23). Further analysis revealed that the increase in temperature due to frictional heat amounts to approximately 270 °C, which is over three times more than the temperature increase resulting from plastic deformation. This examination clearly shows that the process of generating frictional heat is the primary mechanism responsible for elevating the material temperature, as opposed to plastic deformation.

In another study, Mohan et al. [46] investigated the effect of frictional heat generation and plastic deformation on the generated heat using the CFD method. They discovered that as tool rotational speed increases, friction-generated heat reduces while plastic deformation increases (Figure 24). Plastic deformation accounts for 48.84 percent of the total heat output at 18,000 rpm but increases to 58.32 percent and 66.21 percent at 21,000 and 24,000 rpm, respectively.

As mentioned earlier, traverse and rotational speeds play an essential role in the mechanical and microstructural properties of the welds. In previous works, the authors investigated the impact of these parameters on temperature distribution. The results of this study showed that as the rotational speed increases, as shown in Figure 25, the workpiece temperature increases. The rise in temperature results from increased rotational speed, generating additional heat through friction. Furthermore, by increasing the traverse speed, the temperature within the workpiece is diminished. As the traverse speed rises, the welding process becomes expedited, allowing the tool-less opportunity to heat the component.

Higher tool rotational speed also results in a HAZ expansion and a higher maximum temperature value. Opposite results can be obtained by traverse speed variation. Figure 25 shows the HAZ width and temperature profiles in an FSW process by varying traverse speed in a fixed rotational velocity of 700 rpm. The higher the traverse speed, the narrower the HAZ, and the lower the peak temperature.

Cooling is an essential aspect of FSW, which can significantly affect the quality and properties of welded joints. In FSW, the metal around the rotating tool becomes plasticized and heated, and then it cools and solidifies behind the tool. The cooling rate in FSW can be affected by various factors, such as welding parameters, material properties, tool design, and cooling medium. The cooling rate can influence the microstructure and mechanical properties of the weld, including its hardness, strength, ductility, and toughness.

One of the main benefits of FSW is that it is a low-heat input process, which minimizes thermal distortion and reduces the need for post-weld heat treatment. However, excessive cooling rates or inadequate cooling can lead to defects in the weld, such as porosity, cracking, or incomplete fusion. Various cooling methods can be used to control the cooling rate in FSW, such as air cooling, water cooling, or active cooling, using a cooling channel built into the tool. These methods can help optimize the weld microstructure and mechanical properties, increase its fatigue life, and reduce the risk of defects.

Cooling during friction stir welding in some alloys reduces grain growth and thus strengthens the mechanical properties and improves the welding quality. This is especially true for aluminum alloys. So far, various cooling methods have been performed, and their effect on welding specimens has been investigated. Modeling of cooling conditions during the process is generally performed by determining a convection heat transfer coefficient according to cooling conditions. So far, several studies have investigated cooling methods using numerical models. The authors investigated different cooling methods and their effects on the sample’s temperature distribution [143]. The results indicate that in the absence of cooling, the maximum temperature observed among all conditions was 493 °C. However, it is worth noting that the maximum temperature observed for all samples remained below the melting point of the base material, indicating that a solid-state process had taken place. Figure 26 clearly illustrates that the use of coolants during welding leads to a reduction in weld temperature. Specifically, using air and water coolant decreased maximum temperature by 22.8% and 36.3%, respectively, compared to non-cooled samples. Moreover, the peak temperature of water-cooled FSP samples was lower than that of air-cooled samples, likely due to the higher convective coefficient and specific heat of water relative to forced air, resulting in a higher cooling rate. Therefore, it can be inferred from these findings that water possesses the highest cooling capacity, while the non-cooled samples exhibit the lowest cooling capability.

Figure 27 illustrates the temperature distribution of the FSP tool and the surface of the top sample at different stages of the process. The figure showcases that using coolants has led to an increase in temperature gradient at the stir zone while simultaneously narrowing down the elevated temperature distribution zone. As a result, heat is concentrated closer to the FSP tool in the cooling assisted samples, causing restriction of the HAZ. This restriction effect boosts the mechanical properties of the FSP samples.

## 5. Strain Distribution

Strain is a physical quantity that measures the amount of deformation or change in the length of a material per unit length. In the FSW process, strain refers to the localized deformation or distortion that occurs in the metal during the welding process. Over the FSW, the revolving tool generates heat and pressure, which causes the metal to become plasticized and flow around the tool. As the tool moves along the joint line, the metal is subjected to different degrees of strain depending on the local conditions. For example, the metal near the front of the tool experiences high levels of strain due to the intense deformation caused by the pin and shoulder.

The distribution pattern of strain in FSW can be affected by various factors, such as process parameters, tool design, material properties, and joint geometry. Excessive strain can lead to defects in the welded joint, such as cracks, voids, or residual stresses, compromising its strength and durability. Therefore, understanding and controlling the strain distribution within acceptable limits is an essential aspect of FSW process optimization and quality assurance. Researchers use numerical simulations, experimental tests, and non-destructive evaluation techniques to investigate the strain and other mechanical properties of welded joints made by FSW.

Maximum strains are measured on the AS where the tool traverse speed and peripheral velocity are positively combined. This suggests that the rotational and traverse speeds have a significant effect on deformation and material flow. The effective strain distribution and contours in the weld zone are illustrated in Figure 28 and Figure 29. It is evident from both figures that the AS experiences higher effective strain than the RS.

Figure 30a showcases the strain distribution in AZ91 friction stir processing, further supporting the observation that the AS experiences higher effective strain. On the other hand, Figure 30b illustrates how effective strain is distributed in the cross-section of an FSP sample behind the tool pin. The figure shows that effective strain in the cross-section increases from the bottom to the top surface. This increase is attributed to the greater plastic deformation near the top surface as a result of the tool shoulder, which accelerates material flow near the top surface. Consequently, this phenomenon causes the shape of SZ to become conical [109]. Furthermore, larger effective strain values on the AS lead the SZ to expand and incline toward the AS (Figure 30c).

Figure 31 displays the strain profiles in the joint cross-section for various shoulder and pin diameters. The figure shows that the strain profiles around the weld line exhibit asymmetry. It can also be inferred that an increase in the pin diameter results in a rise in maximum strain from 30 mm/mm to 45 mm/mm. On the other hand, a comparison between Figure 31a,b indicates that shoulder diameter has a lesser effect on maximum strain than the pin diameter. However, increased shoulder diameter expands the plastically deformed area due to additional material softening and heat generation [102].

A study by Meena et al. [145] utilized finite element analysis to simulate the FSW of polycarbonate. The researchers observed that the highest strain rate occurs in the SZ but sharply decreases at the TMAZ. A low strain rate regime in the SZ indicates poor material flow due to low plasticization, which may lead to discontinuity formation at the border of the SZ and HAZ. When high tool traveling speed and low heat input are used, fluctuations in strain rate can form tiny planar cracks around the SZ. The simulation results indicate that a growth in the heat input raises the peak strain rate in the SZ, as depicted in Figure 32.

Similarly, accelerating the tool rotation speed in the constant traverse speed also intensifies the strain rate. On the other hand, increasing the traverse speed in the constant rotational speed lowers the strain rate. The researchers noted that a typical low heat input would reduce the strain rate, and consequently, the susceptibility of crack generation and propagation around the joint line will increase. Additionally, an increase in tool rotational speed causes the expansion of SZ.

In a study by Memon et al. [142], underwater friction stir welding (UFSW) was simulated for welding on low-carbon steel. The simulation’s findings demonstrated that the steel strain rate in the SZ was comparatively higher in the FSW joint than in the UFSW joint. An investigation of the impact of process parameter variations on the mechanical properties during FSW processes was carried out by creating a 3D FE model. To ensure accuracy, the numerical results were compared with experimental results. The findings revealed that there is a strong correlation between microstructural evolution and plastic strain.

The strain distribution during FSW with various tools has been analyzed, and the results are depicted in Figure 33 [108]. The study compares the strain in joints produced using FSW/FSP and shoulderless and pinless tools. The strain in the joint fabricated using the FSW tool was observed to be significantly higher than those produced using the shoulderless and pinless tools. This effect was attributed to the combined influence of the tool pin and the shoulder.

Additionally, the resulting strain was found to be localized in two distinct zones: one affected by the shoulder and the other by the tool pin. The shoulder-affected zone was located near the top surface of the joint and experienced a significant impact from the tool shoulder during the process. On the other hand, the pin-affected zone was dominated by the cylindrical pin and experienced a greater material strain.

It is worth noting that the maximum strain occurred at the intersection of these two zones, where the tool pin and shoulder effects overlapped. In contrast, in the absence of a pin, the strain was only observed in the shoulder-affected area, which was much lower than that observed for the FSW/FSP tool. Similarly, in the case of a shoulderless tool, only the area affected by the pin showed minimal strain.

These observations highlight the importance of the combination of the shoulder and pin effects in inducing material flow and strain in both zones under the influence of the tool. The shoulder component raises the temperature and facilitates material softening, whereas the pin component induces strain and material flow. In the absence of the tool shoulder, there is inadequate heat generation to soften the material, resulting in minimal strain. Similarly, without the tool pin, the flow of material is restricted to the plate’s upper surface owing to the tool shoulder rotation (Figure 33).

## 6. The Role of the Residual Stress

Residual stress is a common phenomenon that occurs in many welding processes, including friction stir welding. In the FSW process, residual stresses are generated due to the thermal and mechanical loadings. During FSW, the material experiences high temperatures and extreme pressures, resulting in plastic deformation and recrystallization. This, in turn, causes residual stresses to develop within the weld region.

There are two primary sources of residual stress in FSW: thermal and mechanical. Thermal stresses arise from the temperature gradients that occur during the welding process. The outer regions of the workpiece are heated and cooled more rapidly than the inner regions, leading to differential thermal expansion and contraction. This can cause tensile or compressive stresses to form in the weld. On the other hand, mechanical stresses arise from the tool movement over the material as it undergoes plastic deformation. These stresses are caused by the interaction between the tool and the material, with the tool exerting pressure and forces on the material in different directions.

The presence of residual stresses can have both positive and negative effects on the properties of the welded joint. On the positive side, residual stresses can improve the fatigue resistance of the material by reducing the likelihood of crack initiation. They can also contribute to increased strength and stiffness in the joint. However, the negative effects of residual stresses can be significant. Tensile residual stresses can lead to cracking and deformation, while compressive stresses can cause buckling and distortion. In addition, residual stresses can affect dimensional stability and contribute to corrosion and stress corrosion cracking.

To mitigate the negative effects of residual stresses, various techniques can be employed to reduce their magnitude. These include post-weld heat treatment, shot peening, and stress relief annealing. By managing residual stresses in FSW, it is possible to produce high-quality welds with improved properties and reliability.

To ensure optimal joint quality and minimize residual stress in FSW, it is important to predict the clamping force applied to the plates accurately. This is especially crucial for robotic FSW, where understanding the mechanics of the process and developing effective regulation models require precise force history data. A 3D FE model can be used to analyze the thermal history and stress evolution during FSW, enabling the computation of mechanical forces along the longitudinal, vertical, and lateral axes. By studying the relationship between factors such as tool rotational and traverse speeds, fixture release, and calculated residual stress, we can gain valuable insights into optimizing the welding process.

Due to the simultaneous application of heat and accompanying constraints in fusion welding, the workpiece develops a complicated combination of thermal and mechanical strains. Residual stresses, referred to as residual stresses in fusion welding, typically reach the yield strength of the base material. However, these stresses are relatively insignificant due to the low-temperature solid-state nature of FSW. Rigid and robust fixtures of FSW, on the other hand, place a far greater restraint on the joint plates, resulting in a higher residual stress value. These constraint stresses are caused by the weld’s inability to contract.

Many researchers have verified their proposed numerical model based on the other parameters, such as temperature history in the welding workpiece or the force history applied to the tool, and then reported the achieved residual stress via the simulation model. However, others verified the residual stress results achieved through the simulation models directly by the experiments. Lemos et al. [146] employed the XRD technique to measure the residual stress in two transverse and longitudinal directions. In this method, they applied XRD experiments with the starting point at the base material on the RS of the joint and then passing through the weld and finishing at the opposite side on the base material. In this method, they used the conventional sin^2^ψ method with Bragg–Brentano geometry. Generally, in terms of experimental residual stress measurements, the application of non-destructive methods like X-ray diffraction and neutron diffraction are available, along with destructive methods like cut compliance and the contour method. Some researchers believe that measuring the residual stress in the FSW samples normally includes cutting a test piece of the welded structure; this will lead to a potential relaxation of residual stresses [147]. For this reason, Threadgill et al. [148] claimed that a test sample at an approximate length of eight times the diameter of the tool is essential. To review more details in this regard, see ref. [149]. 

Khandkar et al. [143] utilized a sequentially coupled thermomechanical FEM to investigate the residual stresses developed during FSW. The initial step in their model involves conducting thermal analysis to generate temperature profiles during the welding process. Subsequently, these outcomes serve as thermal input for the mechanical analysis, enabling the prediction of strains and residual stresses resulting from temperature fluctuations within confined metal plates. Some considerations are developed in works that numerically study the residual stresses in FSW joints. First and foremost, the tool’s local mechanical action, particularly that of the tool pin, is ignored. Only the thermal flux is considered to study the process and macro-effect on the material. Furthermore, the thermal models used to describe the heat flux caused by the tool action are always axisymmetric; therefore, the effect of asymmetric material flow in FSW is ignored. 

Most of these limits were overcome by the model presented by Fratini et al. [62]. Using the FEM model, they calculated a real thermal flow that occurs during an FSW operation. Such temperature histories are supplied to a separate elastoplastic FEM model at each node of the model. As a result, the residual stress status for the butt joint under investigation is determined. In this method, the final stress condition in the material after the clamping constraints were released was highlighted in terms of longitudinal stress, i.e., the stress in the transverse stress, welding direction, and normal stress. The contours of transverse and longitudinal residual stresses are depicted in Figure 34a,b. Both transverse and longitudinal residual stresses are distributed asymmetrically, as expected, given the asymmetric nature of the material flow. The asymmetric pattern of stresses is also visible in Figure 34c, which depicts normal residual stresses. The longitudinal residual stresses are positive with the external zones under compressive stress, whereas the transverse residual stresses are responsible for the welded plate in terms of the out-of-plane distortion. It can be seen in Figure 34c that the through-thickness variation of the residual stress can be ignored.

By examining the residual stress patterns in the transverse section of a joint, valuable insight can be gained. The central region of the transverse section, as depicted in Figure 35, displays the mean residual stress. Although there is an absence of normal stresses in this diagram, longitudinal stresses are slightly greater than their transverse counterparts. Notably, the AS exhibits higher longitudinal stresses compared to the RS, particularly near the tool shoulder edge [150].

The investigation examined the impact of cooling on the stress formation during FSW [151]. The primary objective was to determine whether the use of liquid CO_2_ cooling systems, which are practical in several settings, could effectively decrease residual stress. The analysis indicated that a considerable reduction in residual stress could be attained, especially along the weld line, which is dependent on the location, power, and size of the cooling sinks. Techniques employed near the boundary between the HAZ and TMAZ were observed to be more adept at lowering the stress at the weld centerline compared to those applied under the tool shoulder periphery.

The application of the laser ultrasonic method was examined to identify and quantify defects, along with measuring residual stress levels in the welds created through FSW [152]. Combining the Fourier domain synthetic aperture focusing technique with laser ultrasonic testing yields highly accurate outcomes in detecting a lack of penetration in butt joints. The detection threshold is consistent with reduced mechanical qualities. Moreover, ultrasonic frequencies up to 200 MHz enable the identification of kissing bonds in lap joints. The laser ultrasonic technique can also be employed to explore residual stresses produced by FSW. The residual stress patterns measured across the weld line correspond well with findings from a finite element model and strain gauge measurements.

A framework for analysis was introduced to forecast the residual stresses that arise from laser shock peening of an FSW 2195 aluminum alloy specimen. This was achieved through the utilization of LS-DYNA, a finite element software [153]. An analytical framework was utilized to determine the residual stresses caused by laser shock peening. The approach involved utilizing the generated pressures as forces in an explicit transient analysis, followed by conducting an implicit spring-back analysis to identify the final residual stresses. To verify the accuracy of the methodology, a comparison was made between the residual stresses obtained from the analysis and those from a test specimen of laser-peened base material that was not subjected to friction stir welding. To simplify the study, discrete, separate materials were defined for variable material qualities and regions. The peening transient analysis integrated the welding residual stresses along with the peening residual stresses, and the blind hole method was employed to examine the residual stresses of FSW that had undergone various welding parameters [153]. The energy parameter obtained from the FE analysis results was used to correct the strain-releasing coefficients A and B. The findings revealed that longitudinal residual stresses were asymmetrically distributed at different sides of the weld center, with substantial longitudinal residual stresses on the AS and comparatively low longitudinal residual stresses on the RS. The transverse residual stresses amounted to around 12.7% of the base metal strength. 

A study aimed to investigate how the type of welding parameters and alloy affect the distribution and magnitude of residual stresses in FSW [154]. It was found that the strains produced were significantly lower than the yield strength of the materials at room temperature. On age hardenable structural alloys, an “M-shaped” distribution of residual stress was consistently observed, with a more pronounced effect in the 6082-T6 alloy compared to the 2024-T3 alloy. However, a “plateau” distribution was observed in 5754 H111 alloy. The differences in microstructural variations in the weld center were identified as the primary cause of the low magnitudes and discrepancies in longitudinal residual stress distribution. To explore the impact of tool traverse speed on residual stress and heat distribution, a 3D numerical simulation of FSW was conducted [155]. The proposed model considered the fact that material properties are influenced by temperature. The resulting residual stresses revealed that heat distribution is non-uniform across the material thickness, indicating variations in its distribution. However, the model only considered the impact of heat when predicting residual stress. As a result, minor differences were observed between the simulation results and experimental data. This was primarily attributed to the fact that other factors beyond heat, such as microstructural variations, may also contribute to the residual stress distribution in the actual material.

Ultrasonic measurements are a useful tool for estimating through-thickness residual stresses in aluminum plates during FSW. This method relies on the acoustic-elasticity law, which describes the relationship between material stress and acoustic waves. Specifically, longitudinal critically refracted (LCR) waves are utilized to evaluate ultrasonic stress, propagating parallel to the surface within an effective depth. The through-thickness longitudinal residual stresses in FSW 5086 aluminum plates were analyzed by examining the dispersion of LCR waves generated using ultrasonic transducers operating at various frequencies (1, 2, 4, and 5 MHz). The FELCR method, which combines FE analysis and the LCR method, was employed to achieve the desired through-thickness longitudinal residual stress distribution [156]. The findings of the FE simulation and the ultrasonic measurement were found to be in good agreement. As a result, the FELCR approach may be used for the FSW plates with success. 

A research team consisting of Kaid and colleagues [157] has devised a structural-thermal model that is both non-linear and transient, capable of simulating residual stress during friction stir welding of aluminum alloy in three dimensions (3D). Their findings indicated that the residual stress increases as the welding speed increases.

A recent investigation utilized a sequential simulation process to numerically evaluate residual stresses [83]. In the beginning, the temperature information was acquired from the CEL model and subsequently utilized in the mechanical model to examine the distribution of residual stress across the entire volume of the dissimilar workpiece. The longitudinal stress component exhibited a greater magnitude compared to the transverse component. It changed from tensile stress within the central welding region to compressive stress beyond the shoulder diameter before transitioning back to tensile stress near the outer boundaries of the alloys. Alternatively, the transverse residual stress had a kidney-shaped profile with a compressive force below the tool probe and shoulder zone before flipping to tension with limited value beyond the shoulder diameter. The results indicated that the RS exhibited higher residual stresses than the AS (Figure 36).

The prediction of residual stresses in the joining process of duplex stainless steel (DSS) to a Cu alloy through FSW was conducted by Shokri et al. [158] using thermomechanical simulation based on the CEL approach. The study revealed that DSS exhibited higher residual stresses than Cu alloys. Moreover, an increase in rotation speed resulted in a decrease in longitudinal residual stress, whereas an increase in travel speed led to an increase in longitudinal residual stress. However, both speed parameters had a negligible impact on transversal residual stress.

Geng and colleagues [159] conducted a numerical analysis of the thermomechanical conditions during friction stir lap welding of dissimilar Al/steel materials. Using a three-dimensional Eulerian-based finite element model, they investigated the distribution of residual stresses. Their findings showed that increasing the rotational speed led to a higher cooling rate and welding temperature, increasing residual stress levels in the weld zone. In addition, they observed different shapes of residual stresses near the interface—an M-shape in the Al alloy and a W-shape in the steel—due to the thermomechanical processing conditions and the mismatch of thermal expansion coefficients. Furthermore, increasing the rotational speed led to a decrease in near interfacial compressive stress on the steel side.

To control and eliminate the residual stress and distortion of the FSW joint, He et al. [160] employed an external stationary shoulder. The impact of rotational speeds on temperature and stress distribution during welding was analyzed through the utilization of a thermomechanical model. They observed that the region where SZ and HAZ met had a significant amount of high local tensile residual stress. The cause of this stress was attributed to the mechanical force and high temperature generated by the rotational shoulder. In contrast, when utilizing stationary shoulder friction stir welding, the weld experienced an additional tensile strain from the stationary shoulder. This beneficially counteracted the compressive plastic strain and allowed for better residual stress control. Moreover, the peak tensile residual stress generated during stationary shoulder friction stir welding was about 45.9% lower than that produced by conventional FSW.

A numerical model that linked the finite element method was deployed to examine the residual stresses resulting from thermal cycling due to FSW [161]. The anticipated longitudinal stresses displayed a “W” profile and reached their maximum at 300 MPa, with tensile stress peaks within the weld and compressive stresses beyond it. The utilization of high rotational speed and low welding velocity, known as “hot” welding conditions, notably amplified residual stresses, primarily in the transverse direction. Conversely, “cold” welding conditions produced lower residual stresses. Neutron diffraction validated the size and residual stress distribution computed through the finite element model.

Yan et al. [162] conducted numerical simulations to analyze the impact of mechanical loads on the FSW process. Their findings indicate that the downforce effectively minimizes residual stress, while torque causes an uneven residual stress distribution. Furthermore, the mechanical loads entirely alter the residual distortion pattern from a saddle state to an anti-saddle state. A numerical model was created by Jin and Sandström [163] to analyze distributions of residual stress following the FSW of copper canisters. The model coupled elastoplasticity and heat transfer analyses. The simulation findings reveal that the residual stress pattern within the material is influenced by the circumferential angle and exhibits asymmetry around the weld line. The weld line and its vicinity exhibit both compressive and tensile stresses. In general, the maximum tensile stress, as determined through FE analysis or assessed via hole drilling or X-ray diffraction methods, remains below 50 MPa.

The impact of cooling on the formation of stresses in FSW was studied to examine whether liquid CO2 cooling systems, applied at feasible locations, could produce substantial reductions in residual stress [151]. The simulations indicated that significant reductions in residual stress could be achieved by varying the size, power, and location of the cooling sinks, particularly at the weld line. The approaches used were more effective at reducing the welding centerline stress rather than those at the HAZ/TMAZ boundary.

Bastier et al. [164] conducted a simulation of FSW consisting of two main stages. The first stage involved using a 3D-mixed FE model to determine the flow of material and temperature field during FSW. The second stage utilized an elastoviscoplastic constitutive law to estimate the residual state resulting from the process. The longitudinal residual stress field displayed a bimodal profile with two peaks in the area with a high gradient of dissolved precipitates. The study observed that lower rotational speeds and higher traverse speeds resulted in lower residual distortions and temperatures.

To analyze the effects of tool traverse speed on residual stress and heat distribution, a three-dimensional numerical simulation was conducted for FSW [155]. The developed model incorporated temperature-dependent parameters to account for material characteristics. The analysis of residual stresses revealed an asymmetrical and varying heat distribution throughout the thickness. Specifically focusing on the impact of heat, the study primarily examined its contribution to the resulting residual stress. It was acknowledged as the primary factor introducing slight disparities between the simulation and experimental outcomes.

Furthermore, the investigation also explored the effects of alloy type and welding parameters on the magnitude and distribution of residual stresses across the FSW process [154]. The stresses exerted are significantly lower than the yield strength of the base material at room temperature. Age hardenable structural alloys, particularly 6082-T6 alloy compared to 2024-T3 alloy, consistently exhibit an “M-shaped” distribution pattern. Conversely, a “plateau” distribution is observed in the 5754 H111 alloy. The relatively low magnitude and variations in the longitudinal residual stress distribution can primarily be attributed to microstructural modifications occurring at the SZ center. 

Eivani et al. [165] created a model that combined SPH with ANFIS to assess the residual stress in the FSW of AZ91 magnesium alloy. They developed a complimentary ANFIS and SPH approach, where the ANFIS was trained effectively and subsequently utilized to predict the impact of FSW process parameters. This research aimed to control the impact and outcome of the FSW process parameter on the residual stress of the welded specimens. The findings indicate that the temperature has the most significant effect, while the strain rate and strain have a relatively lower influence on the high-temperature mechanical properties of the specimens. The residual stress evaluation in the study by Zina et al. [166] was conducted using the thermomechanical model specifically designed for FSW with aluminum alloy A6061-T6. The residual stresses observed were influenced by the FSW processes, particularly the welding mixing, and temperature, both of which were interconnected with the welding parameters. Notably, an increase in welding speed was observed to result in higher residual stress levels. However, it is important to highlight that the residual stresses predicted by the FE model never surpassed 54% of the elastic limit.

Ranjole et al. [167] constructed a 3D FE model to conduct numerical simulations that predict the residual stress resulting from the dissimilar joining of AZ31 and AA5083 alloys using FSW. As the rotational speed increases, it is shown that the stresses at the SZ also increase. The peak longitudinal stresses are found to increase as the rotational speed increases as a result of the added heat input from the friction between the tool and the material. On the other hand, when the traverse speed varies, there is minimal variation at the SZ but high variation as one moves away from the weld zone among the process parameters. The peak longitudinal stresses are observed to be higher for lower traverse speeds due to the shorter tool interaction time, resulting in a greater temperature difference and longer heat accumulation time within the materials. From the simulation results, it can be concluded that the rotational speed has a greater impact on residual stresses at the weld zone, whereas the traverse speed minimally affects the increase in residual stress at the weld zone.

## 7. Forces and Torque during FSW Processing

The force and torque applied to the tool during FSW are the determining parameters in the life of the welding tool. Due to the rotational and traverse speed during the process, forces are applied to the tool that causes erosion and thus reduce its life and increase the costs of the process. In addition, erosion of the tool may contaminate the joint and thus reduce the joint properties. Also, the torque on the tool is directly related to the power required for the welding process, and more power is required as the torque increases. For this purpose, measuring and controlling the force during the process is necessary to optimize the tool design, increase the life of the tool, design the retainers, increase the quality of the weld, and thus increase the strength of the weld. Generally, during the FSW, three axial, lateral, and longitudinal forces are applied to the tool (Figure 37). The vertical force acting on the tool acts to lift the tool. Because of this need to maintain the position of the tool during welding, downward vertical force is essential. The linear passage of the FSW tool through the material produces longitudinal forces on the tool. The longitudinal acting force is positive in the traverse direction and parallel to the tool motion. The formation of this force occurs due to the resistance of the material to the tool’s movement. It is logical to infer that as the temperature of the material surrounding the tool increases, this force will diminish. Due to the Magnus effect, the combination of tool traverse and rotational speed results in an asymmetric flow field surrounding the FSW tool, as well as a lateral force on the FSW tool that is perpendicular to the linear motion direction [168]. Over the FSW, the tool position is controlled in two ways by: 

The vertical position of the tool;The force applied to the tool, which causes a change in the position of the tool.

Welding forces have been measured so far in two ways: numerically and experimentally. In experimental models, the forces applied to the tool during the process are measured by designing a special dynamometer. In numerical methods, the forces acting on the tool are generally estimated using a finite element simulation model.

Some articles discuss reliable numerical modeling of the forces generated throughout the process. Ulysse [169] presented a 3D FEM viscoplastic model for FSW of thick aluminum plates using FIDAP (fluid dynamics analysis package). The author utilized the model to forecast the forces operating on the tool pin after attaining a fair agreement between the expected and measured temperatures. 

Colegrove and Shercliff [39] employed commercial CFD software to numerically investigate the effects of pin geometry on FSW, using both 2D and 3D models. Despite this approach, they found that the software’s ability to predict welding forces was inadequate. Buffa et al. [58,103] are a few of the authors reporting successful modeling of the FSW process using DEFORM-3D. They only modeled a cylindrical, smooth tool pin; however, they did validate the model by comparing experimental and anticipated temperatures. Shojaeefard et al. [102] reported successful modeling of the forces generated during FSW using DEFORM-3D software.

Figure 38 depicts the tool torque and forces that were documented by Trimble et al. [59] during their testing. During the initial plunge, when the workpiece makes contact first with the pin and then with the shoulder, the forces exerted on the FSW tool experience significant fluctuations because of the interplay between softening (as a consequence of heat generation) and work hardening (under axial shear and compression) [168,170]. During the initial plunge of the FSW tool into the material, when the shoulder reaches the desired depth, both the axial force and torque reach their maximum values. Subsequently, as the tool moves through the material, the forces on the FSW tool stabilize at a steady state, which is typically lower than the maximum forces experienced during plunging. However, tool failure is still risky during the plunge stage.

Numerical analysis along with experimental observation of the FSW process using a threaded pin was carried out by Atharifar et al. [168]. In the context of friction stir welding of AA6061, analysis was conducted on the various forces acting on both the pin and shoulder components. Specifically, the axial, longitudinal, and lateral forces were computed, revealing that increasing rotational speed led to a corresponding increase in axial forces while decreasing tool traverse speed resulted in the same effect. This finding was corroborated by experimental and FEA data. Interestingly, the calculated longitudinal stresses on both the tool pin and shoulder decreased as the rotational speed was raised and the tool traverse speed was lowered. This can be attributed to the fact that higher rates of heat generation caused by increased rotational speed result in lower flow stress, leading to reduced longitudinal force. Dynamic pressure distribution changes along the welding line were also identified as a factor affecting longitudinal force, with traverse speed having less impact than rotational speed on both axial and lateral forces. Moreover, it was found that the axial, lateral, and longitudinal forces experienced by the tool shoulder were considerably greater than those encountered by the tool pin. Finally, high moments were observed at low rotational and high traverse speeds. Table 2 presents a summary of how the forces acting on the FSW tool are affected by travel and rotational speeds [171]. It is important to note that there are signs of either +/− in each cell, indicating if an increase in the column parameter results in an increase or decrease in the corresponding row parameter. Additionally, the symbol ~ indicates weak or no effect.

Atharifar et al. [168] investigated the variations of forces and moments that acted on the FSW tool by changing the traverse and rotational speed. They found the longitudinal force diminishes as the rotational speed increases. Raising the traverse speed causes a rise in the longitudinal force. They stated that the maximum magnitude of this force is reached when the tool’s rotational speed is at its minimum (300 rpm), and the traverse speed is at its maximum (210 mm/min), and the opposite is true as well. To attain fast welding while minimizing resistive force on the FSW tool, it is necessary to tilt both the rotating and traverse speed during the process.

The variation of total axial force as a function of process parameters is investigated in this study. When the rotational and traverse speeds reach their lowest and highest points, respectively, the minimal axial force is obtained. As a result, when the traverse and rotational speeds rise, the axial force increases. Increased rotating speed increases the lateral force substantially. They show that tool traverse speed does not affect lateral force. 

Numerical calculations depicted in Figure 39 display the forces exerted on the welding tool at different speeds. All forces experienced a significant increase as traverse speed escalated. The reason for this is the reduction in workpiece thermal softening caused by reduced heat generation, which raises the material resistance that acts on the FSW tool as it moves along the weld line. Conversely, raising the rotational speed had the opposite effect because it produced more heat [59].

Figure 40 illustrates the computation of the impact of tool diameter on total torque, as well as the sliding and sticking components. Specifically, the sliding torque (ML) exhibits exponential growth with increasing tool shoulder diameter, while the sticking torque (MT) experiences a sharp decline. Nevertheless, the decrease in sticking torque was not significant enough, ultimately resulting in an increase in total torque.

## 8. Material Flow during the FSW Process

The material flow plays a critical role in the FSW process, as it affects the microstructure, mechanical properties, and overall weld quality. Material flow refers to the movement and mixing of the two materials being joined along the joint line. As the rotating tool moves along the joint line, it creates a continuous mixing and deformation of the materials, resulting in a homogeneous microstructure. The material first undergoes plastic deformation in front of the tool, followed by lateral flow around the tool to form a solid-state bond. This continuous mixing and deformation ensure that the welding process occurs without melting the material, making it ideal for high-strength materials and alloys.

The importance of material flow in FSW is highlighted by its effects on the microstructure and mechanical properties of the weld. Proper material flow ensures a uniform distribution of grain structure, which results in better toughness, strength, ductility, and corrosion resistance. On the other hand, improper material flow can lead to defects such as voids, porosity, and cracks, which can reduce the strength and reliability of the joint.

The material flow in FSW is a complex process that is influenced by multiple variables, such as welding parameters, tool features, and the material being welded. Understanding the characteristics of material flow is crucial for achieving high-quality welds and optimal tool design. However, being a complex phenomenon, the underlying deformation mechanism during FSW is not yet fully understood. It is important to note that several factors can affect the flow of material during friction stir welding. These include the tool features, such as shoulder and pin design and their relative dimensions, welding parameters, material properties, workpiece temperature, and other relevant considerations. The material flow within the SZ, the region of the weld where the tool exerts its maximum influence, is likely to involve multiple deformation processes. The exact nature of these processes and how they interact with each other is still being studied by researchers and engineers in the field [28].

A study on the material flow of a workpiece has shown that it can be separated into three distinct layers: the top, middle, and bottom. The material flow characteristics of the AS and RS are found to differ significantly. On the RS, the material particles do not enter the AS, and those on the top surface are rotated towards the back but are not pushed into the stir zone by the tool. On the other hand, the material particles on the AS, which are rotated to the RS on the top surface, pile up near the border of the SZ at the RS. This accumulation of material causes weld flash to occur. [6]. 

The material flow within the sheet thickness is depicted in Figure 41 and Figure 42. Upon comparing it to the pattern of material flow on the top surface, one can infer that the material particles located at the center of the sheet may move into the area underneath the shoulder, unlike those on the top surface. The welding tool exerts pressure to push back the material particles on the RS, while the ones on the AS can be rotated by the tool for multiple rotations before being shed off from the spinning pin. Empirical evidence has also confirmed this type of material flow [6].

Figure 42 displays the pattern of material flow on the bottom surface, which indicates a significant reduction in the rotating zone at the center of the sheet thickness and on the bottom surface. This phenomenon is responsible for the inverse trapezoidal shape of the stir zone.

The influence of tool traverse and rotational speed on material flow has been studied, and it has been discovered that the stirring effect rises as the rotational speed increases. On the other hand, extreme increases in tool rotating speed might result in the creation of weld flash and, as a result, negatively impact weld quality.

Tutunchilar et al. [131] explored the material flow during FSP of LM13 alloy. Using Deform-3D software, they simulated the process, assuming a shear friction coefficient of 0.5. Figure 43 illustrates the material flow pattern that takes place at the weld center. The initial position of points along the center line is shown in Figure 43a, while their positions over the FSW are depicted in Figure 43b–e. Points P7 and P8 near the bottom of the SZ exhibit a semicircular movement from the leading edge to the trailing edge. This behavior was anticipated due to the frictional shear force generated by the tool shoulder. Despite the movement of the tool pin behind it, the material adjacent to the top surface stretches towards the advancing side. This phenomenon was expected. Notably, the final location of all points, except for P1, corresponds to the SZ form, which was a fascinating discovery.

Flow patterns in the AS and RS are depicted in Figure 44. As shown, points stated at the AS spin around the pin before returning to the AS. However, the points at the RS (Figure 44) do not behave in this way, and after passing through the tool, they are distributed over both sides. Furthermore, points near the SZ root remain at the RS, and material extending toward the AS accelerates as one gets closer to the top surface. In other words, the majority of materials from all sides, after passing the tool over them, are eventually located on the AS.

Jain et al. [173] investigated the effects of smooth conical and threaded conical pin profiles on the material flow characteristics. They showed the material movement around the pin through the RS, which is slightly stretched towards the AS, holding its final location almost behind the initial location. Moreover, the thread on the tool produces a skewing flow resulting in a vertical material movement. Roy et al. [174] introduced a 3D thermal pseudo-mechanical model within an Eulerian framework to simulate the FSW of AA6061 aluminum alloy under quasi-steady-state conditions. According to their findings, increasing the rotational speed and traverse speed of the pin boosted material flow, resulting in a distinct vortex on the AS. A CFD model was developed to simulate an aluminum steel dissimilar butt weld using Comsol v4.3 finite element software [175]. The simulated material flow was found to be laminar and influenced by the shape of the welding tool. 

A three-dimensional numerical model was showcased to analyze the material flow during the FSW process [176]. The findings suggest that the material ahead of the pin undergoes an upward motion owing to the pin’s extrusion effect and rotates along with it. On the other hand, after the tool rotation, the material experiences a downward movement and settles in the wake. A study was conducted to explore the direction of material flow during FSW of transparent polyvinyl chloride (PVC), to simulate flows in aluminum alloys. The investigation involved the use of a high-speed video camera, which helped capture the material flow in action. Particle image velocimetry (PIV) was also used to measure the material flow velocity. Numerical simulations were carried out with the help of DEFORM-3D FE software to understand the phenomenon further. Findings from the study revealed that the material flows in different directions at the AS and RS, with material velocities ranging from about 2 to 20 mm/s and 1 to 5 mm/s, respectively.

A study was conducted using a finite volume model of the FSW process to investigate how shoulder and pin geometry alterations impact the plastic flow behavior of the material [177,178]. It was observed that decreasing the width of the screw groove or reducing the pin cone angle could increase the material velocity inside the SZ. Additionally, for a left screw pin tool rotating clockwise, the direction of material flow near the pin is downward, while the material flow direction near the TMAZ is upward, which contrasts with the flow direction for a right screw pin.

The study examined the material flow during ultra-thin 2024 aluminum alloy plate friction stir welding [179]. The copper powder was utilized as a marker material to track the material flow. The investigation employed both visual experimentation and numerical computation to gain insights into the material flow. By analyzing the results, the study revealed that rotational speed significantly impacted material flow. As the rotational speed/welding speed ratio remained constant, an increase in rotational speed led to an increase in the flow velocity of plasticized material. Conversely, the flow velocity of plasticized material decreased as the distance from the weld center increased. Additionally, the maximum and minimum values of the flow velocity of plasticized material were observed at the intersection of the triple spiral groove of the shoulder and both sides of its outer diameter edge, respectively.

Bhattacharjee and colleagues [64] employed a 3D thermomechanical approach that combined the Eulerian–Lagrangian methods to examine typical flaws that arise during FSW. Their findings indicated that welding joints with slight tunnel defects were apparent when tilt angles were at 0°. Su et al. [180] developed a CFD model to investigate the relationship between FSW tool eccentricity and periodic material flow. The findings of their study indicate that tool eccentricity can be regarded as the primary cause of the observed periodic material flow behavior during the FSW process. 

Luo and colleagues [181] utilized the particle tracking method to investigate the behavior of material flow in different regions of a weld seam. The upper surface material was observed to move downwards due to the shoulder extrusion, while the lower portion material moved upwards in a spiral fashion as a result of the tool pin. Moreover, the amount of material flowing on the AS was found to be greater than that on the RS. The researchers also noted that abnormal material flow tended to cause defects like holes and cracks at a low rotational speed and high traverse speed. Specifically, they identified that when the welding speed was increased by two times, holes were formed in the lower portion of the AS (as depicted in Figure 45b). Additionally, a crack defect occurred after welding when the rotational speed was reduced by half (Figure 45a).

The impact of various pin profiles, such as conical and tapered hexagonal, on material flow during FSW was investigated by Naumov et al. [182]. In Figure 46, the flow stress iso-surfaces of 200 MPa were used to illustrate the variation in material flow between the two probe profiles. The iso-surfaces appeared similar for both probe profiles located directly below the shoulder and near its periphery. However, it shows distinctive morphologies for the distribution of flow stress around the tool pins. Naumov et al. demonstrated that the material reaches the stress of 200 MPa near all edges of the tapered hexagonal probe due to high strain rate values in these areas, in contrast to the pin with a conical profile where 200 MPa flow stress could only be observed below and around the pin bottom tip.

DEFORM™, an implicit Lagrangian code, was employed by Buffa [183] to elucidate the weld properties and material flow during FSW of Ti6Al4V and AISI 304. Their approach involved a single-block model, and to accommodate the distinct nature of both materials within the same joint, a numerical trick was implemented. The material model used was bi-phasic, with each phase representing one of the two materials—Ti6Al4V and AISI 304. The single-block workpiece was initially constructed entirely from Ti6Al4V, with a simulated “phase transformation” induced only in the region corresponding to the steel sheet. Once the transformation was completed, phase change was disabled throughout the object, thus yielding a workpiece composed of two distinct materials. Buffa et al. [177] utilized the same numerical approach to analyze the impact of process parameters on temperature distribution, strain distribution, and material flow in lap joints of AA2024 aluminum alloy and Ti6Al4V titanium alloy through FSW. To model a dissimilar FSW joint, they incorporated a fictitious phase transformation. The top sheet was created using titanium. During the experiment, an upward material flow was noticed at the sides of the SZ in the aluminum sheet, while a downward material flow was generated at the ccenterof the SZ in the titanium sheet. The combination of these two opposing material flows resulted in a distinct “waived” profile observed in both the macrographs and predicted by the model.

A numerical model based on computational fluid dynamics was developed by Padmanaban et al. [184] to forecast the material flow during FSW of dissimilar aluminum alloys—AA2024 and AA7075. Their analysis revealed that augmenting the rotational speed and shoulder diameter resulted in an upsurge of material flow while increasing the traverse speed reduced the material flow within the SZ. A coupled Eulerian–Lagrangian approach was employed by Das et al. [70] to simulate material mixing and defects numerically. They utilized the EVF method to anticipate the presence of dissimilar welding materials on the RS and AS of the weld centerline. An EVF value of 0.5 indicated that a single element contained both materials.

During the tool rotation, the material on the RS was transferred to the advancing side, while an equal amount of material from the AS was pushed into the RS. At the end of the dwelling stage, the cross-sectional view of the material flow is illustrated in Figure 47c,d. Figure 47c depicts the presence of some AZ31B on the advancing side, whereas Figure 47d shows the minimal existence of AA6061 on the RS. This demonstrated that material from the RS flowed more easily into the AS than vice versa at the end of the dwelling stage due to the lower modulus of elasticity of AZ31B compared to AA6061. The easy plasticization, rotation, and material deposition around the dwelling tool were noted as a result. The simulation results demonstrated that the welding plate position significantly influenced the formation of defects, both on the AS and RE. For example, when AZ31B was applied to the AS, the surface tunnel defect nearly penetrated half of the workpiece’s thickness. Moreover, the numerical model successfully simulated the development of defects caused by wear-induced alterations in tool dimensions. With increased tool traverse and rotational speeds, the pin length decreased by up to 30% after the process, leading to the formation of surface tunnel defects.

In a study by Mirzaei et al. [185], numerical modeling was used to explore the double shoulder friction stir welding of AZ91 magnesium alloy using the CEL method. The investigation focused on the impact of pin profile on material flow during welding (Figure 48). Results indicated that the movement of material in the shoulder-driven and pin-driven zones was distinct, with material flowing from the AS towards the RS in the latter. The authors noted that while the trigonal pin could move larger boxes of material due to its longer sweeping lever, the hexagonal pin tool produced the most even material movement and optimal material coalescence. Conversely, the inward conical pin tool resulted in weak material flow patterns and tunnel cavity formation in the welding zone. However, the pin with an outward conical profile created a subsidiary material movement that improved material mixing, leading to increased strain, finer grain size, and higher mechanical properties compared to the pin with an inward conical profile.

In a research article by Akbari et al. [80], dissimilar friction stir lap welding of aluminum and brass using the CEL method was studied to investigate material mixing and SZ formation. The model provided successful predictions of SZ formation and material mixing at the Al/CuZn34 interface, with aluminum penetrating the center of brass and ear-shaped brass penetrating the Al sheet at the periphery of the SZ (Figure 49).

Furthermore, the authors demonstrated that decreasing the traverse speed increased the amount of dispersed brass within the aluminum matrix while increasing the traverse speed had the opposite effect. This outcome was attributed to improved heat production and favorable material flow.

## 9. Defect Prediction Using the Numerical Method

During FSW, defect prediction using the numeric method involves using numerical models and simulations to predict potential defects in the weld joint. The defects that can occur in FSW can be classified into three main categories: inadequate heat, design errors, and excessive heat. The tunnel defect is a volumetric flaw that occurs when there is a loss of material around the tool in the weld zone. This type of defect is commonly observed in samples that have been welded with both large and small values of welding speed. The occurrence of a kissing bond results from inadequate fusion of the weld material, leading to a mechanical bond instead of a metallurgical one. This type of defect is primarily detected in the aerospace industry during welding processes. Kissing bonds have the potential to weaken joints significantly, making it challenging to identify and rectify such defects. Insufficient plunge force during the welding process leads to a lack of fill. This defect occurs as a result of an incorrect selection of weld attributes. However, these defects can be more easily monitored. The FSW process can lead to the occurrence of different defects on the surface and within the material. The plunge action of the tool into the workpiece primarily causes the formation of a flash. As the tool rotates, more material gets deposited on the RS, contributing to increased flash formation.

So far, several researchers have used numerical methods to investigate and predict the formation of defects during FSW. Liu et al. [186] developed a comprehensive thermomechanical model for observing material movement around a tool. Their findings highlighted the complex nature of material flow on the AS horizontal and vertical planes, which differs greatly from that on the RS. Notably, the researchers discovered a region with low material density located at the back of the pin. This discovery sheds light on the underlying cause of defects, such as grooves or voids in the material. Using the CEL approach, Chauhan et al. [121] studied the impact of various welding settings and tool pin profiles on void formation. The findings showed that FSW with lower traverse speeds or higher tool rotational speeds could produce fewer voids. For the FSW of the dissimilar alloys AA2024-T3 and AA6061-T6, it was necessary to ascertain the thermal and subsequent residual stress conditions. Hossfeld [187] simulated the entire FSW process of AA 5182 using the CEL approach. The level of resolution attained in this study makes it possible to see how burrs and internal voids form.

Das et al. [70] compared the experimental flash obtained and the predicted flash. On average, there was a difference of approximately 1 mm in the predicted flash size compared to the experimental value. This deviation can be attributed to the consideration of mesh size and human error during measurements (Figure 50).

Moreover, they predicted the existence of a surface tunnel on the AS through numerical analysis. The experimental surface tunnel had a width of around 1 mm consistently, while the predicted result was approximately 0.8 mm. However, at a specific location in the defect’s initial region, the width of the tunnel measured around 2 mm (Figure 51). They claimed that the slight discrepancy between the predicted and experimental outcomes could be attributed to errors in the machining process or measurements.

Guerdoux and Fourment [188] established an FE model utilizing ALE formulation in FORGE software. The study employed Hansel Spittel’s material model, re-mesh technology, and Norton’s law to replicate the frictional contact between material and tool. The researchers discovered that flash production on the plate top surface and void development occurred during FSW. The study predicted the appearance of flashes and voids, which were confirmed using experimental data related to welding torque and force. The FE model provides valuable insights into the stages of welding and material flow.

Chen et al. [189] introduced a CFD model incorporating shear stress boundary conditions and non-uniform contact pressure distribution. This model was specifically designed to analyze the flow of plastic materials and the formation of welds in FSW. The researchers investigated the underlying mechanism behind the occurrence of void defects and explored the influence of tool pin size on plastic material flow and weld formation.

Their analysis revealed that increasing the diameter of the pin tip led to an expansion in the width of the void defect, accompanied by a decrease in its height. The larger pin tip diameter resulted in elevated temperatures near the tool pin tip, thereby enhancing the flow of plastic materials. Consequently, both the horizontal and vertical migration distances of the plastic material within the workpiece increased, leading to void defects with wider width but reduced height. Furthermore, the researchers observed that increasing the pin root diameter caused a decrease in the height of the void defect. This can be attributed to the fact that the vertical migration distances of plastic materials at the upper portion of the workpiece were amplified. However, the width of the void defect did not exhibit a distinct trend with changes in the pin root diameter.

Zhu et al. [190] conducted a study on the prediction of wormhole defects arising from different welding parameters. Using a CFD model, they visualized these defects using the distribution of tracing particles incorporated into numerical models. The inclusion of tracing particles in the CFD model allowed for the examination of material plastic flow behavior during the welding process as a post-processing step. The tracing particles originated from the starting point on the plane and proceeded in the direction of the negative X-axis. Under the influence of the welding tool, they advanced toward the backside. Once reaching a certain distance from the tool, the Z and Y coordinates of these particles remained constant. Subsequently, an observation plane was positioned behind the tool to evaluate the distribution of materials in the joint region post-welding.

Figure 52a,b illustrates the distribution of tracing particles in the joint area under five sets of parameters. In the observation plane, continuous grey zones indicate areas with densely arranged particles, while blank zones indicate areas with a low particle flow. The simulation results showed that blank zones were present in all five simulations with a non-uniform friction force boundary, specifically on the AS of the SZ. This indicates that the material, which initially covered the entire joint area, could not form dense joints after the tool pin’s action. Additionally, there was a notable tendency to form wormholes on the AS of the SZ, but this phenomenon did not occur in simulations using a partial sticking boundary. The simulation results were validated through experimentation, as shown in Figure 52c. Analyzing the wormhole distribution in the experimental results, it was observed that when the tool pin rotation speed of 800 rpm and the bottom diameter increased from 4 mm to 6 mm, the size and position of the wormholes remained relatively unchanged. However, at the rotational speed of 800 rpm and bottom diameter of 3 mm, the wormhole defects noticeably decreased in size. Furthermore, maintaining a bottom diameter of 3 mm and reducing the rotational speed from 800 rpm to 600 rpm resulted in an obvious increase in the wormhole defects’ width (W) and height (H).

In their study, Ajri and his colleagues [191] examined the occurrence of cavities and tunnel defects in Al 6061 T6 alloy during FSW. Both numerical and experimental methods were employed. The researchers found that the peak temperature reached during the formation of tunnel defects was below the solidus melting point, while the temperature was higher in the case of cavity defect formation. Additionally, they observed that the pin successfully drove the material to the AS of the weld when cavity defects formed but failed to do so during tunnel defect formation. Tunnel defects were more common at lower rotational speeds, while cavity defects were more prevalent at higher welding speeds.

Using the CEL approach, Zhu and colleagues [188] constructed a 3D thermomechanical finite element model to analyze and predict the creation of defects during FSW. The findings demonstrate a close agreement between the microstructure observed in experimental studies and the estimated shape of the equivalent plastic strain zone. Additionally, the developed FE model successfully predicts the presence of voids in the weld, as shown in Figure 53. However, it is worth noting that the model tends to overestimate the size of the void. The estimated void size is 2.18 mm × 0.55 mm, whereas the measured result is 1.09 mm × 0.7 mm.

Choudhary and colleagues [84] formulated a numerical model utilizing the CEL method to effectively analyze and anticipate various types of defects, including void, tunnel, root defects, and cavities. Upon examination, it was observed that a tilt angle of zero degrees corresponded to the occurrence of tunnel defects. Additionally, a tilt angle of one degree correlated with cavity and void defects. Remarkably, welds completed with a two-degree tilt angle exhibited defect-free outcomes. Conversely, welds conducted with parameters employing a three-degree tilt angle exhibited flaws. The formation of these defects is deeply interconnected with factors such as temperature history, material flow, and axial force.

## 10. Microstructural Modeling and Simulation

Over the past few years, numerous models have been suggested to replicate microstructure transformations in metallic alloys during the FSW process and resulting hardness profiles. The ultimate aim of these models is to ascertain the final properties of combined materials and the positive/negative consequences of the FSW process on them. The development of these approaches is typically tailored based on the authors’ objectives and ideas and is executed at a specific scale. Furthermore, the methods commonly discussed in the literature can be divided into three primary categories:Molecular dynamics models;Precipitation modeling;Grain evolution modeling.

### 10.1. Molecular Dynamics (MD) Models

Molecular dynamics (MD) models are a type of computational simulation that use Newton’s laws to study the behavior of atoms and molecules in a system over time. In the context of FSW, MD models simulate the movement of atoms around the weld zone as they heat up and interact with one another during the welding process.

MD models can provide information on the evolution of the microstructure during FSW, including the formation of dislocations, grain boundaries, and other crystal defects. They also offer insights into the role of different parameters, such as temperature, pressure, and deformation rate, on the welded joint’s final properties.

One of the significant advantages of MD models is their ability to capture the atomic-level details of the welding process, which is impossible with other modeling approaches. However, MD models are computationally expensive and require high-performance computing resources to run simulations for a sufficient duration to obtain meaningful results.

There have been some studies that suggest the feasibility of defect analyses through molecular dynamics simulations. Dmitriev et al. [192] conducted an atomic scale modeling study to investigate the mechanisms responsible for structural inhomogeneity during FSW when significant plastic deformation occurs. The simulation involved the movement of atoms induced by the tool’s motion, resulting in non-equilibrium states in the crystal lattice that were analyzed and discussed.

Nikonov et al. [193] proposed a simulation to reproduce atom movement induced by mixing when a cylindrical tool passes between two inter-crystallite boundaries of similar grain orientation. In this instance, copper was utilized as the material, and the repositioning of atoms was visually represented using different shades of grey to depict the material’s movement. The region affected by FSW had a thickness equivalent to the tool size, approximately 3.6 nm. At a greater distance, the atoms maintained their initial positions. The arrangement of atoms in both metals and any structural flaws were presented in the same orientation. The material was displaced towards the right-hand side through a combination of feeding rate and rotational directions. Analysis was conducted using a common neighbor method, which revealed minimal structural defects within the pieces. The original local topology remained well-preserved, as anticipated, in the vicinity of the original boundaries.

Dmitriev et al. [192] argue that molecular dynamics simulations have the potential to enhance our comprehension of the fundamental principles that govern the development of structural inhomogeneity in FSW processes. Nevertheless, merging the outcomes of molecular dynamics simulations with the macro-scale can present certain challenges. Establishing a connection between the process parameters utilized in molecular dynamics simulations and real-world process values can be difficult, thereby complicating investigations into the impact of FSW process conditions on defect occurrence, weld quality, and the optimization of welding parameters. This issue has been extensively addressed in the existing literature.

### 10.2. Precipitate Size Distribution (PSD) Models

In materials science and engineering, precipitation modeling is focused on studying the evolution of precipitates within a metal matrix at the micro-scale. These approaches aim to understand how the size distribution of precipitates changes over time during the FSW process, as induced by temperature and heat evolutions.

Numerous models have been documented in the literature that allow the tracking of phase fractions during the FSW process through additive approaches that are considered semi-analytical. According to Simar et al. [182], these approaches are called “Internal variable models”, relying solely on numerical integration of the material’s thermal cycles. The initial estimation of temperature evolution during the FSW process is conducted on a macro-scale through thermomechanical simulations or analytical estimation. Subsequently, the evolution of precipitates is calculated incrementally using an integrative approach, where the temperature changes serve as input to determine the gradual increase or decrease in precipitate fractions during infinitesimal time intervals. To couple precipitate evolution to FE simulation, the dissolution model is kept simple following the proposal by Myhr and Grong [194].

Frigaard et al. [195] utilized a model to anticipate microstructure changes in the HAZ and present comparisons with experimental observations. Validation of the heat flow model introduced for welding temperature evolution was also conducted through these comparisons. The primary focus of the microstructure evolutions is on the Vickers hardness measurement. Two grades of aluminum, AA7108-T6 and AA6082-T6, were examined. The process model accurately predicts the response of the parent material, as evidenced by the similarities in predicted and expected hardness profiles. As the cooling process takes place, there is an evident reduction in hardness within the HAZ. This decrease can be attributed to the partial dissolution of hardening phases, specifically β″ and η′, present in both grades. Consequently, non-hardening phases, namely β′ and η, begin to grow while the aluminum matrix experiences a depletion of solutes.

The PSD model was first introduced by Wagner and Kampmann [196] and subsequently enhanced by Myhr and Grong [197] through a finite difference approach. This method allows for the monitoring of the evolving precipitate size distribution across all stages of the precipitation process. Compared to the semi-analytical models outlined earlier, this approach provides a more comprehensive understanding of the size distribution and precipitate evolution, which is crucial in predicting complex phenomena and properties such as corrosion behavior or fracture toughness, as previously noted.

Several authors have utilized a model to simulate the evolution of PSD during the heating and cooling stages in FSW processes. Gallais et al. [198] noted that there are numerous models reported in the literature, ranging from Monte Carlo to phenomenological approaches, which cover various time and spatial scales. However, both approaches have limitations and cannot provide relevant information on material enduring non-isothermal transformations at the weld scale. Consequently, the PSD model appears to be the only relevant approach capable of providing valuable data by integrating the entire precipitation stages endured in materials, including growth, nucleation, and coarsening. Simar et al. [199] proposed one of the initial approaches exclusively focused on modeling microstructure evolution in FSW. Their approach specifically took into account the impact of supersaturated solid solutions and tracked the process of dissolution and coarsening of fine-hardening precipitates during thermal cycles. Notably, they achieved successful comparisons between their model predictions and experimental findings.

A PSD model focused on AA7449 aluminum grades, which includes recrystallization mechanisms, was developed by Dos Santos et al. [200]. This model was integrated into a process model that utilizes the CFD framework developed in FLUENT. The PSD model tracks the development of three distinct precipitate populations and considers the impact of grain refinement on precipitation mechanisms in a coupled approach between grains and precipitates. This method can be applied to various aluminum grades and process parameters, and it closely imitates the size distribution and temporal evolution of precipitates.

### 10.3. Grain Evolution (GE) Modeling

Dynamic recovery (DRV) and dynamic recrystallization (DRX) are two major microstructure evolution characteristics. Which one happens in a material during SPD and high temperature depends inherently on the level of stacking fault energy (SFE) [201]. Low and medium SFE materials such as 304 stainless steel typically exhibit discontinuous dynamic recrystallization (DDRX), where nucleation of annealed grains happens during strain, according to Humphreys and Hatherly [202]. In the alloys with a high SFE, such as aluminum alloys and α-iron, a perfect dislocation cannot be simply broken into two fractional dislocations, and the mechanisms of glide, climb, and cross-slip for dislocations play a crucial role. Therefore, a rapid dynamic recovery occurs and effectively prevents the accumulation of adequate dislocations to ignite DDRX during FSW [57]. Thus, subgrain boundaries, which originated from the substructures and are composed of tangled dislocations, are formed with only limited dislocations during deformation. In addition, the continuous material deformation progressively increases the misorientation of subgrain boundaries, leading to the change in low-angle grain boundaries (LABs) into high-angle grain boundaries (HABs). This phenomenon is described as continuous DRX (CDRX) [203].

Furthermore, the high SFE materials may be subjected to a sufficient deformation in one direction (e.g., hot rolling or torsion), which may activate geometric DRX (GDRX) when the criterion that the grain thickness is reduced to by 1~2 times the subgrain diameter is satisfied. Indeed, GDRX suggests that elongation occurs in deformed grains until serrations are pinched off, leading to the appearance of new equiaxed grains and a decrease in grain thickness. Since aluminum alloys are considered high SFE alloys, CDRX and GDRX commonly occur when they are taken into high-temperature SPD [204].

However, there are limited numerical models for both CDRX and GDRX, so current reviews on grain size evolution in the SZ mainly focus on these two approaches. While the mechanisms behind GDRX are generally well understood, its modeling is often based on simple geometric assumptions such as cubic or spherical grain shapes, as proposed by Gholinia et al. [205] for an Al-3Mg-0.2Cr-0.2Fe alloy. Similarly, Gourdet et al. [206] used MacQueen’s GDRX theory based on similar hypotheses. A more sophisticated model was employed by De Pari and Misiolek [207], who utilized a truncated octahedron to produce significant results.

Using FE or analytical models can help predict the thermomechanical history, which can then be coupled with microstructural evolution. Various methods can be used to develop the latter, including fully analytical models or less conventional approaches such as cellular automaton (CA) or Monte Carlo (Potts models). Microstructure evolution modeling can be categorized into three groups: (i)Material models based on physical properties and evolution laws such as DDRX, CDRX, or GDRX models;(ii)Empirical methods commonly used in cellular automaton–finite element (CAFE) models but require extensive calibration steps;(iii)Monte Carlo methods consider final observations as a possible evolution through stochastic simulation.

Limited sources in the literature focus on modeling microstructure evolution in FSW using the DDRX approach to nucleate new grains in aluminum alloys during stirring processes. However, Hofmann and Vecchio [208] developed activities to apply the Derby–Ashby [209] model was utilized to analyze the cooling curves derived from thermocouple measurements of temperature in stirred aluminum alloys. This particular model is specifically connected to DDRX phenomena, and it represents one of the limited instances documented in the literature where DDRX phenomena during the stirring process of aluminum alloys have been modeled.

Hofmann and Vecchio [208] investigated two processes, FSP and submerged FSP (SFSP). The purpose of these processes is to create bulk samples with a microstructure consisting of fine grains through the application of severe plastic deformation on the material. Though FSP and SFSP differ from the FSW processes investigated in this article, both lead to large recrystallization mechanisms when the material undergoes a high stirring mechanism. In both cases, strain and temperature evolutions on the material induce recrystallization, leading to a new microstructure. 

Wan et al. [210] introduced a Zener–Hollomon (Z-H) parameter as a means to represent the grain size transformation process in friction stir welding. This strategy is associated with the formation of new grains along existing grain boundaries, and it aligns with the physical phenomena observed in DDRX methods. The anticipated sizes of the grains vary from 9.32 to 9.62 μm in proximity to the upper surface and from 8.29 to 8.84 μm near the lower surface. As the rotational speed escalates, there is an increase in the average grain size on both the bottom and top surfaces. Specifically, the average grain size progresses from 9.11 μm at 715 rpm to 27.5 μm at 1500 rpm. The authors provide an explanation that emphasizes the heightened significance of cooling during the thermal cycle at higher rotational speeds. This leads to particles experiencing prolonged cooling durations, thereby resulting in extended grain growth and recovery after recrystallization.

The research conducted by Shojaeefard et al. [132] utilized an Avrami model from DEFORM-3^DTM^ software coupled with numerical fields to predict microstructural changes during the FSW process. They also used a modified LJ model coupled with a CA method to simulate recrystallization mechanisms and then compared their experimental results. The study focused on AA1100 aluminum grades. The microstructure evolution model was based on both grain growth kinetics and nucleation rate, categorizing it as a DDRX approach. Figure 54 presents the results obtained by Shojaeefard et al. [132] on AA1100 aluminum grades, which showed good agreement between experimental and simulation measurements in the SZ.

Robson and Campbell [211] present an innovative model for microstructure evolution in the SZ, focusing on recrystallization and grain growth. The foundation of their approach lies in the GDRX model initially proposed by Prangnell and Heason [212]. Additionally, they have developed a process model capable of accurately predicting the primary thermomechanical fields within the SZ. All experimental tests and computations have been calibrated explicitly for AA2524 aluminum alloys. Notably, the research findings have also been extended to explore the influence of dispersoid particles and the effect of cooling rate. The study also takes into account the phenomenon of grain coarsening after stirring.

The widely adopted Gourdet and Montheillet (GM) model [213] presents a comprehensive framework for understanding continuous dynamic recrystallization (CDRX). According to this model, CDRX can be attributed to the interplay of three fundamental mechanisms: dynamic recovery, strain hardening, and high-angle grain boundary (HAGB) migration. The GM model characterizes the polycrystalline structure by analyzing the dislocation density distribution within the joint and sub-joints during deformation.

During the process, a portion of the dislocations generated by strain hardening forms new sub-joints with minimal disorientation angles (approximately 1°). Meanwhile, the remaining dislocations either vanish at the grain boundaries or merge into pre-existing sub-joints. In the latter case, the disorientation of the sub-joints gradually increases, eventually reaching a critical angle (approximately 15°) where they transform into joints. The model considers grain boundaries as mobile interfaces that undergo an elimination mechanism when encountering migrating dislocations. A fraction of the recovered dislocations contributes to forming low-angle grain boundaries (LAGBs) while existing boundaries assimilate others. Additionally, some dislocations are eliminated by HAGBs. 

Applications of Monte Carlo models in FSW modeling have been utilized to analyze the alteration in grain texture maps during the forming process, specifically in response to temperature fields. Some literature exists on developing and applying Potts models to simulate microstructure evolution in FSW processes. Grujicic et al. [214] implemented a thermomechanical model incorporating microstructure evolution through a Monte Carlo simulation algorithm. This model successfully predicted grain structure changes within the weld zones throughout the entire process, including the cooling stage. In this model, temperature variations were computed using an FEM. The competition between recrystallization mechanisms and grain growth, which contributes to grain refinement, was modeled based on local temperature distributions.

Zhang et al. [215] similarly employed a similar approach to study grain growth evolution in AA6082-T6 aluminum grade. Additionally, particles were monitored through thermomechanical simulations to estimate the stirring domain known as the TMAZ. The region experiencing grain growth evolution is also associated with HAZ. Zhang emphasized the efficacy of these methods in accurately predicting grain growth, topological features, and various phenomena such as welding, abnormal grain growth, anisotropic grain growth, and polycrystalline microstructure evolution. It is worth noting that these models are commonly applied to simulate grain structure alterations resulting from recrystallization processes and the corresponding changes in temperature fields leading to texture evolution in different domains. Consequently, this comprehensive approach provides a final representation of the microstructure field induced by the FSW process.

The research conducted by Yu et al. [216] utilized the Monte Carlo technique to create a computational model for simulating the dynamic recrystallization process in FSW of aluminum plates. The study involved selecting a suitable DRX nucleation model designed explicitly for Monte Carlo simulation. Furthermore, they adjusted the Monte Carlo simulation step to ensure its correlation with real-time conditions.

On the transverse cross-section, within the horizontal plane that includes tracking point C2 (positioned 2 mm below the top surface of the workpiece), a total of five points were chosen on both the advancing and retreating sides, as depicted in Figure 55a. These selected points were situated at a distance of 3–7 mm from the center C2 of the weld nugget zone (WNZ), with an increment of 1 mm. The simulated grain structure at these locations within the WNZ and the TMAZ of the final weld was illustrated in Figure 55b,c, respectively, revealing the impact under two distinct welding conditions.

Due to the lower rotational speed of the tool, which is associated with reduced heat input, the grain size in the case of 600 rpm is observed to be smaller than that of 800 rpm, as depicted in Figure 55b,c. As the distance from the joint center (C2, 0 mm) increases, the area occupied by refined dynamically recrystallized grains gradually decreases. Finer grains are observed within a range of 4 mm from the weld center. Complete DRX takes place within a 3 mm range around the weld center.

Additionally, there are notable differences in grain structures between the advancing and retreating sides. On the advancing side, the microstructures within a 5–6 mm distance from the center exhibit a distinct boundary separating the weld and the base metal. However, the microstructure variation is less pronounced in the same region on the RS. In both test cases, not only the grain size but also the distribution of grain structure exhibits noticeable dissimilarities, particularly on the RS of the joint.

According to the illustration provided in Figure 55, it can be observed that the number of DRX grains on the RS is higher under the condition of 600 rpm compared to that at 800 rpm. This suggests that as the heat input increases during FSW, the extent of dynamic recrystallization within the weld diminishes. One potential explanation for this phenomenon is that dislocations become more prone to activation and migration under higher temperature conditions, resulting in significant dislocation annihilation. Consequently, the reduced density of dislocations leads to a decreased likelihood of dynamic recrystallization taking place.

Zhang and colleagues [217] utilized a sophisticated three-dimensional Monte Carlo model, incorporating nucleation in each Monte Carlo step, to simulate the process of grain growth during FSW. Their findings indicated that the presence of equiaxed grains within the SZ escalated as the shoulder’s diameter and rotation speed were augmented. Moreover, an exciting observation emerged, revealing that as the rotational speed or shoulder diameter increased, the grain sizes tended to greater uniformity from the RS to the AS.

In their study, Wu and colleagues [218] employed the Monte Carlo method to simulate grain growth during FSW of AA6082-T6 while considering the effects of precipitation. The results revealed that reducing the volume fraction of precipitation from 0.8% to 0.2% after welding could increase the final grain size in the nugget zone by approximately 39.7%. Additionally, it was observed that both the speed of grain growth and the ultimate grain size on the top surface exceeded those on the bottom surface. This disparity can be attributed to the rise in welding temperature resulting from increased rotation speeds or axial forces, which subsequently lead to decreased volume fractions of precipitates and, consequently, larger grain sizes.

Khodabakhshi and colleagues [219] devised a novel approach that merged CFD modeling with Monte Carlo simulation to forecast the process of grain refinement during the FSP of an Al-Mg alloy. Considering crucial parameters such as tool rotational speed (*w*) and traverse speed (*v*), heat and strain rate distributions were simulated and subsequently utilized as inputs for a statistical model of dynamic recrystallization. As a result of FSP, equiaxed grains were generated, exhibiting average sizes ranging from 3 to 10 µm. These dimensions depended on the heat input index expressed in terms of *w*/*v* ratios falling within the range of 4 to 28 rev/(min·mm).

## 11. Optimization of FSW Based on Residual Stress Modeling

The literature provides several examples of utilizing numerical optimization methods alongside process modeling of FSW. Most of these examples focus on thermal models and aim to achieve optimal process parameters based on predetermined objectives. Additionally, these methods are employed for inverse modeling to determine unknown properties such as heat transfer coefficients [149]. Below, a few noteworthy instances are discussed.

Liao and Daftardar [220] conducted a study using a thermal model in FLUENT, along with two less complex surrogate models, to examine the effectiveness of various optimization algorithms in determining the three process parameters: weld speed, heat input, and shoulder diameter. The highest potential travel speed is primarily limited by the minimum temperature threshold, which directly affects the optimal solutions. The lower this threshold temperature, the greater the achievable travel speed can be. Tutum et al. [221] integrated a gradient-based optimization technique known as sequential quadratic programming (SQP) with a straightforward analytical thermal model. This approach allowed them to determine the optimal welding speed and heat input required to achieve a desired average temperature distribution beneath the tool shoulder during the FSW process.

Tutum and Hattel [222] conducted an optimization study on FSW by utilizing residual stress calculations. In this study, they utilized a thermomechanical model in ANSYS software (without considering material flow) in conjunction with the NSGA-II to identify the optimal values for rotational and traverse speeds. Their objective was to simultaneously decrease the peak residual longitudinal stress in the weld and increase the traverse speed. In addition, when the rotational speed remains constant, increasing the traverse speed typically leads to slightly elevated stress levels in the tension zone. Conversely, if the traverse speed remains fixed, increasing the rotational speed results in lower maximum residual stress levels, although it also widens the tension zone, ultimately resulting in a significantly greater residual tensile force. 

Lu et al. [223] utilized the DEFORM software to create a simulation process model for FSW of 18 mm thick 2219 aluminum alloy. This model enabled them to accurately predict the temperature distribution and optimize the welding parameters. Orthogonal experiments were conducted, taking into account the press amount, rotational speed, tool tilt angle, dwelling time, and plunging traverse speed. The goal was to achieve the minimum temperature difference in the core area of the weldment, with the constraint conditions being the weldable temperature range.

Su et al. [224] utilized a 3D CFD model to comprehensively observe the FSW process, taking into account the impact of polygonal pin profiles. The model was employed to quantitatively analyze the distribution of temperature, heat generation, welding loads, and plastic material flow during various FSW processes using tools with polygonal pin profiles, along with different shoulder diameters, tool rotation speeds, and traverse speeds. In particular, a methodology was developed to optimize the number of flats on the pin by examining the torque components in both the parallel and vertical directions of the pin-side flat region. The results indicated that the optimized number of pin flats increased as the tool rotation speed rose, while the influence of both shoulder diameter and traverse speed were considered to be insignificant.

Fraser et al. [141] proposed a meshfree computational framework to effectively determine numerically optimized process parameters, with a focus on reducing defects in the SZ. They introduced a simulation code, which employs a novel parallelization strategy on the graphics processing unit (GPU). This code enables the identification of the ideal traverse speed and rotational speed, while simultaneously minimizing defect volume using the proposed defect metric.

A thermomechanically coupled FEM was utilized to generate simulated data for FSW of AA5083 aluminum sheets [7]. The FEM produced datasets by calculating the welding force, maximum temperature, and HAZ width for various combinations of rotational speed and traverse speed. These datasets were then used to train an ANN model, acting as a virtual machine, to generate additional datasets with reasonable computational costs. The predicted datasets from the ANN model were subsequently employed in optimization processes to determine the optimal process parameters. These parameters aimed to minimize the width of HAZ, maximize the peak temperature, and minimize the welding force. Through experimental trials, significant improvements in the mechanical properties of the welded joints were achieved by implementing the optimized process parameters.

## 12. Summary Conclusions

Numerical models play a crucial role in the advancement of friction stir welding, providing numerous benefits that contribute to the development and optimization of this innovative joining process. Numerical models simulate the complex thermal and mechanical phenomena occurring during FSW, allowing researchers and engineers to gain a deeper understanding of the process. These models help elucidate the intricate interaction between the rotating tool, workpiece, and heat generation, aiding in the identification of critical parameters and their impact on weld quality. Moreover, numerical models assist in predicting and mitigating defects commonly associated with FSW, such as voids, tunnelling, or excessive flash formation. These models simulate the material flow, temperature distribution, and stress state during welding, enabling the identification of potential defect-prone zones. 

This paper provides a comprehensive review of the research activities and advancements made in numerical analysis techniques for friction stir welding. The focus is on the applicability of these techniques to component manufacturing. To begin with, various types of numerical methods and modeling techniques employed in FSW analysis are examined. These methods include FE analysis, CFD, and other simulation approaches. The advantages and limitations of each method are discussed, shedding light on their suitability for FSW simulations.

Next, the paper delves into the variables that play a crucial role in the numerical modeling of the FSW process. Important factors such as tool geometry, rotational speed, and traverse speed are thoroughly considered. The influence of these variables on the thermal and mechanical behavior during FSW is explored, providing insights into their impact on weld quality.

Furthermore, the paper highlights the modeling of microstructure behavior in FSW. This aspect focuses on predicting the evolution of grain structure, phase transformations, and material properties within the weld zone. The importance of accurately capturing these microstructural changes is emphasized as they directly affect the mechanical strength and integrity of the welded joint. 

## Figures and Tables

**Figure 1 materials-16-05890-f001:**
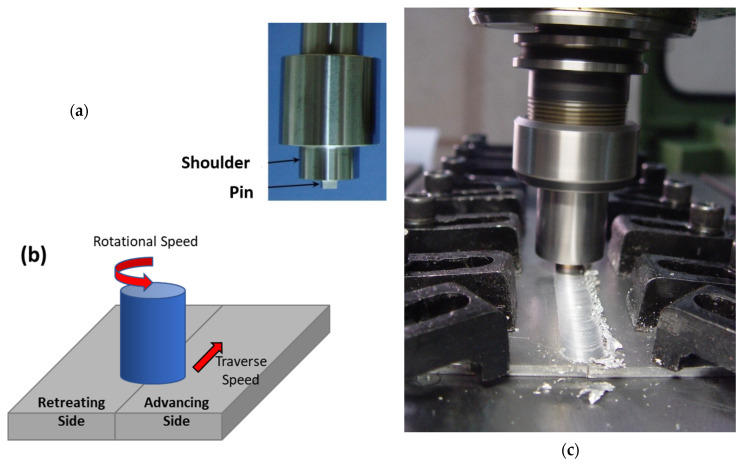
(**a**) A simple FSW tool, (**b**) a schematic of the FSW process, and (**c**) a setup of FSW illustrating the specially designed fixture.

**Figure 2 materials-16-05890-f002:**
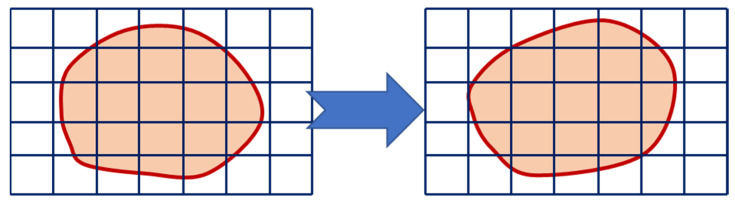
Split operation of the Eulerian formulation.

**Figure 3 materials-16-05890-f003:**
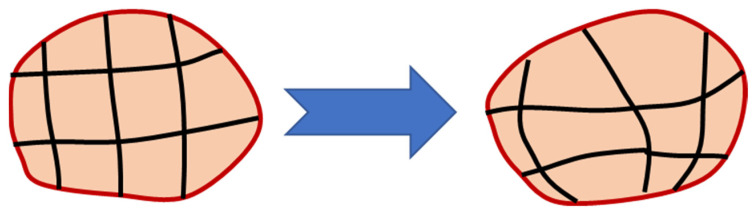
Operation of the Lagrangian formulation.

**Figure 4 materials-16-05890-f004:**
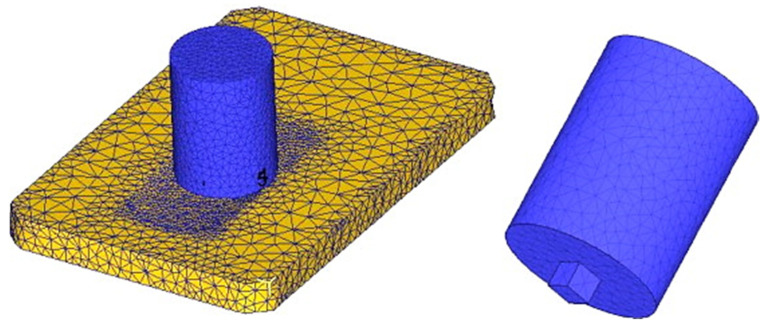
Schematic illustration of the FSP tool and workpiece [109].

**Figure 5 materials-16-05890-f005:**
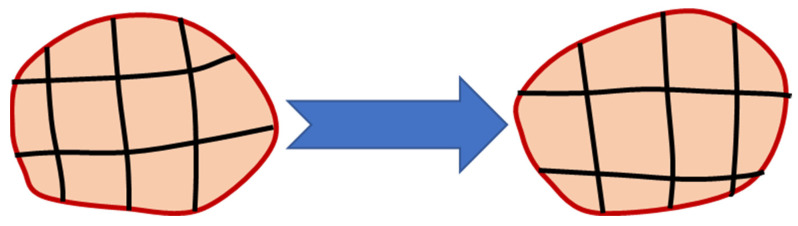
Operation of ALE formulation.

**Figure 6 materials-16-05890-f006:**
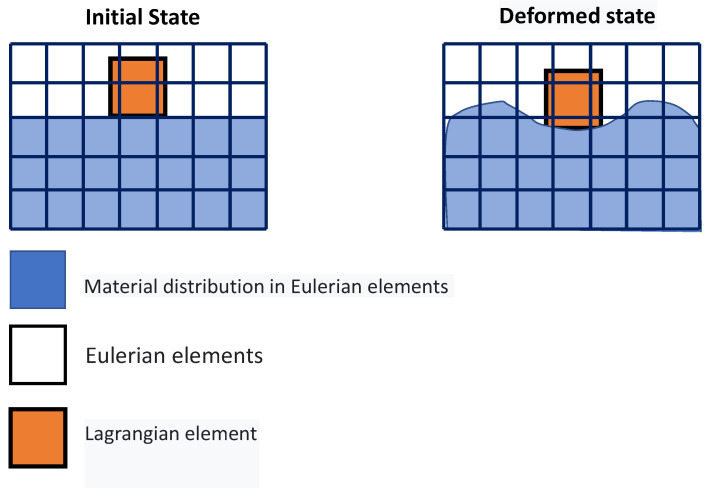
The coupled Eulerian–Lagrangian method.

**Figure 7 materials-16-05890-f007:**
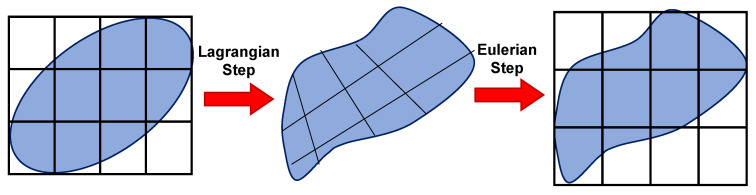
Illustration of split operator used in the CEL approach.

**Figure 8 materials-16-05890-f008:**
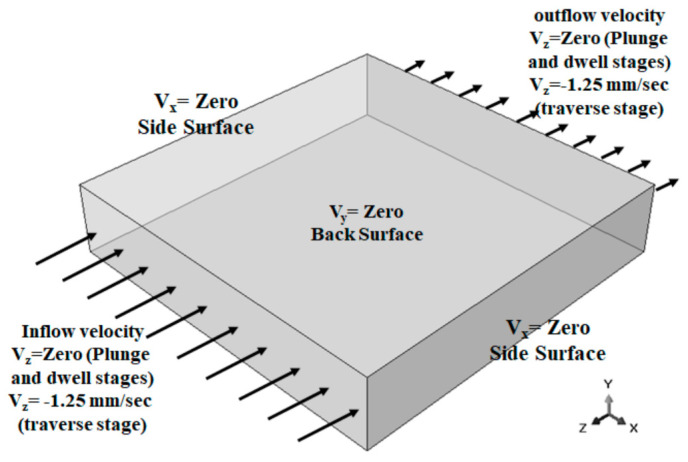
Velocity boundary conditions employed in Ragab et al. model [82].

**Figure 9 materials-16-05890-f009:**
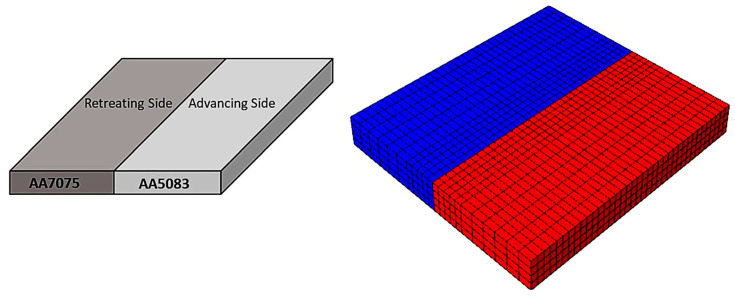
The placement of alloys during FSW. Blue sheet is AA7075 at the retreating side and red sheet is AA5083 at the advancing side [26].

**Figure 10 materials-16-05890-f010:**
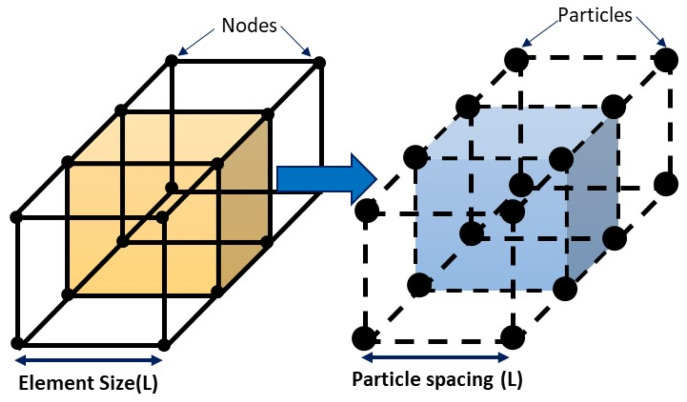
The schematics of the conversion of FEM to SPH models.

**Figure 11 materials-16-05890-f011:**
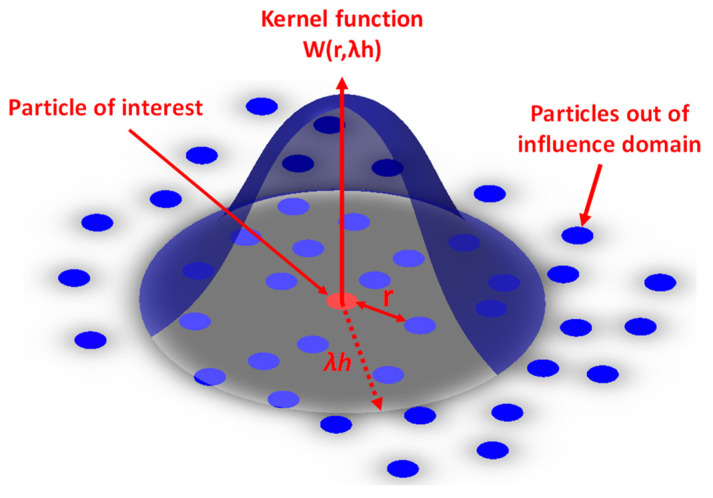
Interpolation in SPH approach.

**Figure 12 materials-16-05890-f012:**
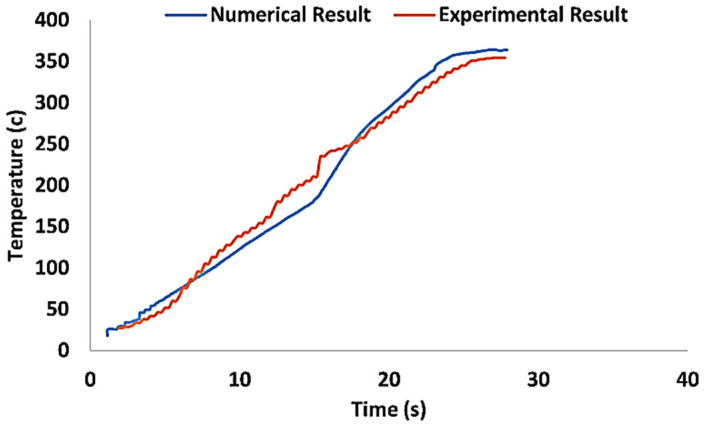
Comparison of welding temperatures obtained from experimental and numerical methods [25].

**Figure 13 materials-16-05890-f013:**
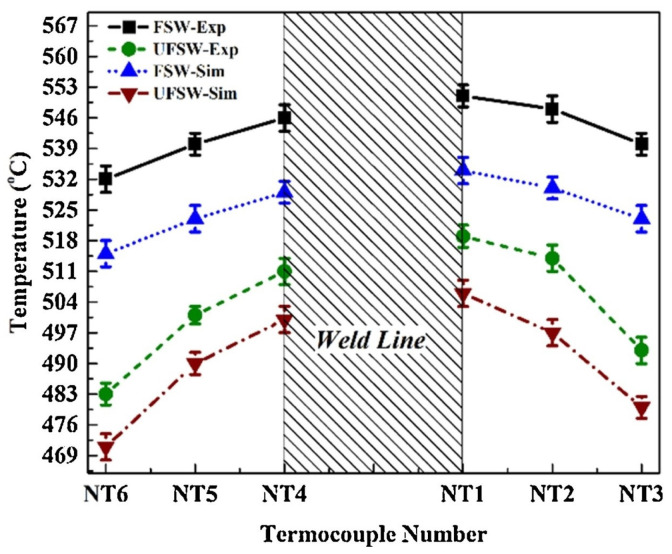
Comparing the maximum temperature values recorded by thermocouples with the results of simulations [130].

**Figure 14 materials-16-05890-f014:**
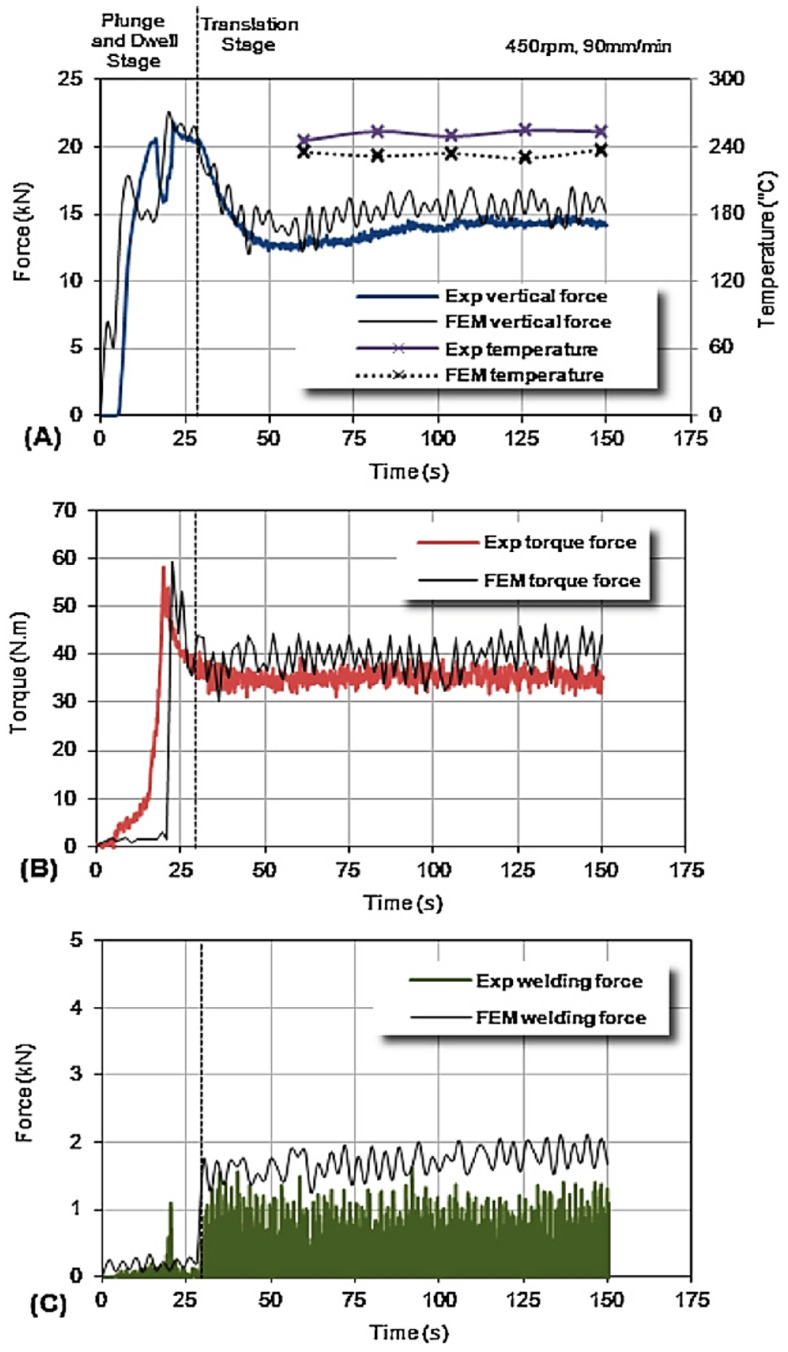
Comparison of experimental and FEM values: (**A**) vertical force and temperature, (**B**) torque and (**C**) welding force [59].

**Figure 15 materials-16-05890-f015:**
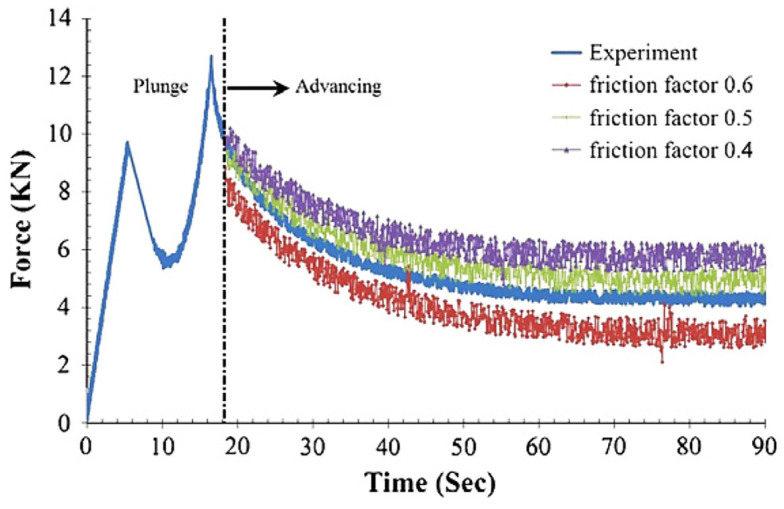
Simulation and experimental axial force [131].

**Figure 16 materials-16-05890-f016:**
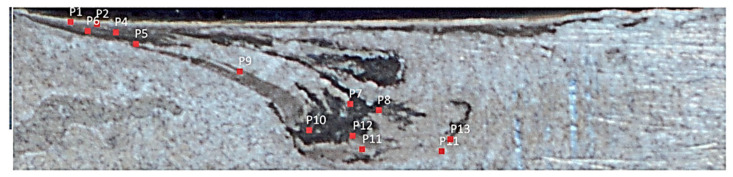
Comparison of simulated and experimental SZ shape. P1–P13 are the points primarily located at the weld centerline but after passing the FSW tool over them, they are distributed at the center and advancing side of the SZ [25].

**Figure 17 materials-16-05890-f017:**
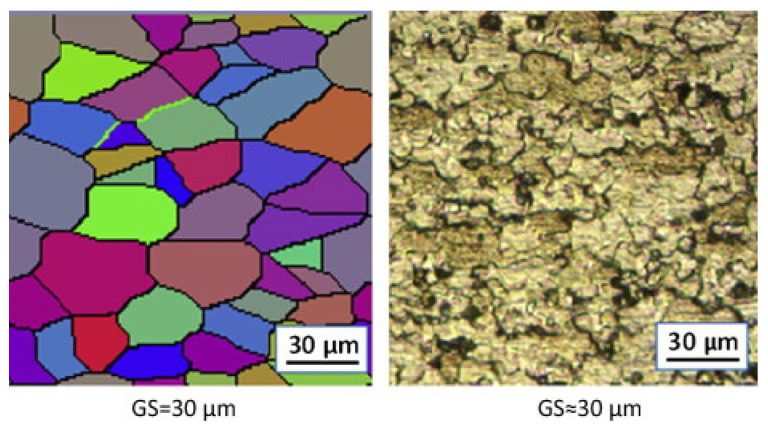
Simulated and experimental microstructure of base metal.

**Figure 18 materials-16-05890-f018:**
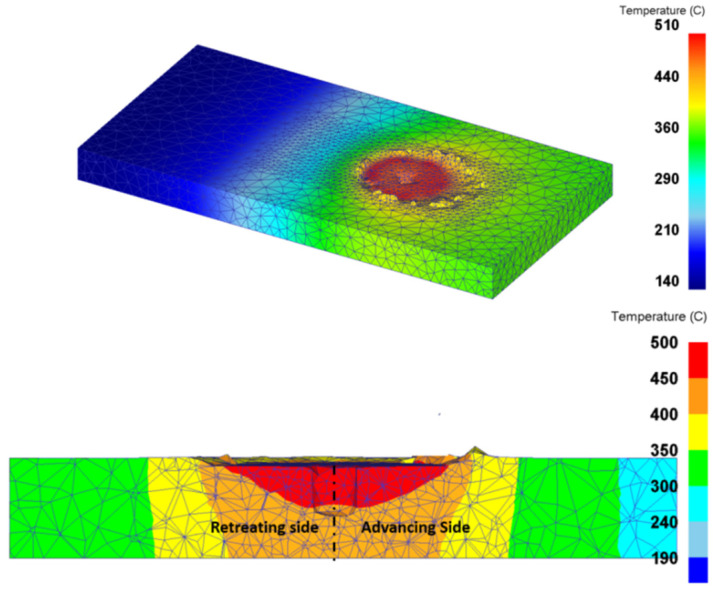
Distribution of temperature in the cross-section of joint the in FSW of aluminum 5083 [38].

**Figure 19 materials-16-05890-f019:**
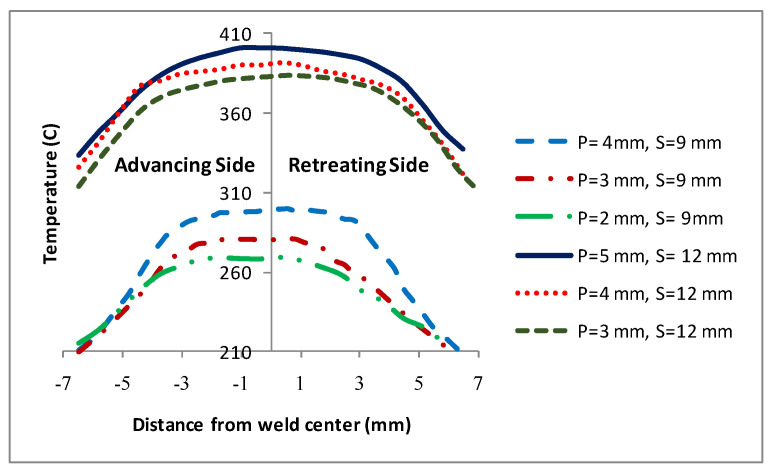
Impact of pin and shoulder diameter on temperature distribution [102].

**Figure 20 materials-16-05890-f020:**
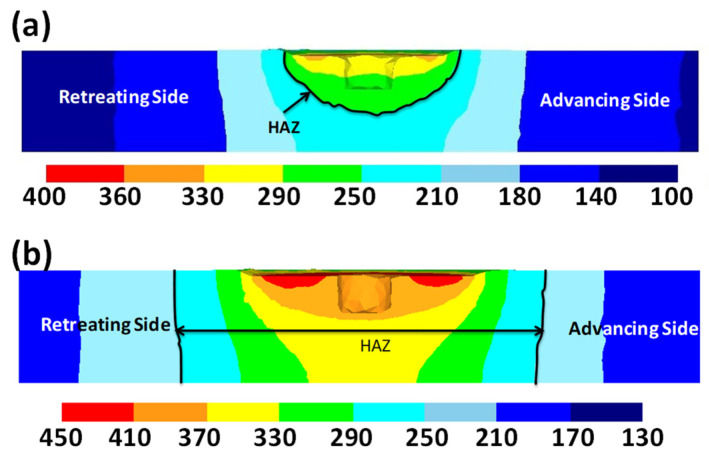
Temperature contours in the weld cross-section. Pin and shoulder diameters: (**a**) 3 and 9 mm, and (**b**) 3 and 12 mm [102].

**Figure 21 materials-16-05890-f021:**
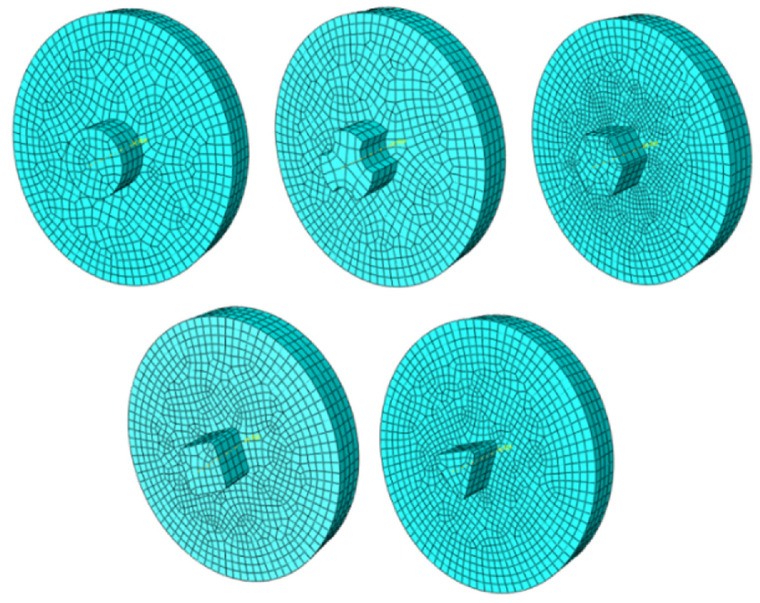
Various pins employed in Akbari et al. investigation [35].

**Figure 22 materials-16-05890-f022:**
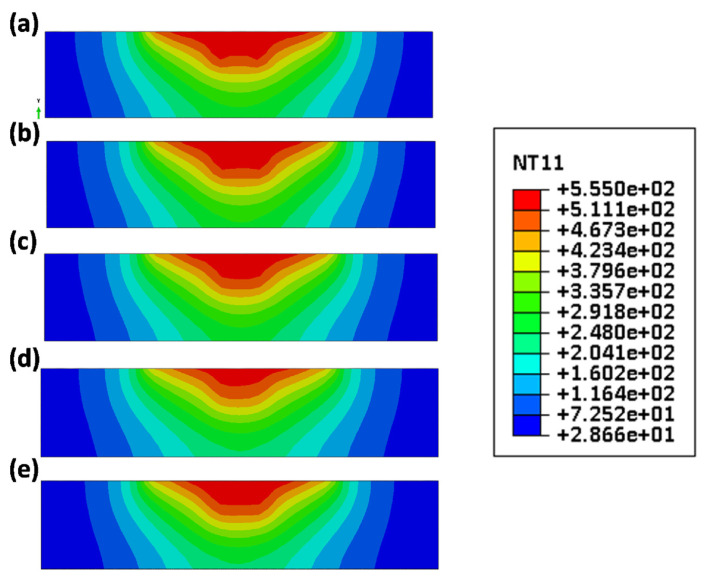
Temperature variation of the cross-section of the weld fabricated with various pin shapes of (**a**) cylindrical, (**b**) hexagonal, (**c**) square, (**d**) triangular, and (**e**) triflate pin profiles [35].

**Figure 23 materials-16-05890-f023:**
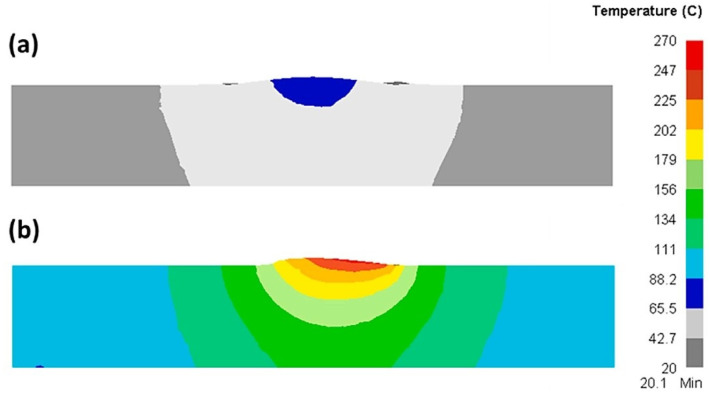
Increase in temperature in the material due to the (**a**) plastic deformation heat and (**b**) frictional heat [108].

**Figure 24 materials-16-05890-f024:**
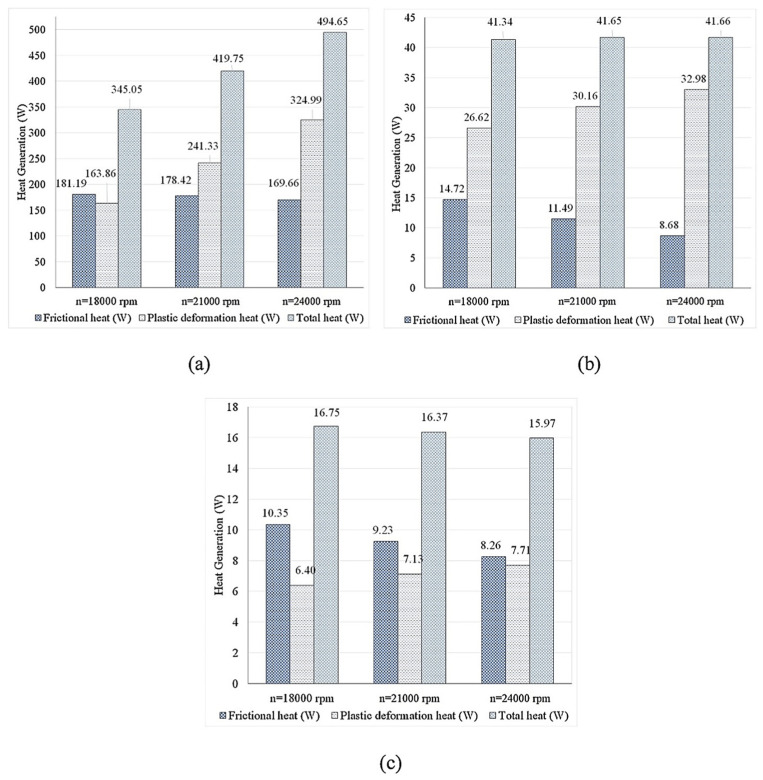
Plastic deformation heat, frictional heat, and total heat generation at different rotational speeds and constant traverse speed of 60 mm/min (**a**) shoulder, (**b**) pin, (**c**) tip [46].

**Figure 25 materials-16-05890-f025:**
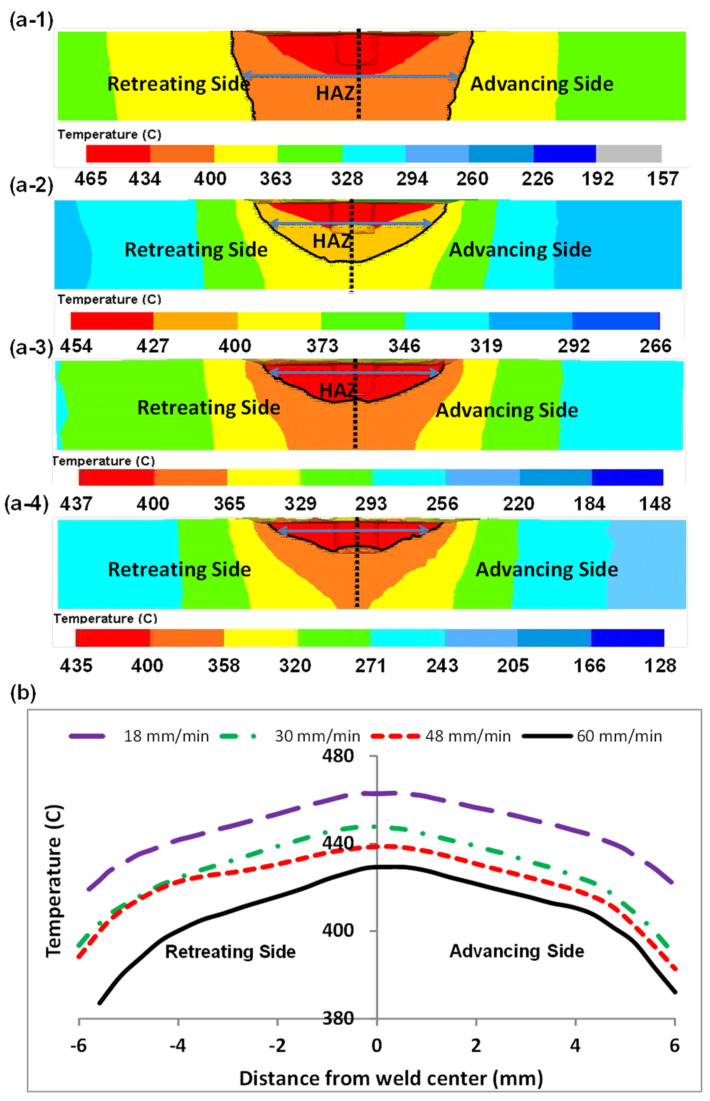
The temperature contours and profiles in the weld cross-section. The rotational speed was 700 rpm, and traverse speeds were (**a-1**) 18, (**a-2**) 30, (**a-3**) 42, and (**a-4**) 60 mm/min. (**b**) Temperature distribution curves in the cross section of welds produced by different process parameters. [7].

**Figure 26 materials-16-05890-f026:**
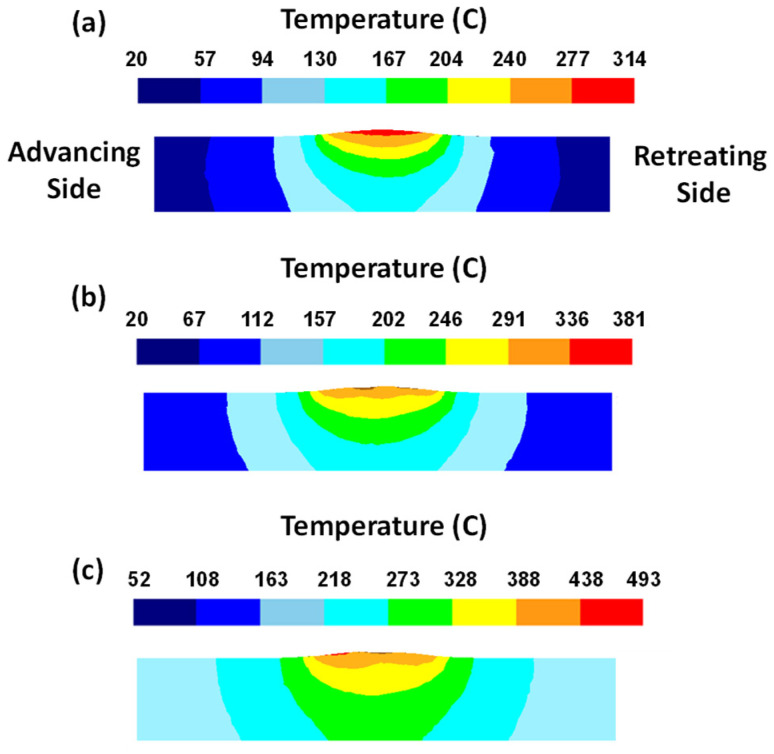
Temperature profiles observed in different cooling conditions: (**a**) water-cooled sample, (**b**) air-cooled sample, (**c**) non-cooled sample [143].

**Figure 27 materials-16-05890-f027:**
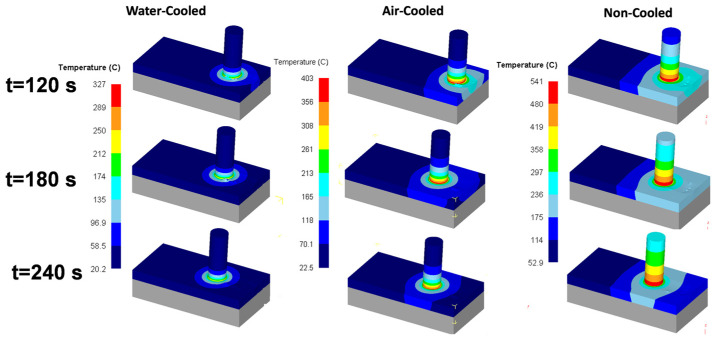
Temperature distribution on the top surface [143].

**Figure 28 materials-16-05890-f028:**
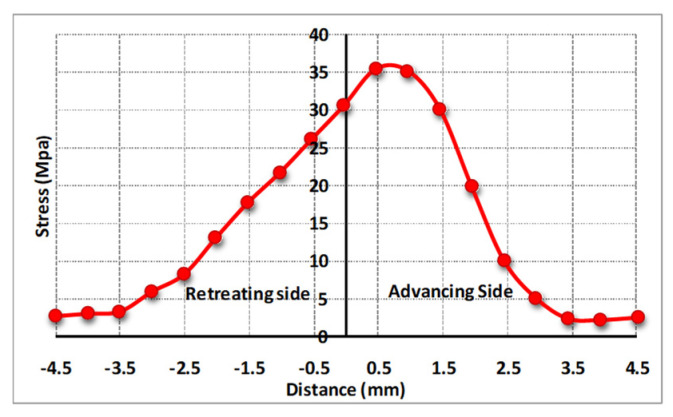
Variation of distributions of effective strain across the cross-section [38].

**Figure 29 materials-16-05890-f029:**
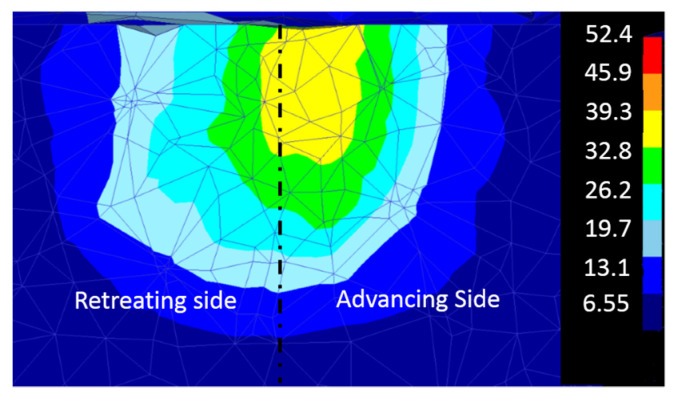
Visualization of strain distribution across the cross-section of FSP samples [38].

**Figure 30 materials-16-05890-f030:**
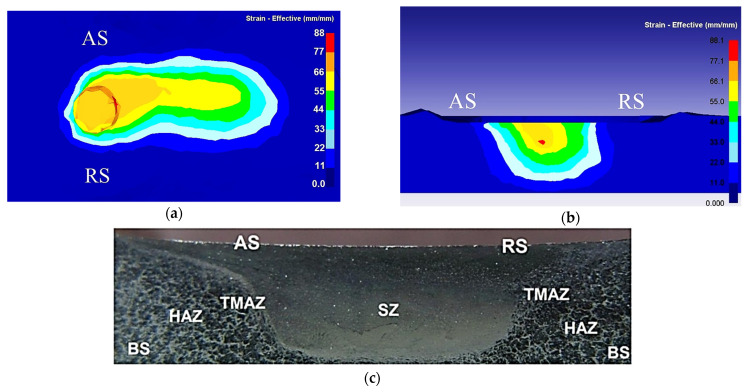
The strain distribution in (**a**) top view and (**b**) cross-section of FSPed sample [109]. (**c**) Cross-section macro-image from FSPed AZ91 magnesium alloy [144].

**Figure 31 materials-16-05890-f031:**
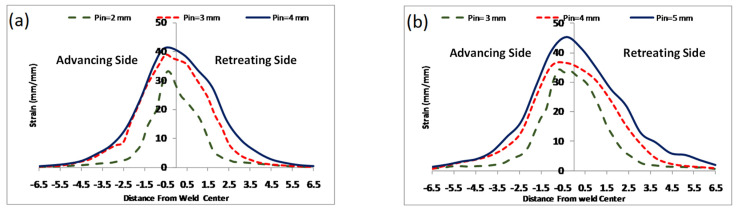
Strain profiles in the transverse section of the weld produced by different tool pin diameters and for shoulder diameters of (**a**) 9 mm and (**b**) 12 mm [102].

**Figure 32 materials-16-05890-f032:**
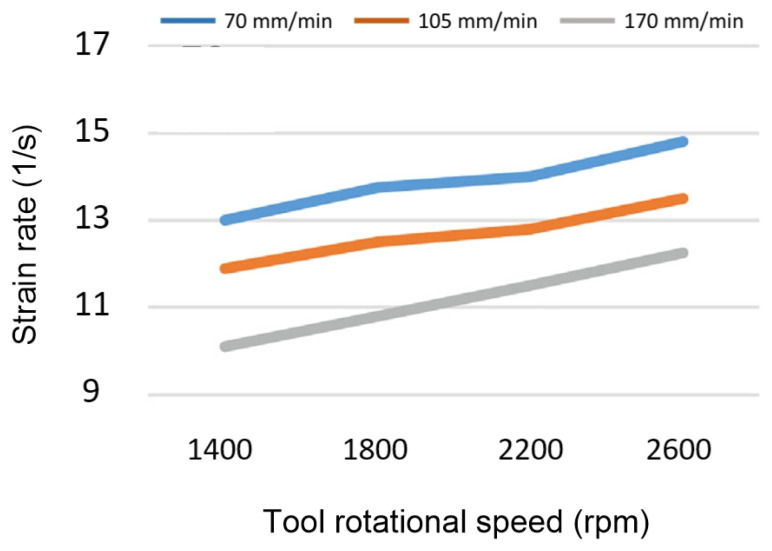
Effect of tool rotational speed on strain rate [145].

**Figure 33 materials-16-05890-f033:**
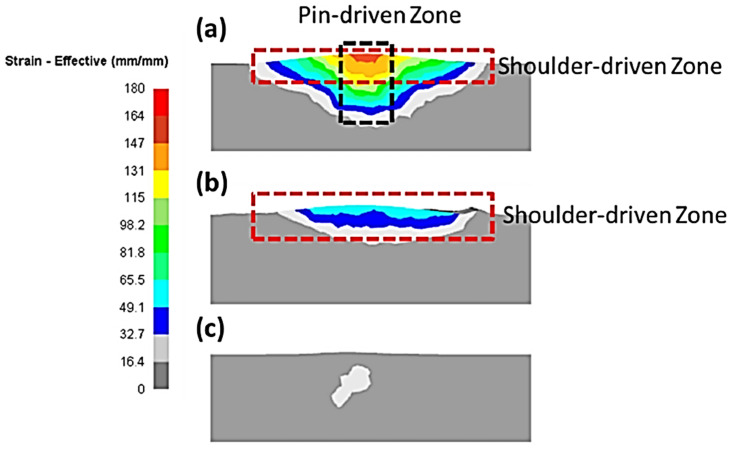
Strain distribution in the cross-section of the workpiece fabricated with (**a**) FSW tool, (**b**) pinless tool, and (**c**) shoulderless tool [108].

**Figure 34 materials-16-05890-f034:**
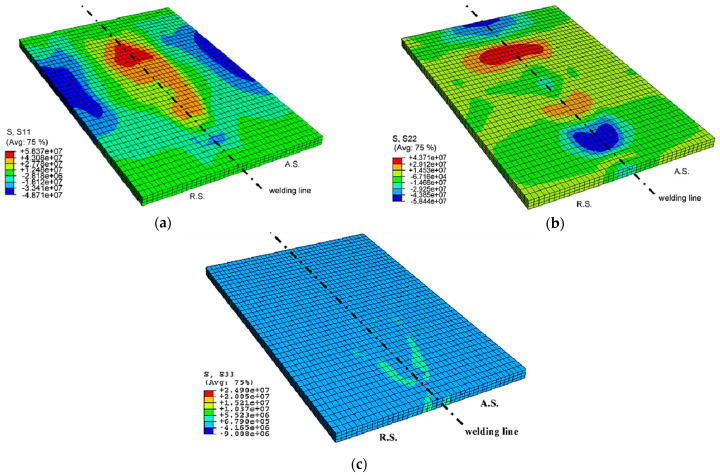
(**a**) Longitudinal residual stresses, (**b**) transverse residual stresses, and (**c**) normal residual stresses. All residual stresses are reported in Pa [62].

**Figure 35 materials-16-05890-f035:**
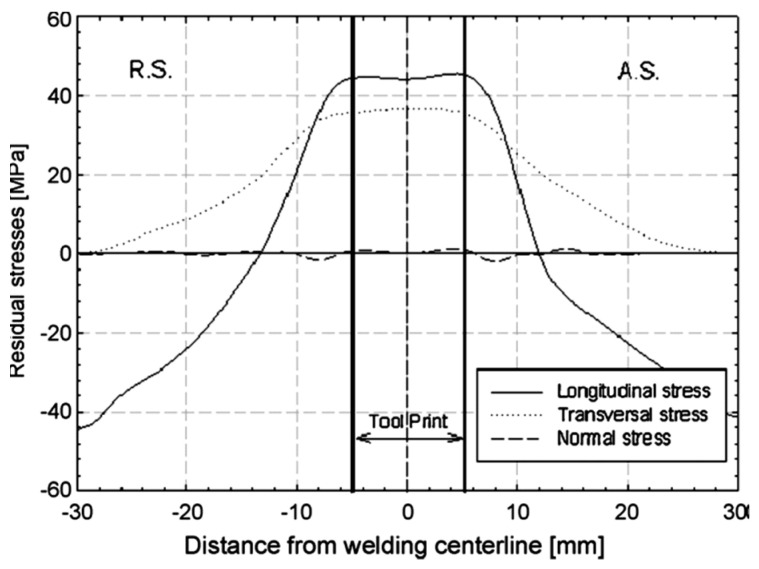
Average residual stresses in the transverse section at the center of the joint; the longitudinal stress exceeds the transverse stress, while the normal stress approaches zero [150].

**Figure 36 materials-16-05890-f036:**
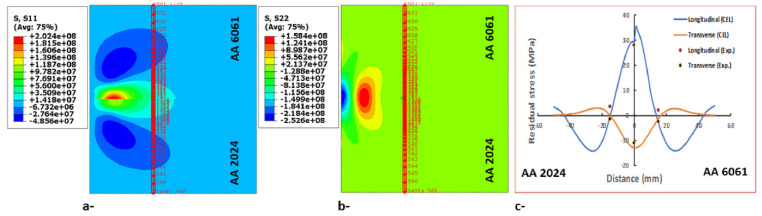
The distribution of residual stress for case A, (**a**) contours in longitudinal direction, (**b**) contours in the transverse direction, and (**c**) experimental and numerical results comparison in the direction shown by the red lines [75].

**Figure 37 materials-16-05890-f037:**
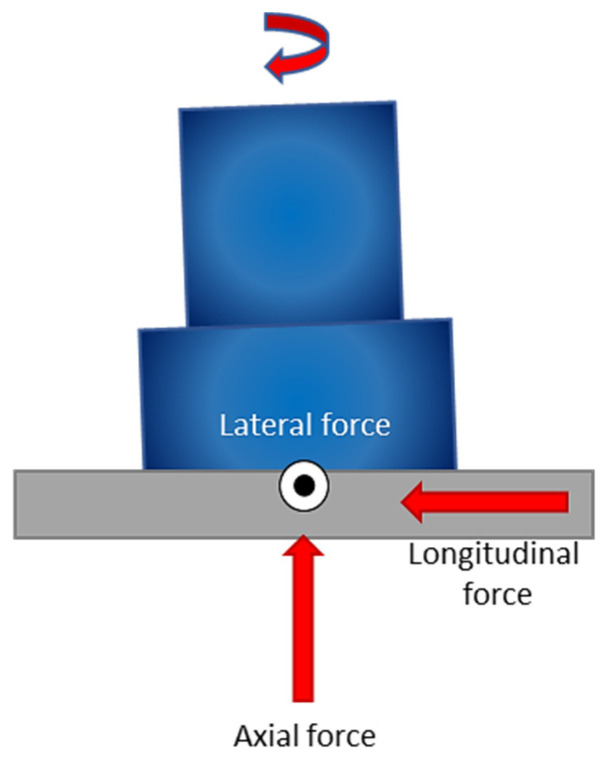
Applied forces on the tool during friction stir welding.

**Figure 38 materials-16-05890-f038:**
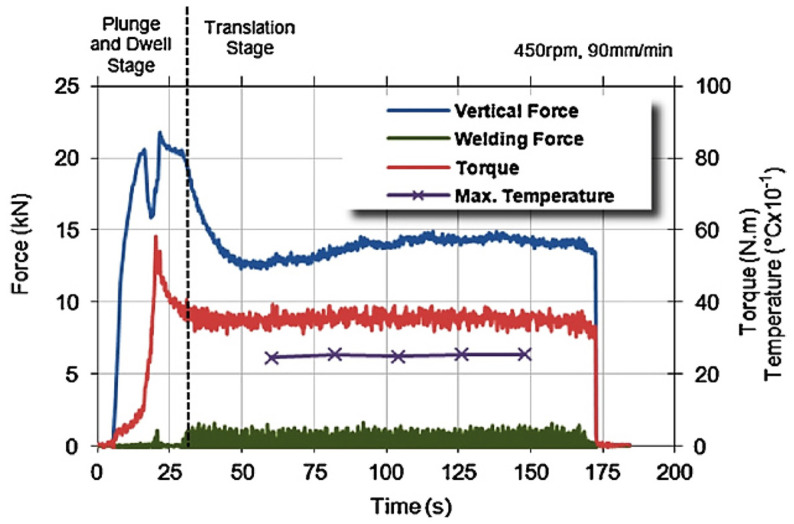
Experimental force and temperate generation [59].

**Figure 39 materials-16-05890-f039:**
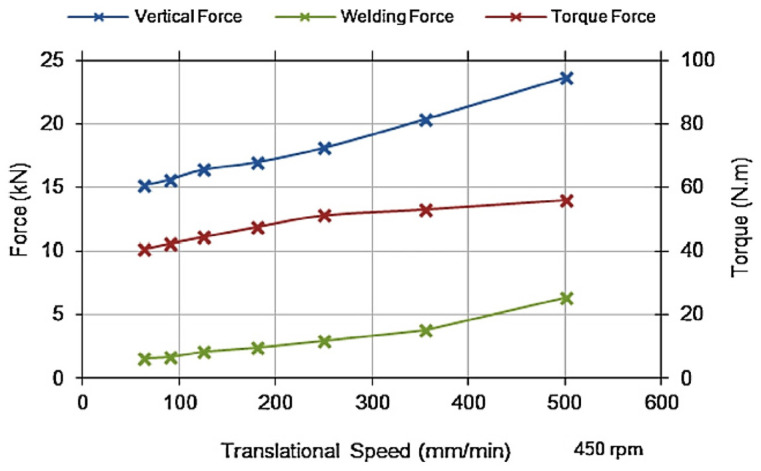
FEM predicted forces for various traverse speeds [59].

**Figure 40 materials-16-05890-f040:**
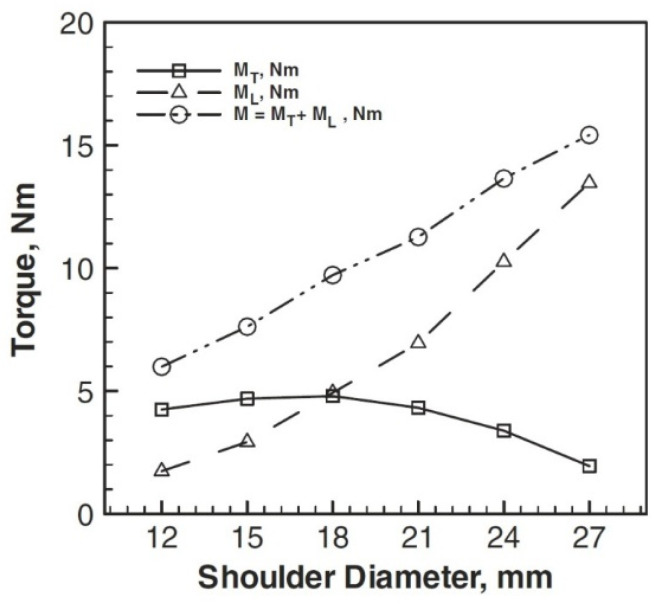
Effect of shoulder diameter on the sliding, sticking, and total torque during FSW [172].

**Figure 41 materials-16-05890-f041:**
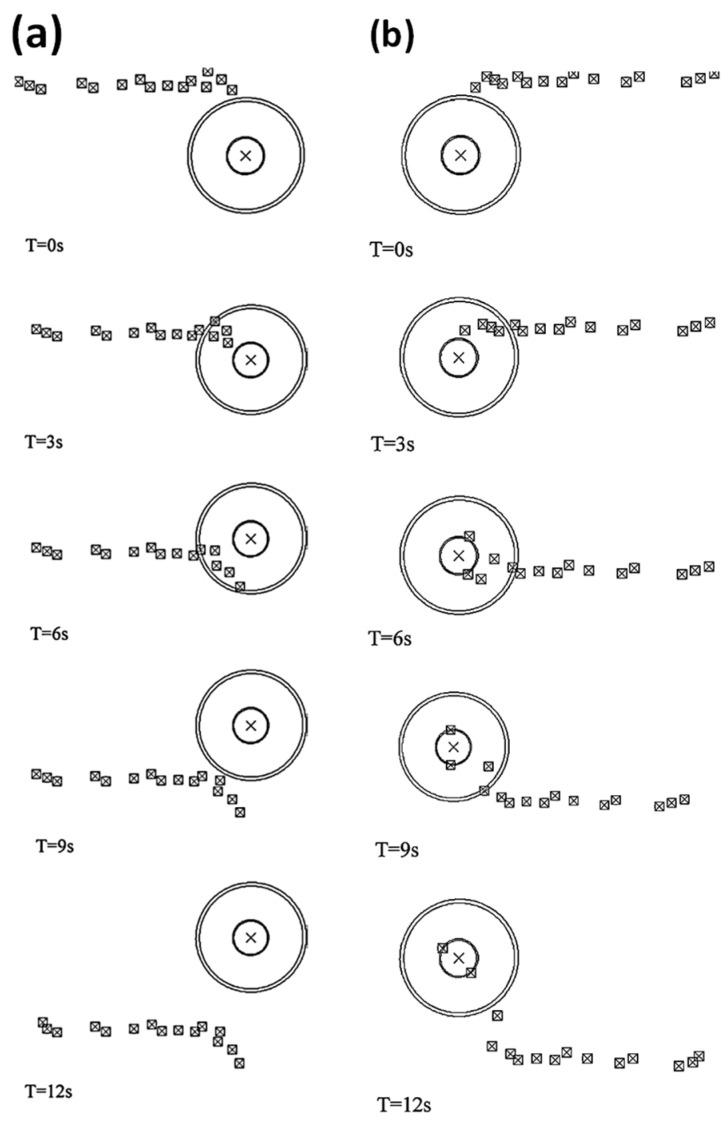
Material flow on the middle surface observed at (**a**) the retreating side and (**b**) the advancing side, with a linear velocity of 2.363 mm/s and a rotational speed of 240 rpm [6].

**Figure 42 materials-16-05890-f042:**
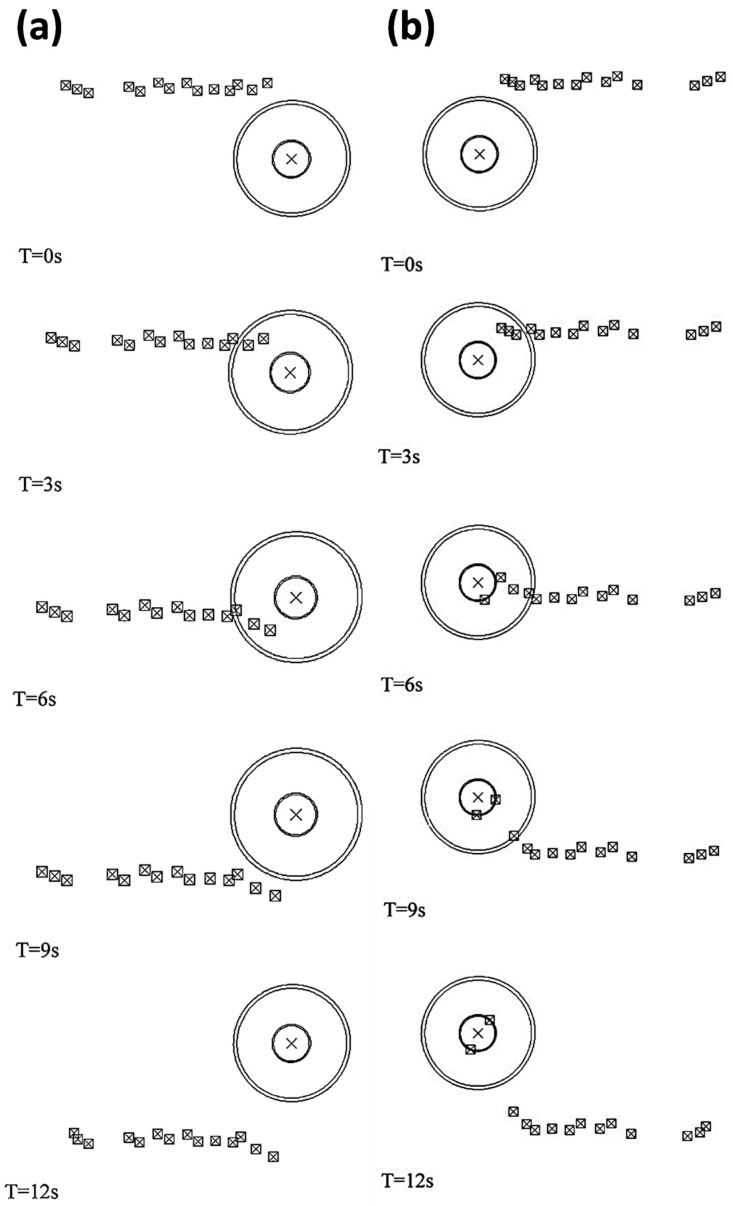
Material flow on the bottom surface observed at (**a**) the retreating side and (**b**) the advancing side, with a linear velocity of 2.363 mm/s and a rotational speed of 240 rpm [6].

**Figure 43 materials-16-05890-f043:**
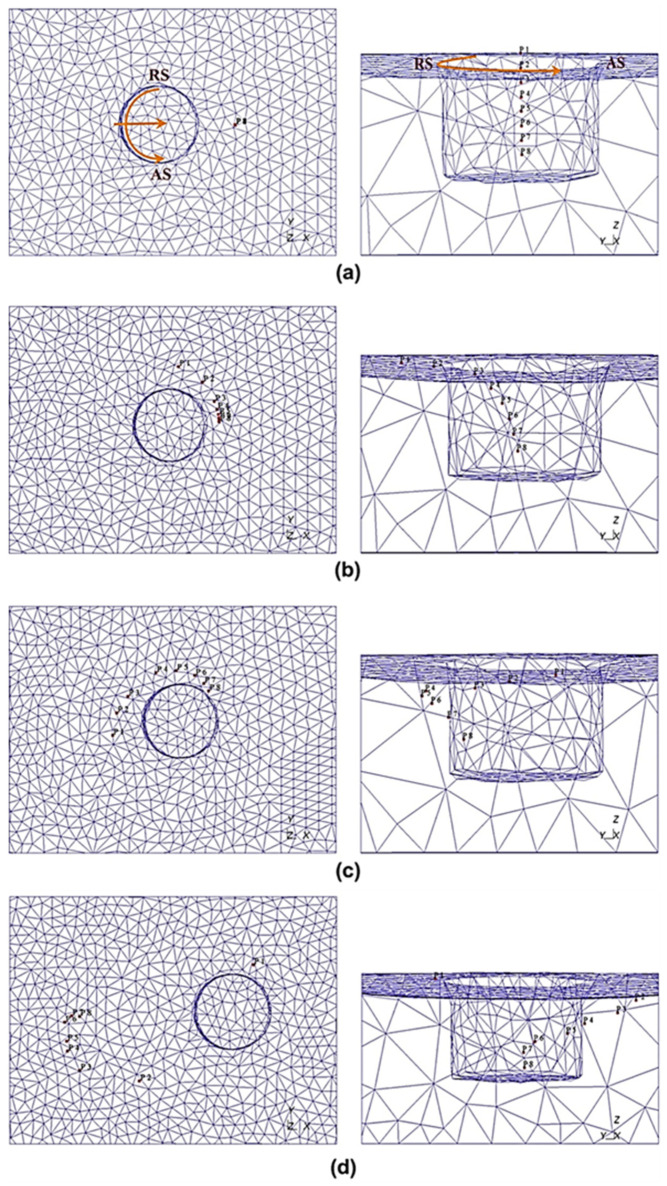
P1–P8 are the centerline points located primarily at the weld centerline to follow the material flow pattern. (**a**–**d**) the progressive material flow by tool traverse during the welding [131].

**Figure 44 materials-16-05890-f044:**
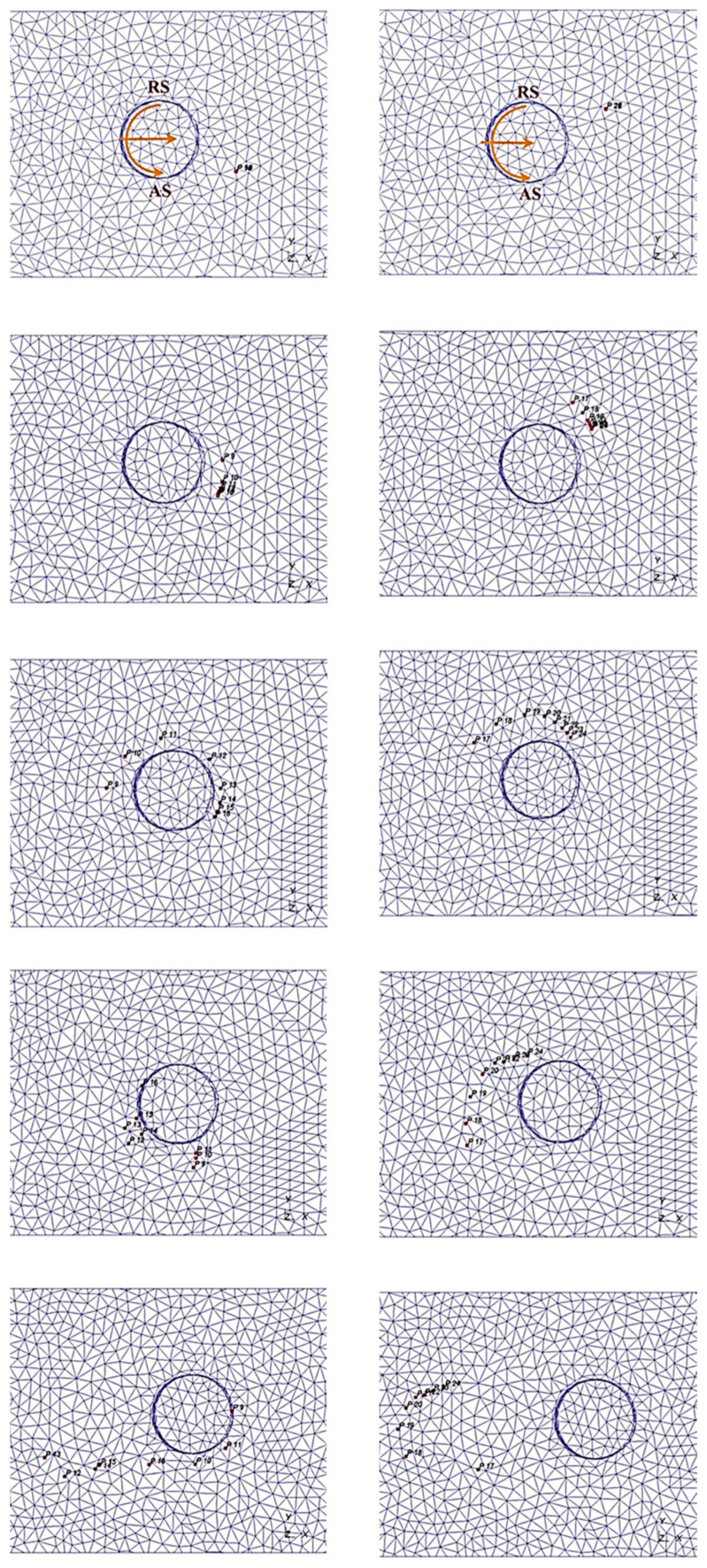
P9–P16 are the points located primarily at the AS to follow the material flow pattern [131].

**Figure 45 materials-16-05890-f045:**
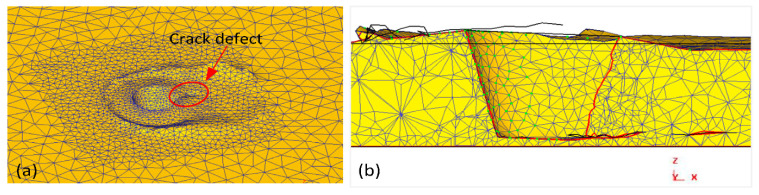
Simulation of FSW along with the defects formation. (**a**) Crack defects and (**b**) hole defects [181].

**Figure 46 materials-16-05890-f046:**
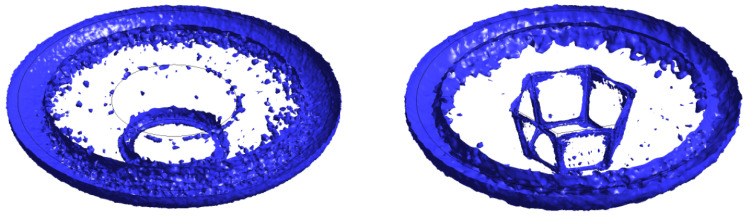
The iso-surface of flow stress is equal to 200 MPa for both FSW−C (**left**) and FSW−H (**right**) samples [182].

**Figure 47 materials-16-05890-f047:**
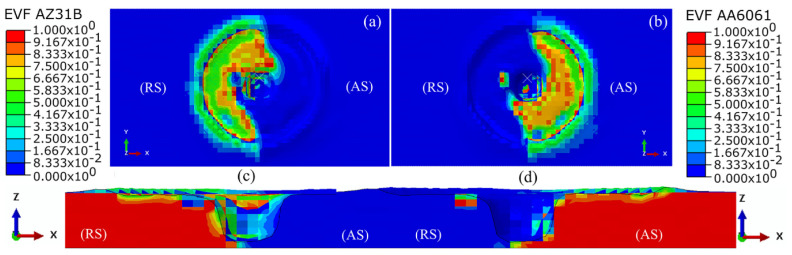
The material flow on the top surface of the stir zone (**a**) retreating side (**b**) advancing side, and the cross-section (**c**) retreating side and (**d**) advancing side.

**Figure 48 materials-16-05890-f048:**
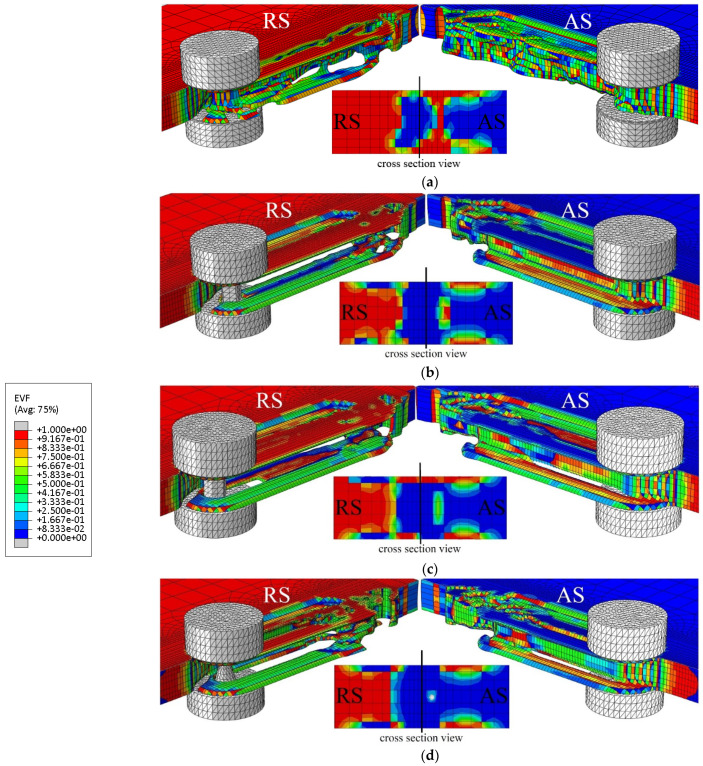
The material movement in the combined cross-section and separated prospective views for the bobbin tool FSW of AZ91 magnesium alloy. (**a**) Trigonal, (**b**) square, (**c**) hexagonal, (**d**) inward conical, and (**e**) outward conical [185].

**Figure 49 materials-16-05890-f049:**
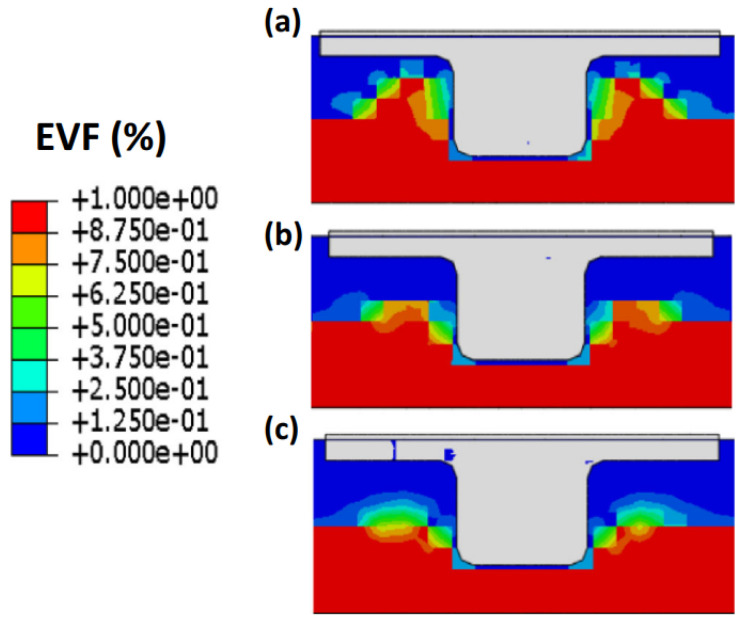
Formation of the SZ in friction stir lap welding of Al–brass at various tool traverse speeds: (**a**) 6.5 mm/min, (**b**) 12.5 mm/min, and (**c**) 25 mm/min [80].

**Figure 50 materials-16-05890-f050:**
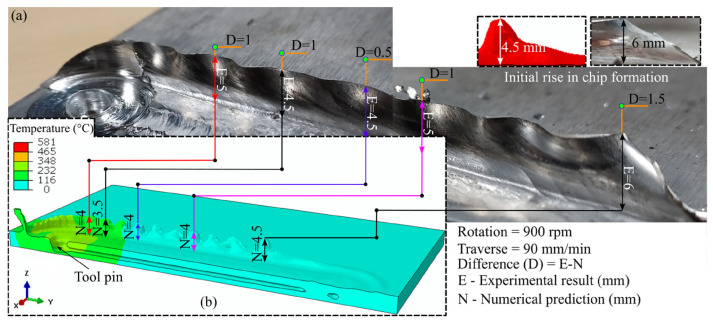
(**a**) The experimental investigation and (**b**) numerical modeling of flash formation on the RS at 900 rpm and 90 mm/min [70].

**Figure 51 materials-16-05890-f051:**
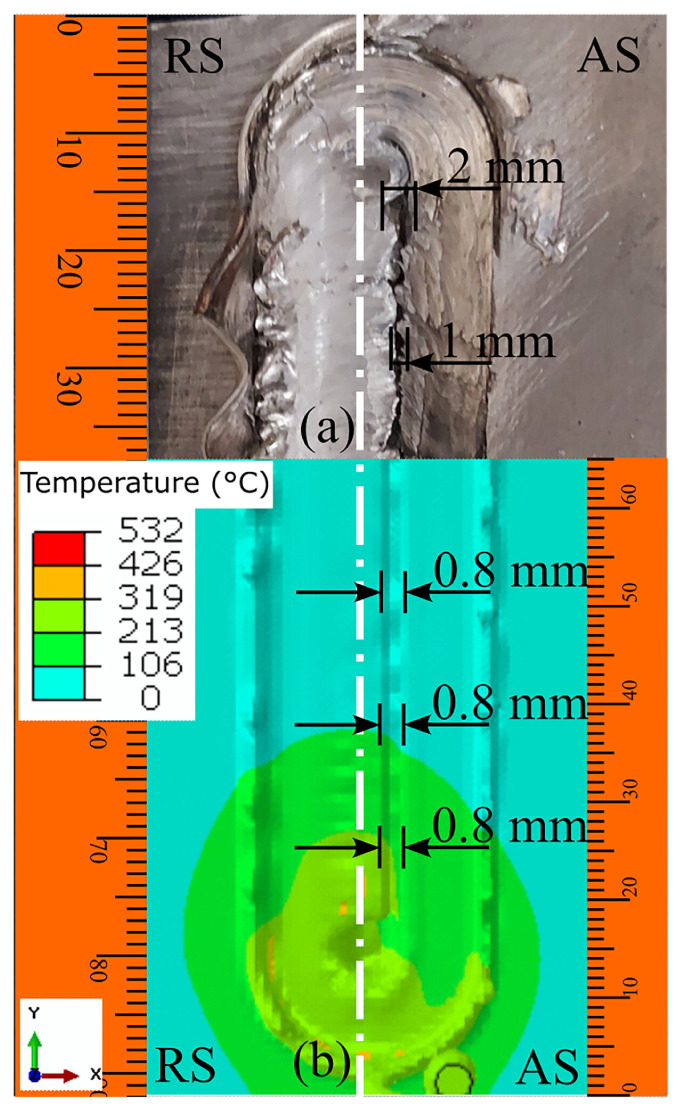
Surface tunnel generated at the SZ during dissimilar AA6061–AZ31B FSW at 1200 rpm and 30 mm/min: (**a**) experimental investigation and (**b**) numerical investigation [70].

**Figure 52 materials-16-05890-f052:**
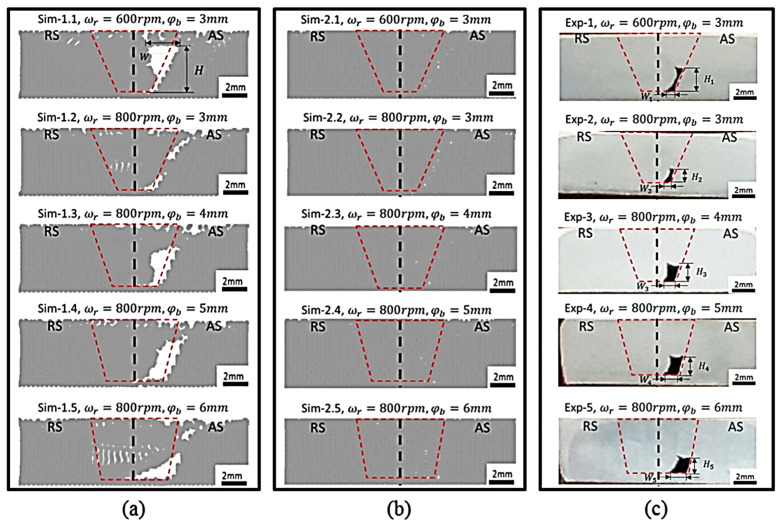
Morphological comparison of the joint area. (**a**,**b**) simulation results and (**c**) experimental results for material flow under 5 sets of process parameters [190].

**Figure 53 materials-16-05890-f053:**
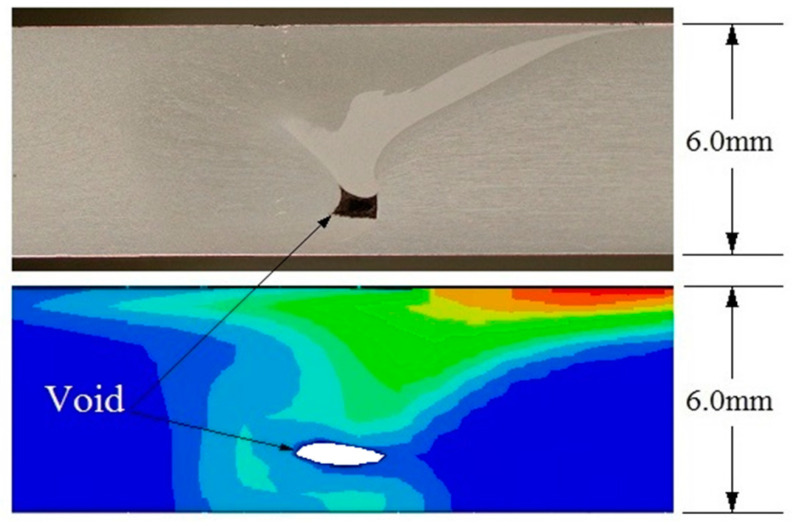
Matching estimated equivalent plastic strain and void with experimentally found [188].

**Figure 54 materials-16-05890-f054:**
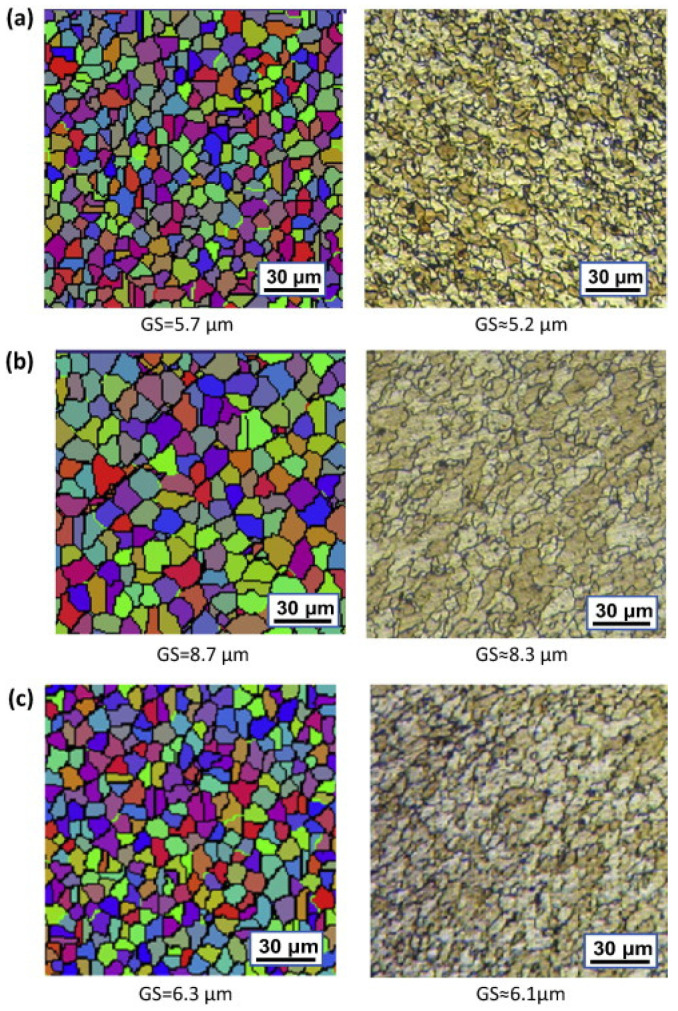
Experimental and simulated microstructure of SZ for traverse speed, rotational speed, and shoulder diameter of (**a**) 120 mm/min, 900 rpm, and 16 mm, (**b**) 120 mm/min, 1120 rpm, and 14 mm (**c**) 80 mm/min, 900 rpm, and 14 mm, respectively.

**Figure 55 materials-16-05890-f055:**
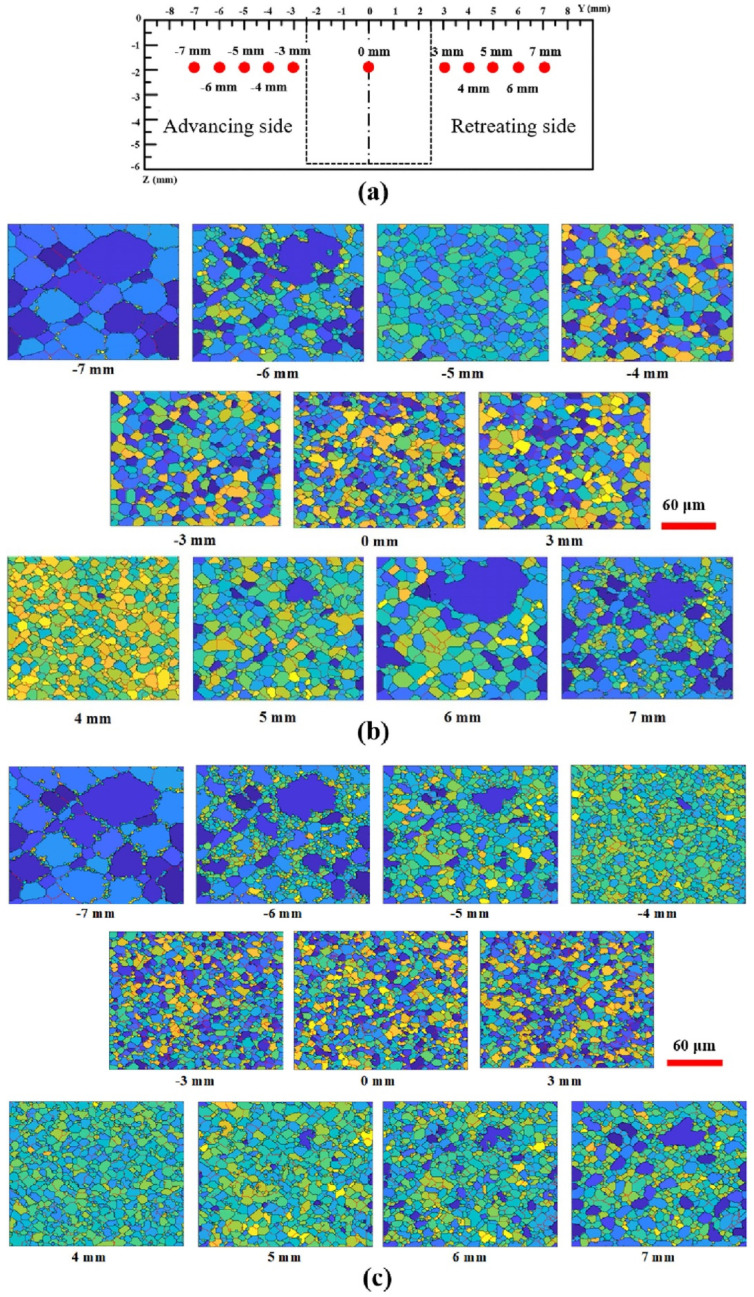
The simulated grain structure in welds. (**a**) Schematic of selected points, (**b**) 800 rpm-100 mm/min, (**c**) 600 rpm-100 mm/min [216].

**Table 1 materials-16-05890-t001:** Comparative analysis showcasing the varying capabilities of different methods in simulation of FSW.

Analysis	Software	Temperature Analysis	Deformation Type	Material Flow	Time	Advantage/Disadvantage
Eulerian	Fluent or Star CCM+	Steady or transient	Viscoplastic	Streamline	Low	Appropriate for metal flow solutions It is challenging to apply this approach in order to accurately represent the deformation of the free borders Inappropriate to evaluate regions with shifting borders
Lagrangian	Forge 3D	Transient	Viscoplastic	Not reported	High	Network and element distortions
	DEFORM- 3D	Transient	Viscoplastic	Point tracking	Moderate	
ALE	ABAQUS/ Explicit	Transient	Elastic-viscoplastic	Point tracking	High	Benefits of both Eulerian and Lagrangian approaches High simulation time
CEL	ABAQUS/ Explicit	Transient	Elastic-viscoplastic	Marker material	High	Benefits of both Eulerian and Lagrangian approaches Greater flexibility in modeling complex systems, such as fluid flow around a moving object The possibility of modeling dissimilar metals
SPH	ABAQUS/ Explicit	Transient	Elastic-viscoplastic	Not reported	Moderate	The lack of a grid and continuum discretization in this approach reduces the likelihood of mesh distortion SPH particles tend to cluster together

**Table 2 materials-16-05890-t002:** Effect of traverse speed and rotational speed on forces and moment.

		Traverse Speed	Rotational Speed
On pin	Axial force	~	+
	Lateral force	~	+
	Longitudinal force	+	−
	Moment of the tool axis	~	−
On shoulder	Axial force	−	+
	Lateral force	~	+
	Longitudinal force	+	−
	Moment of the tool axis	~	−
Total	Axial force	−	+
	Lateral force	~	+
	Longitudinal force	+	−
	Moment of the tool axis	+	−

## Data Availability

No new data were created or analyzed in this study. Data sharing is not applicable to this article.

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
