# Peer review of "A Review on Friction Stir Welding/Processing: Numerical Modeling"

_materials, 2023, doi:10.3390/ma16175890_

Round 1

Reviewer 1 Report

In this paper authors have presented a review of the research activities and advancements made in numerical analysis techniques for friction stir welding and the applicability of these techniques to component manufacturing.

Moreover, system parameters that play a crucial role in the numerical modeling of the friction stir welding and their influence on thermal mechanical behaviour is discussed. 

In my opinion it does not qualify the standards of research studies.

Author Response

Reviewer 1

In this paper authors have presented a review of the research activities and advancements made in numerical analysis techniques for friction stir welding and the applicability of these techniques to component manufacturing.

Moreover, system parameters that play a crucial role in the numerical modeling of friction stir welding and their influence on thermal mechanical behaviour are discussed. 

In my opinion, it does not qualify the standards of research studies.

  • Thanks for your comment. We revised the paper to improve its quality and content, however as you know this paper is a review paper on different numerical models employed for the simulation of friction stir welding technique and has a bit different standard in comparison to the pure research study.

Reviewer 2 Report

The similarity index is 27%, it should be less than 15%, please paraphrase the similar sentences.

The conclusions should be shortened.

Recent references such as "https://doi.org/10.1016/j.jmapro.2022.12.052" and "https://doi.org/10.3390/met13010054" should be cited.

Author Response

Reviewer 2

Comments and Suggestions for Authors

The similarity index is 27%, it should be less than 15%, please paraphrase the similar sentences.

  • I would like to thank you for your meticulous consideration of the manuscript. The text is rewritten and paraphrased again carefully, as the percentage of similarity decreased by more than 5%.
  • However, since this article is a review article. The text of the article is taken from the literature. In this manuscript, an attempt has been made to rewrite the entire text as much as possible, although it is clear from the analysis that even the keywords from the previous articles are detected as similar and have increased the percentage of similarity.

The conclusions should be shortened.

  • The conclusion is summarized accordingly.

Recent references such as "https://doi.org/10.1016/j.jmapro.2022.12.052" and "https://doi.org/10.3390/met13010054" should be cited.

  • The introduced articles are cited in the revised version.

Reviewer 3 Report

The authors submitted a Review article dealing with the numerical simulation of Friction Stir Welding. Although the subject of this manuscript is relevant and thus there are many review articles published on this topic, the authors' contribution was not so evident to the reader compared to existing article reviews in the literature. (Lack of novelty)

e.g., 

1) A review on numerical modelling techniques in friction stir processing:

current and future perspective https://doi.org/10.1007/s43452-023-00688-6

2) A Comparison of Different Finite Element Methods in the Thermal Analysis of Friction Stir Welding http://dx.doi.org/10.3390/met7100450

3) Experimental and numerical analysis of friction stir welding: a review https://doi.org/10.1088/2631-8695/ac7f1e

4) A review of numerical analysis of friction stir welding (https://doi.org/10.1016/j.pmatsci.2014.03.003)

...

The authors did not mention this literature review!

Additionally, this manuscript presents some flaws, where the authors are practically reciting the literature without a thorough discussion or comments on findings reported in the literature. 

This manuscript claims to focus on FSW modelling, thus, the paragraph about FSP is not necessary to include, it should be removed [Line 83-101].

The introduction is too long, it presents typical knowledge for experts in FSW! It does not provide new knowledge. The Introduction section should be thoroughly revised and the subsection 1.2 process parameters is to be removed.

Despite the reference list contains 230 references; however, the majority of this list is not up-to-date. Moreover, the reference list is disorganized and it does not correspond to the references cited in the manuscript.

Consequently, a thorough revision should be achieved to improve the quality and presentation of this review manuscript.  

Moderate Editing of English should be addressed 

Author Response

Reviewer 3

The authors submitted a Review article dealing with the numerical simulation of Friction Stir Welding. Although the subject of this manuscript is relevant and thus there are many review articles published on this topic, the authors' contribution was not so evident to the reader compared to existing article reviews in the literature. (Lack of novelty)

  • In the present review manuscript, in contrast to other review papers, all the simulation models used for FSW/FSP including the CFD ALE, SPH, and CEL methods are considered in depth.
  • The applications of the model results for the prediction of produced samples’ properties are discussed.
  • The material flow for similar and especially dissimilar joints using different numerical models is investigated.
  • The new tools used in FSW such as bobbin and stationary tools are considered in numerical models.
  • The microstructure evolution over the FSW/FSP process is discussed while the other review papers do not include this topic.

1) A review on numerical modelling techniques in friction stir processing: current and future perspective https://doi.org/10.1007/s43452-023-00688-6

  • This paper focuses on the FSP and distribution of reinforcing particles and has not considered many affecting factors such as process parameters and tool design on the welded joint properties.

2) A Comparison of Different Finite Element Methods in the Thermal Analysis of Friction Stir Welding http://dx.doi.org/10.3390/met7100450

  • This article, published in 2017, has only examined some methods used in process modeling and has not examined or compared their results carefully. In addition, the material flow during the process or microstructure modeling is not included.

3) Experimental and numerical analysis of friction stir welding: a review https://doi.org/10.1088/2631-8695/ac7f1e

  • This article, which contains almost 8 pages, just covers some limited issues in the numerical modeling of process and is not a comprehensive review article.

4) A review of numerical analysis of friction stir welding (https://doi.org/10.1016/j.pmatsci.2014.03.003)

  • This article is one of the best review articles written in the field of numerical methods to model the friction stir welding process. However, it is written in 2014 and does not include the research papers published in recent ten years. As we know, several high-quality simulation models are developed in this period and it is time to write a new review article on this topic. In the present review manuscript, almost 140 papers of reviewed references are published after 2014.

Additionally, this manuscript presents some flaws, where the authors are practically reciting the literature without a thorough discussion or comments on findings reported in the literature. 

  • The manuscript is revised and we tried to improve the discussions of findings reported in the literature.

This manuscript claims to focus on FSW modelling, thus, the paragraph about FSP is not necessary to include, it should be removed [Line 83-101].

  • Thank you for your comment, but the authors did not claim the inclusion of FSW and the exclusion of FSP. It is evident that the numerical models used for FSP and FSW are almost the same, so we have chosen to cover all models in this review paper.
  • Therefore, the title of the manuscript is modified and “processing” is also added.

The introduction is too long, it presents typical knowledge for experts in FSW! It does not provide new knowledge. The Introduction section should be thoroughly revised and the subsection 1.2 process parameters is to be removed.

  • That is right. The introduction section is revised accordingly and the subsection 1.2 process parameters is removed.

Despite the reference list contains 230 references; however, the majority of this list is not up-to-date. Moreover, the reference list is disorganized and it does not correspond to the references cited in the manuscript.

  • About 87 of the used references (38%) are related to after 2018. In this article, all new research is tried to be used.
  • The order of references was checked again.
  • Also, some new references that have been published recently were added to the article.

Reviewer 4 Report

The review provides a comprehensive overview of the various numerical modelling approaches applied to different aspects of FSW, from heat transfer to microstructure evolution. After careful evaluation, I have the following comments to help improve your work.

1- What are the main challenges in experimentally measuring strain, stress, and temperature distributions during FSW due to the severe plastic deformation in the weld zone? How do numerical models help address these challenges?

2- How do the different numerical modelling techniques like CFD, ALE, CEL, and SPH compare in their ability to handle large deformations and material flow during FSW simulation? What are their relative strengths and weaknesses?

3- What are the key differences between CDRX and GDRX models for grain evolution during FSW? Which one provides a more physics-based understanding of the underlying mechanisms?

4- How can factors like tool geometry, process parameters, and cooling conditions be optimised through numerical modelling to improve FSW quality and minimise defects?

5- Validation of the numerical models against experimental data is mentioned briefly, but more details on the validation procedures and quantitative metrics used to evaluate the model accuracy would strengthen the review. Were parameters tuned to match experiments, or were the models predictive?

6- Additional discussion comparing the computational expense and accuracy between different modelling methods would further enrich the review.

7- The section on residual stresses could be expanded with more examples of how process parameters affect the residual stress distributions. Residual stresses are identified as a critical output of models, but their quantification methodologies are not elaborated on. What techniques are commonly used to measure residual stresses in FSW joints? How are residual stress predictions from models verified?

8- The review discusses different modelling approaches like CFD, ALE, CEL, and SPH but does not provide much detail on the governing equations or mathematical models used in each one. Could the authors provide more information on the specific equations, assumptions, simplifications, etc., used in the models referenced?

9- Microstructural modelling is covered, but the validation strategies for grain size, precipitate distributions, etc., are not discussed. How are microstructural modelling predictions confirmed against real microstructures?

10- The review mentions defects like tunnelling and voids but does not detail how these defects are predicted and quantified in models. Can the authors expand on specific defect prediction methodologies?

11- Tool wear during welding can affect process forces, heat generation, and defects over time. Is tool wear considered in the models discussed? If so, how is the wear progression modelled?

12- How sensitive are model outputs, like temperature, residual stress, grain size etc., to inaccuracies in input parameters like friction, material properties, boundary conditions etc? Are sensitivity analyses conducted?

13- Are coupled multi-physics models required to capture all relevant physics, or can sequential modelling approaches provide sufficient accuracy? How do computation times compare?

14- Has model order reduction been applied to develop lower-order models capturing the essential physics but at reduced computational cost?

15- The manuscript could benefit from proofreading to avoid grammatical or typos:

15-1- In the abstract, "Over the FSW, the revolving tool..." should be "During FSW, the revolving tool..."

15-2- In the introduction, " Boost the expectation of supply by the researchers" is awkwardly phrased. Consider rephrasing.

15-3- In the grain evolution modelling section, "low and medium Stacking Fault Energy (SFE) materials such as 304 stainless steel typically exhibit Discontinuous Dynamic Recrystallization (DDRX)" should say "materials with low and medium Stacking Fault Energy (SFE) such as 304 stainless steel typically exhibit Discontinuous Dynamic Recrystallization (DDRX)"

15-4- In the abstract, "numerical methods are used to investigate these parameters and better understanding the process" sounds incorrect. Should it be "numerical methods are used to investigate these parameters and better understand the process"?

15-5- In the temperature distribution section, "As shown in Figure 15, as the diameter of the tool pin increases, the temperature of the stir zone increases" sounds incorrect. Should "Figure 15" be "Figure 14"?

Author Response

Dear Reviewer,

Thank you very much for your valuable suggestions.

The review provides a comprehensive overview of the various numerical modelling approaches applied to different aspects of FSW, from heat transfer to microstructure evolution. After careful evaluation, I have the following comments to help improve your work.

  • I would like to thank you for your positive perspective on this review paper. Your constructive comments

1- What are the main challenges in experimentally measuring strain, stress, and temperature distributions during FSW due to the severe plastic deformation in the weld zone? How do numerical models help address these challenges?

  • Indeed the experimental measuring of the strain, stress and temperature is very challenging and most of the times is not possible. For instance to measure the temperature (which is easier than the other two parameters) in the weld nugget zone a sensor must be inserted in this zone while the FSW rotating tool passes through this position and destroyes the sensor before detecting completely the temperature history. So, some limited experiments are done in this regard and the temperature history is normally measured in a position almost 5-20 mm far from the weld center. These locations do not reflect the nugget zone temperature however the measurments can assist to predict the temperature in the nugget zone. More crucial issues are available for the stress and strain distributions. However the simulation models can be verified using these local and insufficient experimental mesearments and then predict all the required data including the force and torque, and temperature, stress and strain histories and distributions.

2- How do the different numerical modelling techniques like CFD, ALE, CEL, and SPH compare in their ability to handle large deformations and material flow during FSW simulation? What are their relative strengths and weaknesses?

  • In each section related to the different numerical modelling techniques like CFD, ALE, CEL, and SPH, the pros and cons of that technique is described and highlited in the text

3- What are the key differences between CDRX and GDRX models for grain evolution during FSW? Which one provides a more physics-based understanding of the underlying mechanisms?

  • The deifinitions paragraphs in the section of “10.3. Grain evolution (GE) modeling” is comprehensively revised and rewrriten for more clarification.
  • In the FSW of high SFE alloys such as aluminum alloy both CDRX and GDRX are possible for occurrence. Because the straninig and material flow from the directional point of view is very complicated and depends on the tool design as well; although the flow of material is mostly directed from the leading edge to the trailing edge of the tool through the retreating side. We tried to find some referenced in the literature considering whether the CDRX or GDRX is dominant over the FSW process however nothing speciallized is found and I think that is a good topic for ourselfs future research.

4- How can factors like tool geometry, process parameters, and cooling conditions be optimised through numerical modelling to improve FSW quality and minimise defects?

  • A section with the title of “11. Optimization of FSW based on residual stress modelling” is added to the end of this review manuscript.

5- Validation of the numerical models against experimental data is mentioned briefly, but more details on the validation procedures and quantitative metrics used to evaluate the model accuracy would strengthen the review. Were parameters tuned to match experiments, or were the models predictive?

  • The validations of numerical models by experimental results are described for each numerical model type including the validations using temperature, force, torque, strain and other experimental results.
  • Indeed depending on the the model type some parameters such as firction coefficient is modified to match the experiments such as force and temperature.

6- Additional discussion comparing the computational expense and accuracy between different modelling methods would further enrich the review.

  • Comparing the results and accuracy of different numerical models is very difficult and exactly is impossible due to the lack of information provided by the related papers such as mesh size, run time, the computer configuration, etc. on the other hand some models have a capability while the other does not, e.g. material flow or point tracking, etc.
  • However, Table 1 for comparative analysis showcasing the varying capabilities of different methods is added to the text.

7- The section on residual stresses could be expanded with more examples of how process parameters affect the residual stress distributions. Residual stresses are identified as a critical output of models, but their quantification methodologies are not elaborated on. What techniques are commonly used to measure residual stresses in FSW joints? How are residual stress predictions from models verified?

  • More examples of residual stress prediction by numerical models are reviewed, added to the text and highlighted.
  • A paragraph for residual stress validation methods used in FSW is added to the text.

8- The review discusses different modelling approaches like CFD, ALE, CEL, and SPH but does not provide much detail on the governing equations or mathematical models used in each one. Could the authors provide more information on the specific equations, assumptions, simplifications, etc., used in the models referenced?

  • Thanks for the suggestion, the governing equtions for each type of models are provided and highlighted in the text and some others are referenced for deep considerations.

9- Microstructural modelling is covered, but the validation strategies for grain size, precipitate distributions, etc., are not discussed. How are microstructural modelling predictions confirmed against real microstructures?

  • The microstructure validation methods are added accordingly.

10- The review mentions defects like tunnelling and voids but does not detail how these defects are predicted and quantified in models. Can the authors expand on specific defect prediction methodologies?

  • The defects prediction methods are added to the text accordingly.

11- Tool wear during welding can affect process forces, heat generation, and defects over time. Is tool wear considered in the models discussed? If so, how is the wear progression modelled?

  • The effects of tool wear on the welded sample defects are not considered by numerical models and the tool wear is considered itself using simulation techniques.
  • Tool wear is a very interesting topic and consists of several techniques and observations. Indeed the topic is for our next review paper and due to the over-lengthening of this review paper we prefere to cover this topic in a separate review.

12- How sensitive are model outputs, like temperature, residual stress, grain size etc., to inaccuracies in input parameters like friction, material properties, boundary conditions etc? Are sensitivity analyses conducted?

  • In the different sections which present different numerical models, the sensitive analysis is presented separately for each model.

13- Are coupled multi-physics models required to capture all relevant physics, or can sequential modelling approaches provide sufficient accuracy? How do computation times compare?

  • In terms of accuracy, almost all models are verified and have acceptable accuracy. However to compare the defferent models’ results to each other is a bit complicated topic since there is no research article comparing the achieved results by different methods in a single paper. The results of different papers written by different research group cannot be compared from the accuracy point of view since there are several affecting parameters from computer configuration, the ability of researchers in controlling the model and simulation, upto the mesh size and computation times.
  • The computation time is just presented by very limited works and they are not comparable due to the afromentioned reseans.

14- Has model order reduction been applied to develop lower-order models capturing the essential physics but at reduced computational cost?

  • Indeed the low order models were presented around 2000-2005 and are not applicable these days.
  • Your suggestion is very good and worthy and can be used industrially, but it has not been noticed by researchers for research articles in recent years.

Comments on the Quality of English Language

15- The manuscript could benefit from proofreading to avoid grammatical or typos:

15-1- In the abstract, "Over the FSW, the revolving tool..." should be "During FSW, the revolving tool..."

  • The text is revised and controlled grammatically once more.

15-2- In the introduction, " Boost the expectation of supply by the researchers" is awkwardly phrased. Consider rephrasing.

  • The sentence is revised.

15-3- In the grain evolution modelling section, "low and medium Stacking Fault Energy (SFE) materials such as 304 stainless steel typically exhibit Discontinuous Dynamic Recrystallization (DDRX)" should say "materials with low and medium Stacking Fault Energy (SFE) such as 304 stainless steel typically exhibit Discontinuous Dynamic Recrystallization (DDRX)"

  • The sentence is revised.

15-4- In the abstract, "numerical methods are used to investigate these parameters and better understanding the process" sounds incorrect. Should it be "numerical methods are used to investigate these parameters and better understand the process"?

  • The sentence is revised accordingly.

15-5- In the temperature distribution section, "As shown in Figure 15, as the diameter of the tool pin increases, the temperature of the stir zone increases" sounds incorrect. Should "Figure 15" be "Figure 14"?

  • The sentence is revised accordingly and the figures’ number is checked.

Round 2

Reviewer 1 Report

As it is a review paper, so a thorough discussion on the existed literature is desired in terms of comments and reasons and direction of improvement. The present form is somewhat acceptable as, authors have improved the introduction section, defect prediction methods and the references are updated. Keep in view the fact that it is a review article, it should be critical review article. Anyways, it can be accepted in the present form now.

English language is fine and acceptable.

Author Response

Dear Reviewer,

Thank you very much for your comments and final acceptance of our manuscript.

Tomasz Sadowski

Reviewer 3 Report

The authors have by far improved their manuscript compared to the initial submission.  

However, there are still uncorrected points that should be addressed in the manuscript to be suitable for publication.

1) Check to define all the variables in the added equations

2) Check the typos error in line 673

3) In Figure 16 (page 21): The comparison between simulated and experimental SZ shapes is not visible!! (i.e., there are no simulation results highlighted!)

4) The quality of some figures needs to be improved (e.g., Fig.14, Fig.15

4) The authors missed to include this recent work on the effect of friction models on the FSW results (DOI: 10.1177/14644207211070965)  and also a paper on the comparative study between the FSW modelling using ALE and CEL formulations (DOI: 10.1007/978-3-030-86446-0_36)

5) Refs 159 and 196 are not visible (written in other languages than English)  and the reader could not reach such references to verify their content. The authors should write in English these references or at least change them with other equivalents in the literature.

Moderate editing of English language required

Author Response

Dear Reviewer,

Many thanks for your valuable comments.

With best regards,

Tomasz Sadowski

Response to the remarks:

1) Check to define all the variables in the added equations

  • All variable was checked and defined

2) Check the typos error in line 673

  • The error was corrected.

3) In Figure 16 (page 21): The comparison between simulated and experimental SZ shapes is not visible!! (i.e., there are no simulation results highlighted!)

  • The simulated shape in this paper is represented by the red dots overlaid on the experimental image.

4) The quality of some figures needs to be improved (e.g., Fig.14, Fig.15(

  • The original quality of these images was used.

4) The authors missed to include this recent work on the effect of friction models on the FSW results (DOI: 10.1177/14644207211070965)  and also a paper on the comparative study between the FSW modelling using ALE and CEL formulations (DOI: 10.1007/978-3-030-86446-0_36)

  • The introduced references were used in article.

5) Refs 159 and 196 are not visible (written in other languages than English)  and the reader could not reach such references to verify their content. The authors should write in English these references or at least change them with other equivalents in the literature.

  • These references were replaced.

Reviewer 4 Report

After revision by the authors in response to the initial review process, I am writing to formally recommend the acceptance of the revised manuscript titled "A Review on Friction Stir Welding/Processing: Numerical Modeling".

Author Response

Dear Reviewer,

Many thanks for acceptance our manuscript.

With best regards,

Tomasz Sadowski